# Bulk-Calibrated Credal Ambiguity Sets:
# Fast, Tractable Decision Making under Out-of-Sample Contamination

**Mengqi Chen** [1]   **Thomas B. Berrett** [1]   **Theodoros Damoulas** [1 2]   **Michele Caprio** [3]

## Abstract

Distributionally robust optimisation (DRO) minimises the worst-case expected loss over an ambiguity set that can capture distributional shifts in out-of-sample environments. While Huber (linear-vacuous) contamination is a classical minimal-assumption model for an $\varepsilon$-fraction of arbitrary perturbations, including it in an ambiguity set can make the worst-case risk infinite and the DRO objective vacuous unless one imposes strong boundedness or support assumptions. We address these challenges by introducing bulk-calibrated credal ambiguity sets: we learn a high-mass bulk set from data while considering contamination inside the bulk and bounding the remaining tail contribution separately. This leads to a closed-form, finite $\mathrm{mean} + \sup$ robust objective and tractable linear or second-order cone programs for common losses and bulk geometries. Through this framework, we highlight and exploit the equivalence between the imprecise probability (IP) notion of upper expectation and the worst-case risk, demonstrating how IP credal sets translate into DRO objectives with interpretable tolerance levels. Experiments on heavy-tailed inventory control, geographically shifted house-price regression, and demographically shifted text classification show competitive robustness-accuracy trade-offs and efficient optimisation times, using Bayesian, frequentist, or empirical reference distributions.

---
[1]Department of Statistics, University of Warwick, Coventry, United Kingdom [2]Department of Computer Science, University of Warwick, Coventry, United Kingdom [3]Department of Computer Science, University of Manchester, Manchester, United Kingdom. Correspondence to: Mengqi Chen <mengqi.chen.2@warwick.ac.uk>.

*Proceedings of the 43rd International Conference on Machine Learning*, Seoul, South Korea. PMLR 306, 2026. Copyright 2026 by the author(s).

## 1. Introduction

Imprecise probability (IP) and distributionally robust optimisation (DRO) both formalise *worst-case* decision-making under *distributional uncertainty*: rather than committing to a single data-generating law, we protect decisions against the worst-case scenario in a set of plausible laws.

For a decision $x \in \mathcal{X}$ and random outcomes $\xi \in \Xi$, we measure performance by the expected loss $\mathbb{E}_{\xi \sim Q}[f_x(\xi)]$, where $f_x$ is a loss function and $Q$ is a probability law for $\xi$. In this paper, distributional uncertainty means that the test-time law $Q$ is only known to lie in a specified set of distributions. In IP theory (Levi, 1980; Walley, 1991; Augustin et al., 2014; Troffaes & de Cooman, 2014; Martin, 2026), this set is a *credal set* $\mathcal{M}$, that is, a convex and (weak$^\star$-) closed set of plausible laws. DRO represents uncertainty via an *ambiguity set* $\mathcal{A}$, often given by a bounded divergence around a nominal centre (Kuhn et al., 2025). In both views, the robust objective is the worst-case expected loss over the uncertainty set; when we take $\mathcal{A} = \mathcal{M}$, this coincides with the IP upper expectation:

$$\overline{\mathbb{E}}_{Q \in \mathcal{M}}\big[f_x(\xi)\big] = \sup_{Q \in \mathcal{M}} \mathbb{E}_{\xi \sim Q}\big[f_x(\xi)\big].$$

The DRO decision rule minimises this worst-case risk over $x \in \mathcal{X}$. Figure 1 demonstrates this relationship (see also Fröhlich & Williamson, 2024).

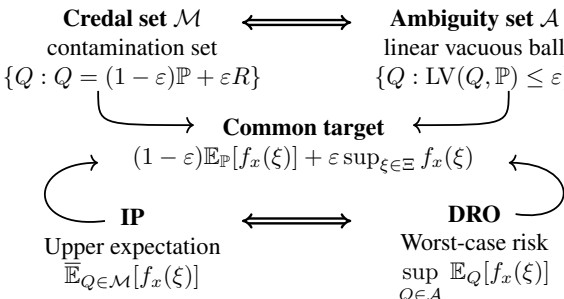

*Figure 1.* Equivalence between IP and DRO, with an example when $\mathcal{A}$ and $\mathcal{M}$ are Huber contamination sets.

A central model in robust statistics and in IP is Huber's $\varepsilon$-contamination (Huber, 1964): a "clean" distribution perturbed by an $\varepsilon$-fraction of a vacuous (uninformative and

arbitrary) distribution. In contrast, much of modern DRO in machine learning is built around $f$-divergences (e.g. general $f$-divergence: Ben-Tal et al. (2013); Namkoong & Duchi (2016); Kullback-Leibler (KL) divergence: Hu & Hong (2013); total variation (TV) distance: Jiang & Guan (2018); Fontem & Ji (2026)) and Wasserstein neighbourhoods (e.g. Mohajerin Esfahani & Kuhn, 2018; Sinha et al., 2018; Gao & Kleywegt, 2023), because they often give smooth objectives and clean dual forms (see the recent review by Kuhn et al. (2025) for a full taxonomy). The influential work by Duchi & Namkoong (2021) notes that divergence-ball DRO formulations are complementary to classical Huber robustness, and calls for a clearer understanding of the connections and contrasts between the two lines of work. Our paper answers this call by translating Huber's $\varepsilon$-contamination set into a DRO objective that remains well posed in unbounded, continuous spaces and is computationally tractable.

Let $\mathbb{P}^\star$ be the unknown data-generating law on $\Xi \subseteq \mathbb{R}^d$, and let $f : \mathcal{X} \times \Xi \to \mathbb{R}$ be the loss function, writing $f_x(\xi) := f(x, \xi)$ for the loss of a decision $x \in \mathcal{X}$. We choose a reference distribution (centre) $\mathbb{P}_c$ based on i.i.d. samples $\mathcal{D} := \xi_{1:n} \overset{\text{i.i.d.}}{\sim} \mathbb{P}^\star$ (for instance, a Bayesian posterior predictive as in Dellaporta et al. (2025; 2026), a frequentist plug-in law $\mathbb{P}_{\hat{\theta}(\mathcal{D})}$ fitted in a parametric family (e.g. Iyengar et al. (2023)), or the empirical distribution $\hat{\mathbb{P}}_n$). Our target is to protect our decision when deployed in an out-of-sample (OOS) environment $\tilde{\mathbb{P}} \neq \mathbb{P}^\star$ modelled by a Huber $\varepsilon$-contamination: $\tilde{\mathbb{P}} = (1 - \varepsilon)\mathbb{P}^\star + \varepsilon\tilde{R}$ with $\tilde{R} \in \mathcal{P}(\Xi)$, the set of all probability laws on $\Xi$.

From an IP perspective, $\varepsilon$ can be interpreted as the probability that the information source is unreliable (Augustin et al., 2014, Section 4.7.3). The upper and lower envelopes of the $\varepsilon$-contaminated credal set can be computed analytically (Wasserman & Kadane, 1990, Example 3), facilitating the derivation of upper and lower expectations. The $\varepsilon$-contamination set also supports robust models with low computational complexity (Caprio, 2025).

In unbounded spaces $\Xi$ with unbounded loss $f_x$, any ambiguity set that includes Huber contamination faces a practical challenge: the worst-case risk contains a supremum over $f_x$, which is infinite unless we restrict the adversary. This issue is recognised in the literature (Chan et al., 2024), and existing contamination-related DRO typically assume that $\Xi$ is bounded (Jiang & Guan, 2018; Fontem & Ji, 2026) or rely on function-class restrictions (e.g., kernel-based DRO in Staib & Jegelka (2019); Zhu et al. (2021)); see Appendix E for a discussion. Our approach avoids requiring bounded $\Xi$ by learning a bulk set $\Xi_0$ with an explicit mass certificate, allocating the contamination budget within $\Xi_0$, and controlling the residual $\Xi_0^c$ by imposing a moment condition. We learn $\Xi_0(\mathcal{D})$ from data in the "accuracy–confidence" form:

$$\Pr\{\mathbb{P}^\star(\Xi_0) \geq 1 - \gamma\} \geq 1 - \delta, \tag{1}$$

for some $\gamma, \delta \in [0, 1]$, where $\Pr$ is the probability measure on the sample space $\Xi^n$ induced by the random training sample $\mathcal{D} \sim (\mathbb{P}^\star)^n$ used to construct the data-dependent set $\Xi_0(\mathcal{D})$. Certifiable values of $(\gamma, \delta)$ depend on sample size, which we will formalise in Lemma 3.2. This leads to a simple robust objective that trades off the mean under our in-bulk centre distribution with a worst-case loss over $\Xi_0$, and supports efficient linear programming (LP) or second-order cone programming (SOCP) formulations for common choices of $f_x$ and $\Xi_0$.

Classical contamination neighbourhoods are closely linked but often studied separately in DRO. The Huber contamination set coincides with the *forward* linear-vacuous (LV) ball in IP, where LV is a one-sided divergence from IP theory (Proposition 2.3), and its worst-case risk is in the mean $+$ sup form (Theorem 2.1). The *reverse* LV ball (the set of $Q$ where $\mathbb{P}^\star$ can be written as $(1 - \varepsilon)Q + \varepsilon R$, i.e. trimming relative to $\mathbb{P}^\star$) yields CVaR, motivating outlier-robust DRO (e.g. Zhai et al., 2021; Nietert et al., 2023; 2024; Li et al., 2025). The symmetric TV ball contains both and combines a CVaR term with a sup term (Jiang & Guan, 2018). We give an alternative proof of this TV objective via a similarity decomposition argument (Alvarez-Esteban et al., 2012), providing a unifying perspective on how adding adversarial mass (forward) and deleting low-loss mass (reverse) interact in contamination neighbourhoods.

Our main contributions include:

1. We introduce bulk-calibrated ambiguity sets that make forward Huber contamination meaningful, instead of vacuous, in unbounded sample spaces with unbounded losses. This setting encompasses commonly used loss functions and data domains, and is relevant for ambiguity sets that admit Huber contamination.

2. We derive a finite worst-case risk objective for the ambiguity set, with tractable LP/SOCP counterparts for common losses and bulk geometries (Section 2, Table 1). We also provide data-driven bulk calibration with finite-sample bulk-mass guarantees and a high-probability risk certificate (Section 3); the calibration is computationally efficient ($O(m \log m)$) and supports flexible and block-wise specification of bulk sets.

3. In experiments (Section 5), LV-based credal ambiguity sets yield superior OOS performance and shorter convex-program solve times than classical DRO baselines on heavy-tailed newsvendor synthetic problems and two real-world ML tasks (California housing regression and CivilComments text classification). Moreover, we demonstrate flexible centre choices (Bayesian predictive, frequentist plug-in, or empirical).

4. We highlight and exploit the link between IP credal sets and DRO by viewing the IP upper expectation as

a DRO worst-case risk. While much of the classical IP literature studies $\varepsilon$-contaminated credal sets in discrete or finite settings, we extend these constructions to continuous, unbounded spaces, providing a practical protocol for transferring frameworks and guarantees between the IP and DRO communities.

In Section 2 we define the bulk-restricted credal ambiguity set for a given bulk set $\Xi_0$. Section 3 introduces our data-driven bulk-set calibration procedure, and Section 4 discusses validation-based selection of the tolerance level $\varepsilon$ when deployment samples are not observed. Section 5 studies the empirical performance of our bulk-calibrated credal ambiguity set in synthetic and real-world experiments, and provides practical insights for choosing the bulk-set geometry and hyperparameters. Finally, we discuss limitations and future work in Section 6. Code to reproduce experiments can be found at `https://github.com/MengqiChenMC/credal-ambiguity-sets-code-repo`.

## 2. Bulk-Restricted LV Credal Ambiguity Set

In this section, we start by defining the bulk-restricted credal ambiguity set and providing two crucial quantities associated with it: the worst-case risk and worst-case distribution. We then show that the ambiguity set can be written as a forward-LV ball, which places it in a familiar divergence-ball DRO form. Finally, we link the LV ball with two other commonly used contamination neighbourhoods (reverse LV and TV balls), providing a unifying interpretation.

### 2.1. Definition and Properties

Let $\mathbb{P}_c$ be a chosen centre distribution on $\Xi$ constructed from data $\mathcal{D}$. For example: a Bayesian posterior predictive, $\mathbb{P}_c(\cdot) = \int_\Theta p(\cdot \mid \theta)\Pi(\mathrm{d}\theta \mid \mathcal{D})$, a frequentist plug-in predictive $\mathbb{P}_c = \mathbb{P}_{\hat{\theta}}$ for a parametric family $\{\mathbb{P}_\theta\}_{\theta \in \Theta}$ with $\hat{\theta} = \hat{\theta}(\mathcal{D})$, or an empirical distribution $\mathbb{P}_c = \hat{\mathbb{P}}_n = \frac{1}{n}\sum_{i=1}^n \delta_{\xi_i}$, which we use to form sample-average approximation (SAA) estimates of expectations under $\mathbb{P}^\star$; see Remark B.3 in Appendix B.6.

Let $\Xi_0 \subseteq \Xi$ be a bounded measurable set. Write normalised restrictions for any probability measure $\mathbb{P}$ with $\mathbb{P}(\Xi_0) > 0$:

$$\mathbb{P}_{\Xi_0}(A) := \frac{\mathbb{P}(A \cap \Xi_0)}{\mathbb{P}(\Xi_0)}.$$

We start by defining the general notion of the worst-case risk and our credal ambiguity set.

Given a decision $x \in \mathcal{X}$, an ambiguity set $\mathcal{A} \subseteq \mathcal{P}(\Xi)$, and a measurable loss function $f : \mathcal{X} \times \Xi \to \mathbb{R}$ (writing

$f_x(\xi) := f(x, \xi)$), the *worst-case risk* is

$$\mathcal{R}_{\mathcal{A}}(f_x) := \sup_{Q \in \mathcal{A}} \mathbb{E}_{\xi \sim Q}[f_x(\xi)]$$

For $\varepsilon \in [0, 1]$, we define our *credal ambiguity set* as the *support-restricted LV set*, which coincides with the Huber-contamination set on $\Xi_0$:

$$\mathcal{A}_{\varepsilon,\Xi_0}^{\mathrm{LV}}(\mathbb{P}_{c,\Xi_0}) := \{(1-\varepsilon)\mathbb{P}_{c,\Xi_0} + \varepsilon R : R \in \mathcal{P}(\Xi_0)\}.$$

Under $\mathcal{A}_{\varepsilon,\Xi_0}^{\mathrm{LV}}(\mathbb{P}_{c,\Xi_0})$, the worst-case risk yields a straightforward closed-form, akin to the upper expectation for an $\varepsilon$-contaminated model in IP theory (Augustin et al., 2014, Equation 4.10):

**Theorem 2.1** (Worst-case risk of support-restricted LV set, proved in Appendix B.1).

$$\mathcal{R}_{\mathcal{A}_{\varepsilon,\Xi_0}^{\mathrm{LV}}(\mathbb{P}_{c,\Xi_0})}(f_x) = (1-\varepsilon)\mathbb{E}_{\xi \sim \mathbb{P}_{c,\Xi_0}}[f_x(\xi)] + \varepsilon \sup_{\xi \in \Xi_0} f_x(\xi).$$

A related objective is trade-off robust optimisation (TRO) (Tsang & Shehadeh, 2025), which mixes the empirical distribution with a chosen ambiguity set to interpolate between empirical risk and a DRO worst-case risk. Although this can resemble a $\mathrm{mean} + \sup$ form, TRO is based on empirical risk minimisation and explicitly treats the mixing weight as a conservatism parameter for existing DRO ambiguity sets rather than as a Huber contamination ratio.

Our *decision rule* is $\hat{x} \in \operatorname{argmin}_{x \in \mathcal{X}} \mathcal{R}_{\mathcal{A}_{\varepsilon,\Xi_0}^{\mathrm{LV}}(\mathbb{P}_{c,\Xi_0})}(f_x)$.

The truncated expectation $\mathbb{E}_{\xi \sim \mathbb{P}_{c,\Xi_0}}[f_x(\xi)]$ can be approximated via SAA with rejection sampling when it is not available analytically. A convergence diagnostic of this approximation for our synthetic newsvendor experiment (Section 5.1) is reported in Appendix F.7 (Figure 10).

Theorem 2.1 immediately gives us the following corollary that identifies the worst-case distribution in $\mathcal{A}_{\varepsilon,\Xi_0}^{\mathrm{LV}}(\mathbb{P}_{c,\Xi_0})$, drawing a parallel with the upper envelope of an $\varepsilon$-contaminated credal set (Wasserman & Kadane, 1990, Example 3). Unsurprisingly, the adversary is attained by putting $\varepsilon$ of the mass on the set of $\xi$ that attains the supremum inside $\Xi_0$.

**Corollary 2.2** (Worst-case distribution, proved in Appendix B.2). *Assume $\varepsilon > 0$; otherwise $\mathcal{A}_{\varepsilon,\Xi_0}^{\mathrm{LV}}(\mathbb{P}_{c,\Xi_0}) = \{\mathbb{P}_{c,\Xi_0}\}$. Fix $x \in \mathcal{X}$ and assume $M_x := \sup_{\xi \in \Xi_0} f_x(\xi) < \infty$. Let $\Xi_0^{\max}(x) := \{\xi \in \Xi_0 : f_x(\xi) = M_x\}$. If there exists $R^\star \in \mathcal{P}(\Xi_0)$ with $R^\star(\Xi_0^{\max}(x)) = 1$, then $Q^\star = (1 - \varepsilon)\mathbb{P}_{c,\Xi_0} + \varepsilon R^\star$ is the worst-case distribution for $\mathcal{A}_{\varepsilon,\Xi_0}^{\mathrm{LV}}(\mathbb{P}_{c,\Xi_0})$. If $\Xi_0^{\max}(x) = \emptyset$, then the worst-case distribution is not attained.*

### 2.2. Credal Ambiguity Set as Forward LV ball

The next proposition gives the classification of $\mathcal{A}_{\varepsilon,\Xi_0}^{\mathrm{LV}}(\mathbb{P}_{c,\Xi_0})$ via the LV distortion, linking the classical contamination set

with a DRO-like divergence-based definition.

**Proposition 2.3** (LV distortion classification of the contamination set, proved in Appendix B.3)**.** *Fix $\varepsilon \in [0,1]$. The following sets are equal:*

$$\mathcal{A}_{\varepsilon,\Xi_0}^{\mathrm{LV}}(\mathbb{P}_{c,\Xi_0}) = \{(1-\varepsilon)\mathbb{P}_{c,\Xi_0} + \varepsilon R : R \in \mathcal{P}(\Xi_0)\}$$
$$\mathcal{B}_{\mathrm{LV}}^{\varepsilon}(\mathbb{P}_{c,\Xi_0}) = \{Q \in \mathcal{P}(\Xi_0) : \mathrm{LV}(Q, \mathbb{P}_{c,\Xi_0}) \leq \varepsilon\}$$

*where $\mathcal{B}_{\mathrm{LV}}^{\varepsilon}(\mathbb{P}_{c,\Xi_0})$ is the ball of radius $\varepsilon$ around $\mathbb{P}_{c,\Xi_0}$ defined by the IP notion of LV distortion (Montes et al., 2020):*

$$\mathrm{LV}(Q,\mathbb{P}) := \sup_{A: \mathbb{P}(A)>0} \frac{\mathbb{P}(A) - Q(A)}{\mathbb{P}(A)}.$$

### 2.3. Links to other Contamination Neighbourhoods

A closely related and popular construction in outlier-robust DRO is the reverse LV ball, defined as $\{Q \in \mathcal{P}(\Xi) : \exists R \in \mathcal{P}(\Xi) \text{ s.t. } \mathbb{P} = (1-\varepsilon)Q + \varepsilon R\}$ (Zhai et al., 2021; Nietert et al., 2023; Li et al., 2025). We compare the worst-case risks of the forward and reverse LV balls for a generic centre $\mathbb{P}$ and space $\Xi$; in our bulk-restricted setting we take $(\mathbb{P}, \Xi) = (\mathbb{P}_{c,\Xi_0}, \Xi_0)$.

$$\text{Forward LV}: \quad (1-\varepsilon)\mathbb{E}_{\mathbb{P}}[f] + \varepsilon \sup_{\Xi} f, \quad (2)$$

$$\text{(Theorem 2.1 applied to } \mathbb{P} \text{ and } \Xi)$$

$$\text{Reverse LV}: \quad \mathrm{CVaR}_{1-\varepsilon}^{\mathbb{P}}(f). \quad (3)$$
$$\text{Föllmer \& Schied (2016, Theorem 4.52)}$$

Here $\mathrm{CVaR}_{1-\varepsilon}^{\mathbb{P}}(f)$ denotes the conditional value at risk of $f$ at level $1-\varepsilon$ under $\mathbb{P}$. The TV distance is symmetric, and an $\varepsilon$-TV ball contains both the forward and reverse $\varepsilon$-LV balls. The worst-case risk of the $\varepsilon$-TV ball is closely related to (2) and (3):

**Proposition 2.4** (Kuhn et al. (2025), Proposition 6.13)**.**

$$\mathcal{R}_{\mathcal{B}_{\mathrm{TV}}^{\varepsilon}(\mathbb{P})}(f) = (1-\varepsilon)\mathrm{CVaR}_{1-\varepsilon}^{\mathbb{P}}(f) + \varepsilon \sup_{\Xi} f,$$

*where $\mathcal{B}_{\mathrm{TV}}^{\varepsilon}(\mathbb{P}) := \{Q \in \mathcal{P}(\Xi) : \mathrm{TV}(Q,\mathbb{P}) \leq \varepsilon\}$.*

Figure 2 visualises the corresponding worst-case distributions. Similar $\mathrm{CVaR}+\max$ forms arise in Lévy–Prokhorov-type DRO and recover TV-DRO as a special case (Bennouna et al., 2025). Intuitively, the reverse LV ball corresponds to "deleting" up to an $\varepsilon$-fraction of low-loss mass from $\mathbb{P}$ (CVaR tail reweighting), while the forward LV ball corresponds to "adding" an $\varepsilon$-fraction of arbitrary mass on worst-case states. The TV ball allows both effects simultaneously, and its worst-case risk is a convex combination of the reverse-LV (CVaR) behaviour and the forward-LV (sup) behaviour. These neighbourhoods are typically studied in isolation. To investigate the close relationship between them, we give an alternative proof of Proposition 2.4 to that of Kuhn et al.

(2025) in Appendix B.4. This proof leverages Theorem 2.1, equation (3), and the similarity decomposition for total variation (Alvarez-Esteban et al., 2012, Proposition 2), allowing us to explore the intrinsic link between the three contamination neighbourhoods.

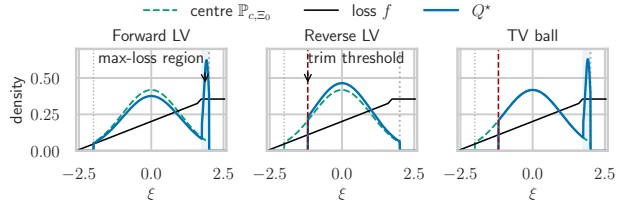

*Figure 2.* Worst-case distributions $Q^\star$ for $\sup_Q \mathbb{E}_{\xi \sim Q}[f]$ under forward LV, reverse LV, and TV balls around a centre $\mathbb{P}_{c,\Xi_0}$ (loss $f$ plateaus at a small region to avoid Dirac deltas).

So far, we have fixed $(\Xi_0, \mathbb{P}_c, \varepsilon)$ and studied the tractable worst-case objective. We now turn to guarantees under the unknown $\mathbb{P}^\star$ when $\Xi_0$ and $\mathbb{P}_c$ are data-dependent.

## 3. Calibrating $\Xi_0$ via Score Selection

We aim to construct the bulk set $\Xi_0$ from $\mathcal{D}$ and certify the finite-sample coverage guarantee of condition (1).

### 3.1. Score Fitting

We observe i.i.d. data $\mathcal{D} = \{\xi_i\}_{i=1}^n \sim (\mathbb{P}^\star)^n$ in $\Xi \subseteq \mathbb{R}^d$. We target bulk mass $1-\gamma \in (0,1)$ with confidence $1-\delta \in (0,1)$. We split the sample into two independent parts:

$$\mathcal{D}_{\mathrm{fit}} = \{\xi_j\}_{j=1}^{n-m}, \quad \mathcal{D}_{\mathrm{select}} = \{\tilde{\xi}_j\}_{j=1}^m.$$

To keep the optimisation tractable later, we work with a family of bulk sets induced by a scalar score,

$$\{\Xi_0(t) : t \in \mathbb{R}\}, \quad \Xi_0(t) = \{\xi : s(\xi) \leq t\},$$

where $s : \Xi \to \mathbb{R}$ is a measurable *score function* built on $\mathcal{D}_{\mathrm{fit}}$, so that increasing $t$ enlarges $\Xi_0(t)$. Typical choices are: **ellipsoid** via $s(\xi) = \|\Sigma_{\mathrm{fit}}^{-1/2}(\xi - \mu_{\mathrm{fit}})\|_2$; or **box** via $s(\xi) = \max_i |\xi_i - \mu_{\mathrm{fit},i}|/w_i$. Here, $(\mu_{\mathrm{fit}}, \Sigma_{\mathrm{fit}})$ are estimated from $\mathcal{D}_{\mathrm{fit}}$ only. For common choices of $f_x$, these bulk sets give $\sup_{\xi \in \Xi_0(t)} f_x(\xi)$ in closed form and admit LP/SOCP reformulations: see Table 1 for examples.

### 3.2. Selection

Given $s(\cdot)$, let $Z := s(\xi)$ for $\xi \sim \mathbb{P}^\star$ and let $F(t) := \mathbb{P}^\star(Z \leq t)$ denote the theoretical CDF of the scores. Form the empirical CDF of the selection scores $F_m(t) := \frac{1}{m}\sum_{j=1}^m \mathbf{1}\{s(\tilde{\xi}_j) \leq t\}$, $\tilde{\xi}_j \in \mathcal{D}_{\mathrm{select}}$, where $\mathbf{1}\{\cdot\}$ denotes the indicator function. By the Dvoretzky–Kiefer–Wolfowitz (DKW) inequality (Dvoretzky et al., 1956; Massart, 1990),

$$\Pr\left\{\sup_{t \in \mathbb{R}} |F_m(t) - F(t)| \leq r_{m,\delta}\right\} \geq 1-\delta, \quad (4)$$

where $r_{m,\delta} := \sqrt{\frac{1}{2m}\log\left(\frac{2}{\delta}\right)}$. Define the uniform one-sided lower envelope

$$L^{\mathrm{DKW}}(t) := \left[F_m(t) - r_{m,\delta}\right]_+,$$

where $[a]_+ := \max\{a, 0\}$ for a generic $a \in \mathbb{R}$. Select the smallest $t$ that clears the target bulk mass:

$$\hat{t}_{\mathrm{DKW}} := \inf\{t : L^{\mathrm{DKW}}(t) \geq 1 - \gamma\}, \quad \widehat{\Xi}_0 := \Xi_0(\hat{t}_{\mathrm{DKW}}).$$

Such $\hat{t}_{\mathrm{DKW}}$ exists when $\gamma \geq r_{m,\delta}$, the smallest certifiable tail budget given $m$ and $\delta$. In practice, $\hat{t}_{\mathrm{DKW}}$ can be estimated via the $(1 - \gamma + r_{m,\delta})$-empirical quantile of the selection scores and computed in $O(m\log m)$ time (see Algorithm 1 in Appendix A). We will show that $\widehat{\Xi}_0$ satisfies Condition (1) via Lemma 3.2.

*Remark* 3.1. Our method is inspired by conformal prediction: we calibrate a scalar score on a data subset, and then pick the smallest threshold that clears a target level. However, conformal prediction delivers finite-sample *marginal coverage* for a future draw under exchangeability, whereas our risk certificate requires a *high-probability bulk-mass*, which is delivered via a uniform concentration bound for the score CDF. We also refer the reader to Johnstone & Cox (2021) for a detailed discussion on nonconformity scores and their geometric implications for robust optimisation.

**Lemma 3.2** (High-probability bulk-mass certificate, proved in Appendix B.5). *Let $\mathcal{D}_{\mathrm{select}} = \{\tilde{\xi}_j\}_{j=1}^m$ be i.i.d. from $\mathbb{P}^\star$ and independent of the score $s(\cdot)$ and $\mathcal{D}_{\mathrm{fit}}$. Define $F_m$, $r_{m,\delta}$, $L^{\mathrm{DKW}}(\cdot)$ as above. When $\gamma \geq r_{m,\delta}$, $\hat{t}_{\mathrm{DKW}}$ exists and*

$$\Pr\left\{\mathbb{P}^\star\left(\Xi_0(\hat{t}_{\mathrm{DKW}})\right) \geq 1 - \gamma\right\} \geq 1 - \delta.$$

*Remark* 3.3. Our bulk set calibration extends directly to *blockwise* bulk sets built as intersections. If $\xi = (\xi^{(1)}, \ldots, \xi^{(k)})$ with $\sum_{i=1}^k d_i = d$, we may choose separate bulk geometries and scores $s_i$ on each dimension block and calibrate thresholds using parameters $(\gamma_i, \delta_i)$ separately. $\widehat{\Xi}_0 = \bigcap_{i=1}^k \{\xi : s_i(\xi^{(i)}) \leq \hat{t}_i\}$ still yields a valid bulk-mass certificate as long as $\sum_{i=1}^k \gamma_i \leq \gamma$ and $\sum_{i=1}^k \delta_i \leq \delta$. This is relevant when the dimensions are asymmetric or heterogeneous (the regression problem in Section 5.2 will be an example). See Lemma C.1 in the Appendix for more details.

Lemma 3.2 certifies the mass of the bulk set calibrated via our DKW-based procedure, but it does not imply uniqueness of the bulk set. In fact, the geometry of the bulk set, decided by the score function, is an important modelling choice. Two major considerations are: (i) keeping $\sup_{\xi \in \Xi_0} f_x(\xi)$ tractable (see examples in Table 1), and (ii) aligning the bulk set geometry with the type of contamination we want to protect against. We will discuss more on the latter in Section 5.2.1.

While naive confidence-set truncations (e.g., the three-sigma rule) can also provide a bounded $\Xi_0$, they cannot, in general, certify (1), so the tail mass outside $\Xi_0$ is uncontrolled. See Appendix E for counterexamples. With the bulk-mass guarantee in Lemma 3.2, we now give a high-probability risk certificate by separating bulk robustness from tail control.

**Theorem 3.4** (Risk certificate, proved in Appendix B.6). *Let $\mathbb{P}^\star$ denote the training distribution and let $\mathbb{P}_c$ be a data-driven centre distribution with $\mathbb{P}_c(\Xi_0) > 0$. Assume that, with probability at least $1 - \delta$, we have $\mathbb{P}^\star(\Xi_0) \geq 1 - \gamma$ for $\gamma \in [0, 1)$. Suppose $\widetilde{\mathbb{P}} = (1 - \varepsilon^\star)\mathbb{P}^\star + \varepsilon^\star \widetilde{R}$, $\varepsilon^\star \in [0, 1)$, $\widetilde{R} \in \mathcal{P}(\Xi)$. Assume the in-bulk training centre mismatch $\varepsilon_c := \mathrm{LV}(\mathbb{P}^\star_{\Xi_0}, \mathbb{P}_{c,\Xi_0}) < 1$. Assume further that $f_x$ is nonnegative[1] and that there exists $p > 1$ with $M_p(x) := \left(\mathbb{E}_{\xi \sim \widetilde{\mathbb{P}}}[|f_x(\xi)|^p]\right)^{1/p} < \infty$ for all $x \in \mathcal{X}$. Let $\rho_{\Xi_0} := \widetilde{R}(\Xi_0)/\widetilde{\mathbb{P}}(\Xi_0)$. With probability at least $1 - \delta$, the following bound holds simultaneously for all $x \in \mathcal{X}$:*

$$\mathbb{E}_{\xi \sim \widetilde{\mathbb{P}}}[f_x(\xi)] \leq (1 - \varepsilon_{\mathrm{eff}})\mathbb{E}_{\xi \sim \mathbb{P}_{c,\Xi_0}}[f_x(\xi)] + \varepsilon_{\mathrm{eff}} \sup_{\xi \in \Xi_0} f_x(\xi)$$

$$+ M_p(x)\left((1 - \varepsilon^\star)\gamma + \varepsilon^\star\left(1 - \widetilde{R}(\Xi_0)\right)\right)^{1/q},$$

*where $q := p/(p - 1)$ and $\varepsilon_{\mathrm{eff}} := \varepsilon_c + \varepsilon^\star\rho_{\Xi_0} - \varepsilon^\star\varepsilon_c\rho_{\Xi_0}$ is the effective in-bulk tolerance.*

The effective tolerance $\varepsilon_{\mathrm{eff}}$ is an upper bound on the deployment in-bulk distortion $\tilde{\varepsilon} := \mathrm{LV}(\widetilde{\mathbb{P}}_{\Xi_0}, \mathbb{P}_{c,\Xi_0})$, and it combines in-bulk centre mismatch $\varepsilon_c$ and the portion of deployment contamination that lands inside the bulk $\varepsilon^\star\rho_{\Xi_0}$. See Remark B.3 in Appendix B.6 for a discussion on sufficient conditions for $\varepsilon_c$ to vanish asymptotically. We impose no additional restriction on the contaminant $\widetilde{R}$ beyond the global $p$-moment condition on $\widetilde{\mathbb{P}}$. If $\widetilde{R}(\Xi_0)$ is large, then more of the contamination lies in-bulk (increasing $\varepsilon_{\mathrm{eff}}$), while the tail term improves because the out-of-bulk mass $\varepsilon^\star(1 - \widetilde{R}(\Xi_0))$ shrinks. Without deployment samples, Theorem 3.4 serves as a structural decomposition since $\varepsilon^\star$, $\rho_{\Xi_0}$ and $M_p(x)$ are generally not observable from the training sample alone.

# 4. Tolerance Selection: Calibrating $\varepsilon$

In the setting of our paper, training data are i.i.d. from $\mathbb{P}^\star$, and the unknown test-time environment may follow a contaminated law $\widetilde{\mathbb{P}} = (1 - \varepsilon^\star)\mathbb{P}^\star + \varepsilon^\star\tilde{R}$. Without extra information on $\widetilde{\mathbb{P}}$, the true contamination level $\varepsilon^\star$ is not identifiable from training data alone. We therefore treat $\varepsilon$ as a robustness budget that controls the level of contamination protection in our LV objective, analogous to how Wasserstein DRO treats its radius as a tunable robustness budget, and similar to the IP intuition discussed in the introduction. In practice,

---

[1] $f_x$ is nonnegative for all losses considered in this paper. If $f_x$ can be negative, the same result holds for $|f_x|$.

*Table 1.* Closed-form values $\sup_{\xi \in \Xi_0(t_k)} f_x(\xi)$ for common loss $f_x$ under two bulk-set geometries. Here the ellipsoid $\Xi_0^{\text{ellip}}(t_k) = \{\xi : \|\Sigma_{\text{fit}}^{-1/2}(\xi - \mu_{\text{fit}})\|_2 \le t_k\}$ and the box $\Xi_0^{\text{box}}(t_k) = \{\xi : \max_i |(\xi_i - \mu_{\text{fit},i})/w_i| \le t_k\}$. Write $C_x := a_x^\top \mu_{\text{fit}} + b_x$, $m_2(a_x) := \|\Sigma_{\text{fit}}^{1/2} a_x\|_2$, and $m_1(a_x) := \sum_i w_i |a_{x,i}|$. The quantities are derived from support-function calculations in convex optimisation (Boyd & Vandenberghe, 2004) and proved in Appendix B.7.

| $f_x(\xi)$ | Ellipsoid $\sup_{\xi \in \Xi_0^{\text{ellip}}(t_k)} f_x(\xi)$ | Box $\sup_{\xi \in \Xi_0^{\text{box}}(t_k)} f_x(\xi)$ |
|---|---|---|
| Linear: $a_x^\top \xi + b_x$ | $C_x + t_k m_2(a_x)$ | $C_x + t_k m_1(a_x)$ |
| ReLU: $\max\{0, a_x^\top \xi + b_x\}$ | $\max\{0, C_x + t_k m_2(a_x)\}$ | $\max\{0, C_x + t_k m_1(a_x)\}$ |
| Absolute value: $|a_x^\top \xi + b_x|$ | $|C_x| + t_k m_2(a_x)$ | $|C_x| + t_k m_1(a_x)$ |
| Piecewise-linear: $\max_{j \le J}\{a_{x,j}^\top \xi + b_{x,j}\}$ | $\max_{j \le J}\{a_{x,j}^\top \mu_{\text{fit}} + b_{x,j} + t_k m_2(a_{x,j})\}$ | $\max_{j \le J}\{a_{x,j}^\top \mu_{\text{fit}} + b_{x,j} + t_k m_1(a_{x,j})\}$ |

we choose $\varepsilon$ by validation tuning: in the California housing experiment (Section 5.2), we use geo-block cross-validation (CV), and in the CivilComments experiment (Section 5.3), we use minimax validation selection.

Appendix D gives a simple and computationally inexpensive score-based alignment diagnostic for the in-bulk centre mismatch $\varepsilon_c$ (Lemma D.1); it also provides a lemma that characterises the LV distortion (Lemma D.3) if deployment samples are available (this is not the setting in our paper).

## 5. Experiments & Empirical Evidence

### 5.1. Bayesian Centre: Heavy-tailed Synthetic Newsvendor with Demand Spikes

We study a $d$-item newsvendor problem. The goal is to choose the order sizes $x \in \mathbb{R}_+^d$ before observing the random demands $\xi \in \mathbb{R}^d$ of items, which requires balancing holding costs for over-ordering against backorder costs for under-ordering. The per-sample cost is

$$f_x(\xi) = \sum_{j=1}^d \left(h[x_j - \xi_j]_+ + b[\xi_j - x_j]_+\right), \quad (5)$$

with holding cost $h = 3$ and backorder cost $b = 8$ (following Shapiro et al. (2023); Dellaporta et al. (2025)). We set $d = 5$ and generate training data i.i.d. from a multivariate Student-$t$ distribution with $\nu = 3$ degrees of freedom. We fit the Bayesian model with Student-$t$ likelihood and a normal-inverse-Wishart prior, and generate posterior predictive samples via a Gibbs sampler (details in Appendix F).

To model rare demand surges (e.g. a sudden promotion or media exposure), we contaminate the test distribution. For contamination levels $\varepsilon_{\text{cont}} \in \{0, 0.1, 0.2\}$, we generate an $\varepsilon_{\text{cont}}$ fraction of test points from a "spike" component (Gaussian with a shifted mean, see Appendix F).

We compare LV to KL-based Bayesian DRO baselines (KL-BDRO (Shapiro et al., 2023), KL-BAS$_{\text{PP}}$ (Dellaporta et al., 2025)[2]), the empirical baseline KL-Empirical (Hu

& Hong, 2013) and the outlier-robust Wasserstein baseline OR-WDRO (Nietert et al., 2023). The Bayesian methods are approximated via SAA using a fixed sampling budget $M = 2500$. We sweep a range of the tolerance parameter $\varepsilon$ and report the OOS performance. For OR-WDRO, we fix the Wasserstein radius $\rho = 5$ and sweep the outlier fraction $\varepsilon_{\text{OR}} \in [0, 0.5)$ (additional $\rho$ value sweep is in Appendix F.6). We report (i) mean–variance frontiers of the OOS cost, and (ii) the mean–standard-deviation criterion MSD $:= \frac{1}{2}(\mu + \sigma)$, where $\mu$ and $\sigma$ are the OOS mean and standard deviation (SD) in Figure 3.

*Table 2.* Run times (seconds) for the Student-$t$ newsvendor experiments across 100 independent replications. Values reported are mean (SD). Test contamination does not affect runtime, so we report a single table for 0% contamination.

| Algorithm | sampling | solve | total |
|---|---|---|---|
| LV (ours) | 1.0 (0.0) | **0.3** (0.0) | **1.3** (0.0) |
| KL-BAS$_{\text{PP}}$ | 1.0 (0.0) | 2.6 (0.4) | 3.6 (0.4) |
| KL-BDRO | **0.1** (0.0) | 2.3 (0.4) | 2.4 (0.4) |
| KL-Empirical | – | 1.7 (0.3) | 1.7 (0.3) |
| OR-WDRO | – | 23.5 (5.0) | 23.5 (5.0) |

Across contaminated test settings (middle and right columns of Figure 3), LV provides the strongest mean–variance trade-off and achieves the lowest MSD values. In the setting without contamination (left column), LV can be too conservative when $\varepsilon_{\text{LV}}$ is large; in this regime OR-WDRO and KL-BDRO attain lower MSD. However, the minimum MSD achieved by LV remains similar to the best baselines even without contamination.

Although the frontiers of KL-BDRO are close to those of LV at small $\varepsilon_{\text{KL}}$ tolerances, it saturates once the $\varepsilon_{\text{KL}}$ exceeds a finite-scenario limit (the OOS mean–variance points for $\varepsilon_{\text{KL}} = 5, 10$ are overlapping); see Remark F.1 in Appendix F. LV continues to trade OOS mean for reduced OOS variance and can yield lower MSD under contamination.

Table 2 reports end-to-end run times under the sampling

---

[2]KL-BAS$_{\text{PE}}$ (Dellaporta et al., 2025) is not applicable because the Student-$t$ likelihood is not in the exponential family.

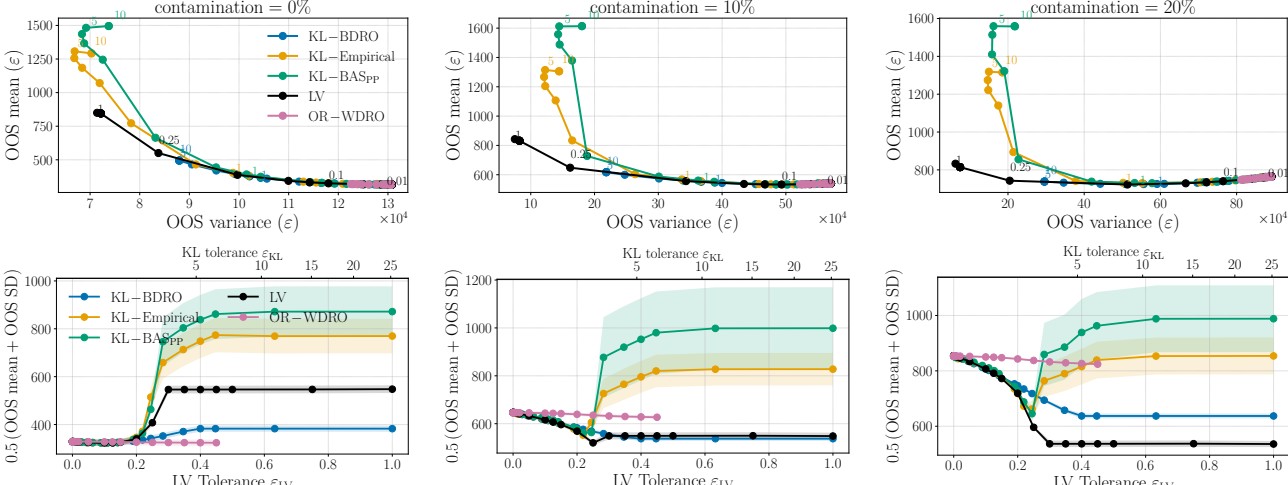

*Figure 3.* Student-$t$ newsvendor (cost: lower-left is better). Top row: OOS mean–variance frontiers for a range of $\varepsilon_{LV} \in (0, 1]$; $\varepsilon_{KL} \in (0, 25]$. OR-WDRO uses $\varepsilon_{OR}$ in $(0, 0.5)$. Each point represents one $\varepsilon$ value. Some $\varepsilon_{LV}$ and $\varepsilon_{KL}$ are marked. Bottom row: MSD $= \frac{1}{2}$(OOS mean + OOS SD) versus $\varepsilon$; shaded regions: 95% confidence bands. From left to right: $\{0, 0.1, 0.2\}$ contamination.

budgets used for Figure 3. LV is fastest because it is *sample-efficient*, which means that it can reach its OOS frontier with far fewer SAA samples than the KL-based baselines. In our implementation, LV solves a fixed-size truncated SAA with size $0.5M$ (a conservative choice for rejection sampling)[3], whereas KL-BDRO and KL-BAS$_{PP}$ use all $M$ samples in their SAA. Importantly, even with this smaller solver input, LV matches or improves OOS performance; the sample efficiency analysis in Appendix F.7 shows that LV's performance is already stable with as few as 50 truncated samples, while KL-based Bayesian methods require much larger SAA budgets to stabilise.

To assess the impact of the tail-budget hyperparameter $\gamma$, we perform a sensitivity analysis in Appendix F.8, which shows that the performance of LV remains stable under a reasonable range. The analysis suggests choosing $\gamma$ just above the minimum certifiable value and leaving a small tail outside the bulk (about 10 scores, so $\gamma \approx r_{m,\delta} + 10/m$) to balance stability and effectiveness.

### 5.2. Frequentist Centre: California Housing Regression under Geographic Deployment Shift

We study the California housing (Pace & Barry, 1997) linear regression task under a geographic deployment shift. The goal is to predict the house value from covariates and extrapolate from an Eastern region to a Western region, which shifts the coastal–inland mix. We train and tune on the Eastern 50%, evaluate on the Western 20%, and exclude

---

[3] Because the number of accepted samples after truncation is random and our optimisation solver requires a fixed problem size.

the intermediate 30% band to alleviate spatial correlation between training and testing. Our centre distribution for LV is a frequentist Gaussian copula centre (Liu et al., 2009), and the bulk set is $\Xi_{0,X} \times \Xi_{0,Y}$, where $\Xi_{0,X}$ is an ellipsoid over the features and $\Xi_{0,Y}$ is an interval over housing prices (see Remark 3.3). Details on our frequentist centre are given in Appendix G.2. Hyperparameters are selected via geo-block CV inside the East region (Appendix G.3). We report test mean absolute error (MAE), root mean squared error (RMSE), the 98th percentile of absolute error $p_{98}(|y - \hat{y}|)$, and the 2% tail risk $\text{CVaR}_{2\%}(|y - \hat{y}|)$.

#### 5.2.1. BULK-SET GEOMETRY CHOICE

In regression under deployment shift, it is natural for the conditional law $Y \mid X$ to change. Using $\Xi_{0,X} \times \Xi_{0,Y}$ decouples covariate geometry from outcome variability, so the tolerance level can trade off mean error against adverse outcomes within a plausible range. Appendix G.7 explains that a joint ellipsoid bulk set in $(X, Y)$ is ineffective for regression because it can hard-code the training $Y - X$ dependence; additionally, box-type support assumptions can be overly conservative because they can allow every dimension to move independently for the adversary, as we discuss in Appendix E.1 and Appendix G.7.

In practice, we advocate for the following *rule of thumb*: use Mahalanobis scores by default; use blockwise bulk sets to separate asymmetric or heterogeneous dimensions in problems such as regression; and only use box scores when independent coordinate-wise deviations are the intended adversary.

### 5.2.2. EXPERIMENT RESULTS

We compare LV to two DRO baselines that trade off mean error and tail error: CVaR and Wasserstein (Chen & Paschalidis, 2018). All methods use a linear predictor and the absolute loss as the base loss. We also include ERM over mean absolute loss and a ridge regression baseline.

*Table 3.* California housing under geographic East → West deployment shift with a 30% gap. Entries are mean (SD) over 100 replications. All error metrics are reported in units of $10^4$.

| Method | MAE | RMSE | $p_{98}$ loss | $\mathrm{CVaR}_{2\%}$ |
|---|---|---|---|---|
| LV (ours) | **10.7** (0.72) | **12.2** (0.77) | **21.9** (1.23) | **23.9** (1.10) |
| CVaR | 13.0 (0.00) | 14.4 (0.00) | 24.7 (0.00) | 27.2 (0.00) |
| Wass | 12.3 (0.02) | 13.7 (0.02) | 24.3 (0.04) | 26.5 (0.03) |
| Ridge | 13.2 (0.01) | 14.6 (0.01) | 25.2 (0.01) | 27.7 (0.01) |
| ERM | 12.7 (0.00) | 14.1 (0.00) | 24.7 (0.00) | 27.7 (0.00) |

*Table 4.* Runtime (seconds) for the DRO methods on the East → West deployment shift. Likelihood time is the additional likelihood-fitting, bulk-calibration and rejection-sampling time, which is non-zero only for LV in our implementation. Total time is the end-to-end time recorded per replication.

| Method | CV | Likelihood | Solve | Total |
|---|---|---|---|---|
| LV | **1.30** (0.02) | **0.01** (0.00) | **0.13** (0.00) | **1.44** (0.02) |
| CVaR | 5.14 (0.09) | – | 0.34 (0.01) | 5.47 (0.09) |
| Wass | 5.80 (0.27) | – | 0.36 (0.02) | 6.16 (0.29) |

Table 3 shows that LV performs best on all four test metrics. Compared to the strongest baseline on this split (Wasserstein), LV reduces MAE by 13% and reduces both $p_{98}$ and $\mathrm{CVaR}_{2\%}$ losses by about 10%. Table 4 shows that LV is also the fastest DRO method in this setting. Additional baselines, parameter sweep trajectories and diagnostic oracle selection in Appendix G show LV maintains favourable MAE–$\mathrm{CVaR}_{2\%}$ trade-offs across hyperparameter settings and alternative split gap sizes (Figure 12 and Table 9).

### 5.3. Empirical Centre: CivilComments Classification

We study the CivilComments binary classification task from WILDS (Koh et al., 2021), which targets robustness under subpopulation-shift. The metrics are mean accuracy and worst-group accuracy on the test data. Worst-group accuracy is the minimum accuracy over 16 identity×label slices.

We represent each comment using a fixed hashed $n$-gram feature map and train a linear classifier head. We avoid fine-tuning a pretrained transformer in this experiment to focus on the goal of comparing *training objectives* under a common and tractable representation, rather than to study representation learning or to match state-of-the-art performance. Holding the feature map fixed makes the comparison more controlled and enables repeated runs and hyperparameter sweeps. All methods share the same input representation

and head family, and differ only in the DRO-induced objective and its stochastic approximation.

Baselines are ERM, GroupDRO (Sagawa et al., 2020), CVaR, and $\chi^2$-DRO (Namkoong & Duchi, 2016). We propose an LV extension of GroupDRO, which we call LV-Group. In this experiment, a data point is $\xi = (x, y, g)$, where $x$ is the feature vector, $y \in \{0, 1\}$ is the label, and $g$ is the group indicator. Here, we write the classifier parameters as $\theta$ to avoid notation overlap. Let $f_\theta(\xi)$ be the per-example cross-entropy loss, and let $\hat{\mathbb{P}}_g$ be the empirical distribution of group $g$. GroupDRO considers the worst-case objective over uncertain group mixture weights,

$$\min_\theta \sup_{\mathbb{P} \in \mathcal{C}_G} \mathbb{E}_\mathbb{P}[f_\theta(\xi)], \ \mathcal{C}_G = \left\{ \sum_{g=1}^G q_g \hat{\mathbb{P}}_g : q \in \Delta_G \right\},$$

where $\Delta_G$ is the $(G-1)$-dimensional probability simplex. This ambiguity set is induced by all mixtures of the empirical group conditionals. Similar forms of $\mathcal{C}_G$ appear in IP literature as the *credal model averaging* credal set, used to obtain robust model-averaged predictions under prior model uncertainty (Corani & Mignatti, 2013).

LV-Group restricts the centres $\mathbb{P} \in \mathcal{C}_G$ to a label-free bulk set $\Xi_0$ (see Appendix H) and adds an $\varepsilon$-contamination layer,

$$\mathcal{C}_{\mathrm{LVG}}(\varepsilon) = \left\{ (1-\varepsilon)\mathbb{P}_{\Xi_0} + \varepsilon R : \mathbb{P} \in \mathcal{C}_G, \ R \in \mathcal{P}(\Xi_0) \right\},$$

where $\mathbb{P}_{\Xi_0}$ denotes $\mathbb{P}$ restricted to $\Xi_0$ and renormalised. This matches the *discounting* construction in IP (Shafer, 1976; Walley, 1991; Moral, 2018): we downweight the structured credal set $\mathcal{C}_G$ and allocate the remaining mass to a vacuous model. Discounting formalises *source unreliability* by weakening the information encoded in a credal set $K$: an unreliability level $\varepsilon \in [0, 1]$ yields a larger set $D(K, \varepsilon)$ with $D(K, 0) = K$, monotone growth in $\varepsilon$, and a vacuous limit at $\varepsilon = 1$. A principled view is to pick a divergence $D_i$ and set $D(K, \varepsilon) = \{q : \exists p \in K, \ D_i(p, q) \le \varepsilon\}$.

The resulting robust risk decomposes as

$$\sup_{Q \in \mathcal{C}_{\mathrm{LVG}}(\varepsilon)} \mathbb{E}_Q[f_\theta] = (1-\varepsilon) \sup_{\mathbb{P} \in \mathcal{C}_G} \mathbb{E}_{\mathbb{P}_{\Xi_0}}[f_\theta] + \varepsilon \sup_{\xi \in \Xi_0} f_\theta,$$

where $f_\theta$ is short for $f_\theta(\xi)$. The first term targets shifts in group proportions, while the contamination term targets contaminations that the grouping does not capture. In addition to adding vacuous distortions to fixed probabilities, LV-Group is an example of adding an IP discounting layer on top of other base DRO ambiguity sets, targeting the base robustness and OOS contamination simultaneously.

In our implementation, the restriction to $\Xi_0$ is approximated by filtering the official training data to $\{\xi \in \Xi_0\}$ and running the objective on minibatches drawn from this bulk-filtered set. The sup term is approximated via a smoothmax

*Table 5.* CivilComments test performance under minimax validation selection. We report mean accuracy and worst-group accuracy (min over 16 identity×label slices). Accuracy values are mean (SD) over 20 replications. GroupDRO-B uses the GroupDRO objective trained on the bulk-filtered data. Gap = mean accuracy − worst-group accuracy; smaller values indicate fairer performance across groups. Acc. is short for accuracy.

| Method | Mean acc. | Worst-group acc. | Gap |
|---|---|---|---|
| LV-Group (ours) | 0.828 (0.005) | **0.516** (0.010) | **0.312** |
| GroupDRO | 0.770 (0.002) | 0.456 (0.005) | 0.314 |
| GroupDRO-B | 0.768 (0.003) | 0.453 (0.007) | 0.315 |
| ERM | **0.891** (0.000) | 0.022 (0.001) | 0.869 |
| CVaR | 0.835 (0.011) | 0.230 (0.051) | 0.605 |
| $\chi^2$-DRO | 0.798 (0.021) | 0.247 (0.058) | 0.551 |

for tractability. As an ablation, GroupDRO-B applies the GroupDRO objective to the same bulk-filtered training set.

Table 5 summarises the results. LV-Group attains the highest worst-group accuracy and improves mean accuracy over GroupDRO. ERM has the highest mean accuracy but fails on the worst-group metric: compared to it, LV-Group only has a 0.063 lower mean accuracy but improves the worst-group accuracy by 0.494. See Appendix H for full details and extra ablations.

## 6. Discussion

We propose credal ambiguity sets by linking the IP notion of upper expectation with the DRO worst-case risk. Specifically, we address the ill-posedness of Huber contamination in distributional uncertainty sets. In unbounded spaces with unbounded losses, any DRO objective where the ambiguity set includes the forward Huber contamination becomes infinite unless the adversary is constrained. We propose focusing on a data-calibrated bulk set $\Xi_0$ and allowing an $\varepsilon$-fraction of adversarial mass within $\Xi_0$. This results in a robust objective with a closed-form expression of mean + sup, remaining computationally tractable for standard losses and bulk geometries. Additionally, we provide a high-probability certificate that distinguishes in-bulk robustness from tail behaviour. Experiments demonstrate a favourable robustness–accuracy trade-off and reduced runtimes in convex optimisation across synthetic and real-world data, with flexible centre choices such as Bayesian predictive, frequentist plug-in, or empirical methods. This demonstrates that the IP–DRO relationship we exploit can be both practical and computationally advantageous compared to traditional divergence-ball DRO.

We also introduce a new perspective on how common contamination neighbourhoods relate. The forward LV ball corresponds to adding adversarial mass, the reverse LV ball to trimming low-loss points, and the TV ball combines both effects. Understanding this connection allows us to interpret

robustness budgets and align the ambiguity model with the type of shift we aim to defend against.

Our California housing experiment illustrates how the geometry of the bulk set can encode conditional distributional shifts. Generally, this is important in problems with asymmetric or heterogeneous dimensions, where an isotropic notion of robustness may misalign with plausible shifts.

There are clear limitations. Our bulk calibration relies on a split-sample DKW selection step. To certify $\gamma$ and $\delta$ around 0.05, the sample size needs to be roughly of order $10^3$, which can be restrictive in small-data scenarios (see Figure 6 in Appendix A). Developing more data-efficient calibration methods for $\Xi_0$ would enhance the framework's applicability in settings where large selection sets are infeasible.

The framework opens several promising avenues for future research. A systematic approach to discounting in ambiguity sets could unify handling of both target drift and contamination, while preserving the upper-expectation interpretation. A natural extension is a discounted version of reverse LV balls, enabling separate modelling of training and deployment contamination, possibly with different ratios. This would produce a generalised TV-type model where the mixture and CVaR tail parameters are not required to be equal, broadening the application scope. In model-based DRO, discounting existing divergence-based ambiguity sets could concurrently address model uncertainty, misspecification, and OOS contamination. Another extension is online bulk set calibration: with new deployment data, we could refit or update the score model and the centre distribution, recompute scores, and rerun the same DKW bulk calibration procedure, while keeping the downstream optimisation objective unchanged. This could be linked to recent work on online conformal prediction methods (Gibbs & Candès, 2024; Gauthier et al., 2025; Hultberg et al., 2026) to establish formal long-run or anytime guarantees.

More broadly, a unified theory relating the geometry of bulk sets, the choice of centre $\mathbb{P}_c$, and loss function $f_x$ could offer principled approaches tailored to robustness objectives and computational constraints. Finally, imprecise probability encompasses many credal sets beyond linear vacuity, and characterising their upper expectations in forms conducive to efficient DRO objectives is a promising direction to expand the design space of robust decision rules.

## Acknowledgements

The authors would like to thank Dr Charita Dellaporta and Jake Hobson for their valuable insights and useful feedback on an earlier version of the draft. We would also like to thank the anonymous reviewers and the Area Chair whose comments greatly improved the paper. Mengqi Chen is

supported by the Warwick Statistics Centre for Doctoral Training and acknowledges funding from the University of Warwick. Thomas B. Berrett was supported by European Research Council Starting Grant 101163546. Theodoros Damoulas acknowledges support from a UKRI Turing AI acceleration Fellowship [EP/V02678X/1]. For the purpose of open access, the authors have applied a Creative Commons Attribution (CC-BY) license to any Author Accepted Manuscript version arising from this submission.

## Impact Statement

This paper presents work whose goal is to advance the field of machine learning, specifically by making contamination-based distributionally robust optimisation well-posed and computationally tractable in continuous, unbounded settings. Distributionally robust methods can improve reliability under distribution shift, which may reduce the risk of rare but severe outlier events in downstream decisions. However, they can also be misused to justify overly conservative policies or to obscure model misspecification if tolerance levels are chosen without validation or domain expertise. The trade-off between robustness to OOS shift and in-sample performance is a fundamental challenge for decision makers.

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

# Supplementary Material for:
# Bulk-Calibrated Credal Ambiguity Sets:
# Fast, Tractable Decision Making under Out-of-Sample Contamination

The Supplementary Material is organised as follows: Appendix A provides additional examples, plots, and pseudocode for the DKW bulk-set calibration; Appendix B collects full proofs of the results stated in the paper; Appendix C presents the blockwise intersection-set extension of the bulk construction and its coverage guarantee; Appendix D gives additional discussion of tolerance selection for $\varepsilon$ and related diagnostics; Appendix E discusses the limitations of existing treatments of the worst-case risk in contamination-related DRO; Appendix F, Appendix G, and Appendix H report additional details and results for the synthetic newsvendor, California housing regression, and CivilComments classification experiments.

## A. Bulk Set Calibration: Examples and Algorithm Details

Figures 4–5 illustrate DKW selection in two-dimensional settings. We demonstrate two shapes: an ellipsoid and a box. Algorithm 1 provides the pseudocode for DKW bulk-set calibration. Figure 6 visualises the sample size requirements in DKW selection: for a fixed confidence level $1 - \delta$, the smallest certifiable tail budget $\gamma$ decreases only at the $m^{-1/2}$ rate, so achieving tight certificates typically requires a non-trivial held-out selection set.

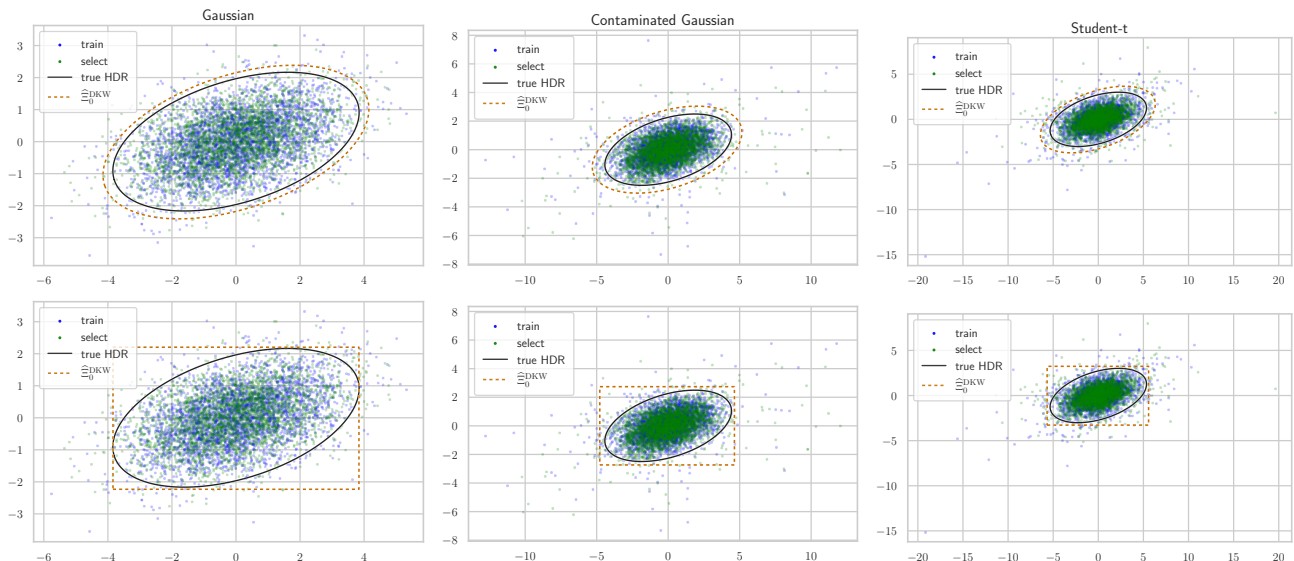

*Figure 4.* Bulk sets for three two-dimensional data-generating processes (left to right): Gaussian, contaminated Gaussian, and Student-$t$. For each process we show (top to bottom): (i) the certified ellipsoidal bulk set $\widehat{\Xi}_0$ at mass $1 - \gamma$ and guarantee $1 - \delta$; and (ii) the certified axis-aligned box bulk set. All plots correspond to $\gamma = 0.05$, $\delta = 0.05$, and $n = 6000$ with $|\mathcal{D}_{\text{fit}}| = |\mathcal{D}_{\text{select}}| = 3000$. The black ellipsoid marks the $1 - \gamma$ highest density region (HDR) of each distribution.

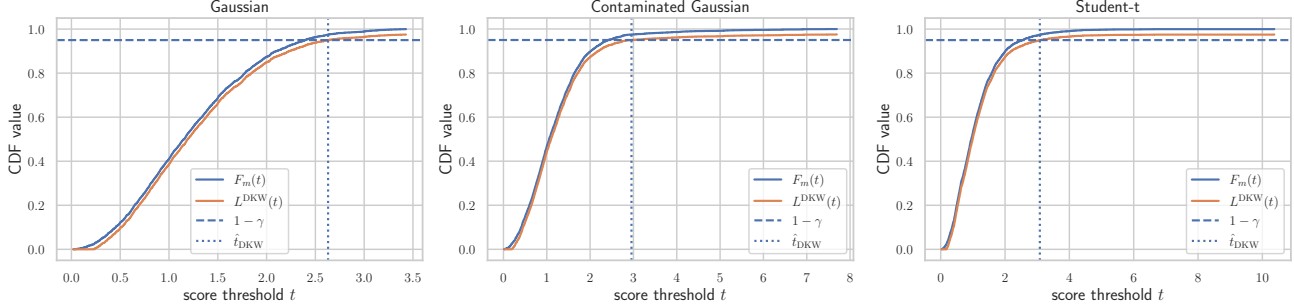

*Figure 5.* Empirical CDF and DKW lower envelope for the ellipsoid bulk sets displayed in Figure 4 under three data-generating processes: Gaussian (left), contaminated Gaussian (middle), and Student-$t$ (right).

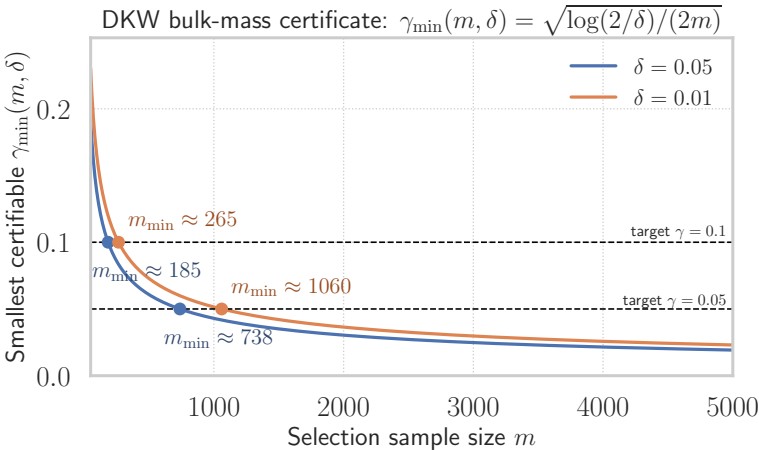

*Figure 6.* Relationship between confidence and selection sample size in DKW bulk-set calibration. For a selection sample size $m$ and confidence $1 - \delta$, the bulk-mass target $1 - \gamma$ is certifiable only when $\gamma \geq r_{m,\delta} := \sqrt{\log(2/\delta)/(2m)}$. Markers indicate the minimum $m$ required to certify $\gamma \in \{0.05, 0.10\}$ at $\delta \in \{0.05, 0.01\}$.

## B. Proofs

### B.1. Proof of Theorem 2.1: Worst-case Risk of the Support-restricted LV Set

*Proof.* Assume $\varepsilon > 0$, otherwise $\mathcal{A}^{\mathrm{LV}}_{\varepsilon,\Xi_0}(\mathbb{P}_{c,\Xi_0}) = \{\mathbb{P}_{c,\Xi_0}\}$ and the result is immediate. Let $M = \sup_{\xi \in \Xi_0} f_x(\xi) \in [-\infty, \infty]$ and write $Q = (1 - \varepsilon)\mathbb{P}_{c,\Xi_0} + \varepsilon R$ with $\mathrm{supp}(R) \subseteq \Xi_0$. Then

$$\int f_x \mathrm{d}Q = (1 - \varepsilon)\int f_x \mathrm{d}\mathbb{P}_{c,\Xi_0} + \varepsilon \int f_x \mathrm{d}R \leq (1 - \varepsilon)\mathbb{E}_{\xi \sim \mathbb{P}_{c,\Xi_0}}[f_x] + \varepsilon M,$$

since $Q$ assigns all its mass to $\Xi_0$. Taking the supremum over $Q \in \mathcal{A}^{\mathrm{LV}}_{\varepsilon,\Xi_0}(\mathbb{P}_{c,\Xi_0})$ yields

$$\sup_{Q \in \mathcal{A}^{\mathrm{LV}}_{\varepsilon,\Xi_0}(\mathbb{P}_{c,\Xi_0})} \int f_x \mathrm{d}Q \leq (1 - \varepsilon)\mathbb{E}_{\xi \sim \mathbb{P}_{c,\Xi_0}}[f_x] + \varepsilon M.$$

For the reverse inequality, if $M = +\infty$ choose $\xi_n \in \Xi_0$ with $f_x(\xi_n) \uparrow +\infty$ and set $R_n = \delta_{\xi_n}$, which obeys $\mathrm{supp}(R_n) \subseteq \Xi_0$. Then

$$\int f_x \mathrm{d}\big((1 - \varepsilon)\mathbb{P}_{c,\Xi_0} + \varepsilon R_n\big) = (1 - \varepsilon)\mathbb{E}_{\xi \sim \mathbb{P}_{c,\Xi_0}}[f_x(\xi)] + \varepsilon f_x(\xi_n) \xrightarrow[n \to \infty]{} +\infty = (1 - \varepsilon)\mathbb{E}_{\xi \sim \mathbb{P}_{c,\Xi_0}}[f_x(\xi)] + \varepsilon M,$$

so the supremum is $+\infty$ and the equality holds.

If $M < \infty$, fix $\eta > 0$ and choose $\xi_\eta \in \Xi_0$ with $f_x(\xi_\eta) > M - \eta$ (by definition of $M$). Let $R_\eta = \delta_{\xi_\eta}$, again admissible. Then

$$\int f_x \mathrm{d}\big((1 - \varepsilon)\mathbb{P}_{c,\Xi_0} + \varepsilon R_\eta\big) = (1 - \varepsilon)\mathbb{E}_{\xi \sim \mathbb{P}_{c,\Xi_0}}[f_x(\xi)] + \varepsilon f_x(\xi_\eta) > (1 - \varepsilon)\mathbb{E}_{\xi \sim \mathbb{P}_{c,\Xi_0}}[f_x] + \varepsilon(M - \eta).$$

Taking the supremum over admissible $R$ and letting $\eta \downarrow 0$ gives

$$\sup_{Q \in \mathcal{A}^{\mathrm{LV}}_{\varepsilon,\Xi_0}(\mathbb{P}_{c,\Xi_0})} \int f_x \mathrm{d}Q \geq (1 - \varepsilon)\mathbb{E}_{\xi \sim \mathbb{P}_{c,\Xi_0}}[f_x] + \varepsilon M.$$

Combine with the upper bound to conclude equality. $\square$

### B.2. Proof of Corollary 2.2: Worst-case Distribution

*Proof.* Fix $x \in \mathcal{X}$. For any $Q \in \mathcal{A}^{\mathrm{LV}}_{\varepsilon,\Xi_0}(\mathbb{P}_{c,\Xi_0})$ we can write $Q = (1 - \varepsilon)\mathbb{P}_{c,\Xi_0} + \varepsilon R$ for some $R \in \mathcal{P}(\Xi_0)$, then

$$\mathbb{E}_{\xi \sim Q}[f_x(\xi)] = (1 - \varepsilon)\mathbb{E}_{\xi \sim \mathbb{P}_{c,\Xi_0}}[f_x(\xi)] + \varepsilon \mathbb{E}_{\xi \sim R}[f_x(\xi)]. \tag{6}$$

---

**Algorithm 1** DKW Bulk-Set Score Selection

---

**Input:** score-fitting data $\mathcal{D}_{\text{fit}}$; DKW selection data $\mathcal{D}_{\text{select}}$; target $\gamma \in (0, 1)$ and $\delta \in (0, 1)$.
**Step 1: Fix nested shape family.**
Fit a score $s(\cdot)$ on $\mathcal{D}_{\text{fit}}$; define $\Xi_0(t) = \{\xi : s(\xi) \le t\}$.
**Step 2: DKW selection on held-out selection scores.**
For each $\tilde{\xi}_j \in \mathcal{D}_{\text{select}}$, set $z_j \leftarrow s(\tilde{\xi}_j)$.
Let $m \leftarrow |\mathcal{D}_{\text{select}}|$.
$r \leftarrow \sqrt{\log(2/\delta)/(2m)}$.
**if** $\gamma < r$ **then**
    **return** "no certificate; smallest certifiable $\gamma = r$" (increase $m$, or relax $\gamma$ and $\delta$).
**else**
    Sort $z_{1:m}$ to obtain $z_{(1)} \le \cdots \le z_{(m)}$.
    $j^\star \leftarrow \lceil m(1 - \gamma + r) \rceil$.
    $\hat{t} \leftarrow z_{(j^\star)}$.
    $\widehat{\Xi}_0 \leftarrow \Xi_0(\hat{t})$.
    **return** $(\hat{t}, \widehat{\Xi}_0)$ and certificate
        $\Pr\{\mathbb{P}^\star(\Xi_0(\hat{t})) \ge 1 - \gamma\} \ge 1 - \delta$.
**end if**
*Complexity:* $O(m \log m)$ time (one sort); $O(m)$ space.

---

Since $\operatorname{supp}(R) \subseteq \Xi_0$, we have $f_x(\xi) \le M_x$ for all $\xi$ in the support of $R$, and thus $\mathbb{E}_{\xi \sim R}[f_x(\xi)] \le M_x$. Substituting into (6) yields

$$\mathbb{E}_{\xi \sim Q}[f_x(\xi)] \le (1 - \varepsilon)\mathbb{E}_{\xi \sim \mathbb{P}_{c,\Xi_0}}[f_x(\xi)] + \varepsilon M_x.$$

By Theorem 2.1, the right-hand side equals $\mathcal{R}_{\mathcal{A}^{\text{LV}}_{\varepsilon,\Xi_0}(\mathbb{P}_{c,\Xi_0})}(f_x)$, so this upper bound is sharp.

Assume there exists $R^\star \in \mathcal{P}(\Xi_0)$ with $R^\star(\Xi_0^{\max}(x)) = 1$. Then $f_x(\xi) = M_x$ holds $R^\star$-almost surely, so $\mathbb{E}_{\xi \sim R^\star}[f_x(\xi)] = M_x$. Plugging $R^\star$ into (6) gives

$$\mathbb{E}_{\xi \sim Q^\star}[f_x(\xi)] = (1 - \varepsilon)\mathbb{E}_{\xi \sim \mathbb{P}_{c,\Xi_0}}[f_x(\xi)] + \varepsilon M_x = \mathcal{R}_{\mathcal{A}^{\text{LV}}_{\varepsilon,\Xi_0}(\mathbb{P}_{c,\Xi_0})}(f_x),$$

which proves $Q^\star$ is worst-case.

Assume $M_x < \infty$ and $\Xi_0^{\max}(x) = \emptyset$. Then $f_x(\xi) < M_x$ for all $\xi \in \Xi_0$, so the function $g(\xi) := M_x - f_x(\xi)$ is strictly positive on $\Xi_0$. For any $R \in \mathcal{P}(\Xi_0)$, if $\mathbb{E}_{\xi \sim R}[f_x(\xi)] = M_x$ then

$$0 = \mathbb{E}_{\xi \sim R}[M_x - f_x(\xi)] = \mathbb{E}_{\xi \sim R}[g(\xi)].$$

Since $g(\xi) \ge 0$ for all $\xi$ and $g(\xi) > 0$ on $\Xi_0$, the equality $\mathbb{E}_{\xi \sim R}[g] = 0$ implies $g(\xi) = 0$ $R$-almost surely, which is impossible because $\Xi_0^{\max}(x) = \emptyset$. Therefore $\mathbb{E}_{\xi \sim R}[f_x(\xi)] < M_x$ for every $R \in \mathcal{P}(\Xi_0)$, and by (6), $\mathbb{E}_{\xi \sim Q}[f_x(\xi)] < (1 - \varepsilon)\mathbb{E}_{\xi \sim \mathbb{P}_{c,\Xi_0}}[f_x(\xi)] + \varepsilon M_x$ for every admissible $Q$. This shows non-attainment. $\square$

### B.3. Proof of Proposition 2.3: LV Distortion Classification of the Contamination Set

*Proof.* Recall the definitions:

$$\mathcal{A}^{\text{LV}}_{\varepsilon,\Xi_0}(\mathbb{P}_{c,\Xi_0}) = \{Q : Q = (1 - \varepsilon)\mathbb{P}_{c,\Xi_0} + \varepsilon R, R \in \mathcal{P}(\Xi_0)\}$$
$$\mathcal{B}^{\varepsilon}_{\text{LV}}(\mathbb{P}_{c,\Xi_0}) = \{Q \in \mathcal{P}(\Xi_0) : \text{LV}(Q, \mathbb{P}_{c,\Xi_0}) \le \varepsilon\},$$

and

$$\text{LV}(Q, \mathbb{P}_{c,\Xi_0}) = \sup_{A : \mathbb{P}_{c,\Xi_0}(A) > 0} \frac{\mathbb{P}_{c,\Xi_0}(A) - Q(A)}{\mathbb{P}_{c,\Xi_0}(A)}.$$

We introduce the credal set defined by eventwise domination:

$$\mathcal{M}(\underline{\mathbb{P}_{c,\Xi_0}}) := \{Q \in \mathcal{P}(\Xi_0) : Q(A) \ge (1 - \varepsilon)\mathbb{P}_{c,\Xi_0}(A), \forall A \in \mathcal{B}(\Xi_0)\}. \tag{7}$$

We will show that $\mathcal{A}^{\mathrm{LV}}_{\varepsilon,\Xi_0}(\mathbb{P}_{c,\Xi_0}) = \mathcal{M}(\underline{\mathbb{P}_{c,\Xi_0}}) = \mathcal{B}^{\varepsilon}_{\mathrm{LV}}(\mathbb{P}_{c,\Xi_0})$.

**First equality:** $\mathcal{A}^{\mathrm{LV}}_{\varepsilon,\Xi_0}(\mathbb{P}_{c,\Xi_0}) = \mathcal{M}(\underline{\mathbb{P}_{c,\Xi_0}})$.

To show that $\mathcal{A}^{\mathrm{LV}}_{\varepsilon,\Xi_0}(\mathbb{P}_{c,\Xi_0}) \subseteq \mathcal{M}(\underline{\mathbb{P}_{c,\Xi_0}})$, we choose any $Q \in \mathcal{A}^{\mathrm{LV}}_{\varepsilon,\Xi_0}(\mathbb{P}_{c,\Xi_0})$. Then $Q = (1-\varepsilon)\mathbb{P}_{c,\Xi_0} + \varepsilon R$ for some $R \in \mathcal{P}(\Xi_0)$. For any $A \in \mathcal{B}(\Xi_0)$, $\overline{Q(A)} = (1-\varepsilon)\mathbb{P}_{c,\Xi_0}(A) + \varepsilon R(A) \geq (1-\varepsilon)\mathbb{P}_{c,\Xi_0}(A)$ since $R(A) \geq 0$. Hence $Q$ satisfies the eventwise lower bound for all $A \in \mathcal{B}(\Xi_0)$.

To show the converse, let $Q \in \mathcal{P}(\Xi_0)$ satisfy $Q(A) \geq (1-\varepsilon)\mathbb{P}_{c,\Xi_0}(A)$ for all $A \in \mathcal{B}(\Xi_0)$. Define the set function $S : \mathcal{B}(\Xi_0) \to [0,\infty)$ by

$$S(A) := Q(A) - (1-\varepsilon)\mathbb{P}_{c,\Xi_0}(A).$$

Then $S(A) \geq 0$ for all $A$ by assumption. Moreover, since both $Q$ and $\mathbb{P}_{c,\Xi_0}$ are countably additive, $S$ is countably additive as a difference of two measures. Its total mass is

$$S(\Xi_0) = Q(\Xi_0) - (1-\varepsilon)\mathbb{P}_{c,\Xi_0}(\Xi_0) = 1 - (1-\varepsilon) = \varepsilon.$$

If $\varepsilon > 0$, define $R := S/\varepsilon$. Then $R$ is a nonnegative and countably additive probability measure on $(\Xi_0, \mathcal{B}(\Xi_0))$, and by construction

$$Q(A) = (1-\varepsilon)\mathbb{P}_{c,\Xi_0}(A) + \varepsilon R(A) \qquad \forall A \in \mathcal{B}(\Xi_0),$$

i.e. $Q \in \mathcal{A}^{\mathrm{LV}}_{\varepsilon,\Xi_0}(\mathbb{P}_{c,\Xi_0})$. Therefore, $\mathcal{M}(\underline{\mathbb{P}_{c,\Xi_0}}) \subseteq \mathcal{A}^{\mathrm{LV}}_{\varepsilon,\Xi_0}(\mathbb{P}_{c,\Xi_0})$ and we have

$$\mathcal{A}^{\mathrm{LV}}_{\varepsilon,\Xi_0}(\mathbb{P}_{c,\Xi_0}) = \mathcal{M}(\underline{\mathbb{P}_{c,\Xi_0}}).$$

**Second equality:** $\mathcal{M}(\underline{\mathbb{P}_{c,\Xi_0}}) = \mathcal{B}^{\varepsilon}_{\mathrm{LV}}(\mathbb{P}_{c,\Xi_0})$.

For any $Q \in \mathcal{P}(\Xi_0)$, we have the following chain:

$$
\begin{aligned}
& Q(A) \geq (1-\varepsilon)\mathbb{P}_{c,\Xi_0}(A) \quad \text{for all } A \in \mathcal{B}(\Xi_0) \\
\Longleftrightarrow\ & \mathbb{P}_{c,\Xi_0}(A) - Q(A) \leq \varepsilon\mathbb{P}_{c,\Xi_0}(A) \quad \text{for all } A \in \mathcal{B}(\Xi_0) \\
\Longleftrightarrow\ & \mathbb{P}_{c,\Xi_0}(A) - Q(A) \leq \varepsilon\mathbb{P}_{c,\Xi_0}(A) \quad \text{for all } A \in \mathcal{B}(\Xi_0) \text{ with } \mathbb{P}_{c,\Xi_0}(A) > 0 \\
\Longleftrightarrow\ & \frac{\mathbb{P}_{c,\Xi_0}(A) - Q(A)}{\mathbb{P}_{c,\Xi_0}(A)} \leq \varepsilon \quad \text{for all } A \in \mathcal{B}(\Xi_0) \text{ with } \mathbb{P}_{c,\Xi_0}(A) > 0 \\
\Longleftrightarrow\ & \sup_{A:\mathbb{P}_{c,\Xi_0}(A)>0} \frac{\mathbb{P}_{c,\Xi_0}(A) - Q(A)}{\mathbb{P}_{c,\Xi_0}(A)} \leq \varepsilon.
\end{aligned}
$$

The last quantity is the definition of $\mathrm{LV}(Q, \mathbb{P}_{c,\Xi_0})$, which proves

$$\mathcal{M}(\underline{\mathbb{P}_{c,\Xi_0}}) = \mathcal{B}^{\varepsilon}_{\mathrm{LV}}(\mathbb{P}_{c,\Xi_0}).$$

*Edge case.* When $\varepsilon = 0$ the three sets reduce to $\{\mathbb{P}_{c,\Xi_0}\}$: $\mathcal{A}^{\mathrm{LV}}_{\varepsilon,\Xi_0}(\mathbb{P}_{c,\Xi_0}) = \{\mathbb{P}_{c,\Xi_0}\}$ by definition; (7) gives $Q(A) \geq \mathbb{P}_{c,\Xi_0}(A)$ for all $A$ and applying it to complements forces $\mathcal{M}(\underline{\mathbb{P}_{c,\Xi_0}}) = \{\mathbb{P}_{c,\Xi_0}\}$; the LV condition becomes $\mathrm{LV}(Q, \mathbb{P}_{c,\Xi_0}) \leq 0$, which gives $\mathcal{B}^{\varepsilon}_{\mathrm{LV}}(\mathbb{P}_{c,\Xi_0}) = \{\mathbb{P}_{c,\Xi_0}\}$. $\qquad\square$

### B.4. Alternative Proof of Proposition 2.4: Worst-case Risk of a TV Ball

*Alternative proof of Proposition 2.4.* Assume $\mathbb{E}_{\mathbb{P}}[|f|] < \infty$ and let $M := \sup_{\xi \in \Xi} f(\xi)$. If $M = +\infty$ and $\varepsilon > 0$, pick $\xi_n \in \Xi$ such that $f(\xi_n) \uparrow +\infty$ and define $Q_n := (1-\varepsilon)\mathbb{P} + \varepsilon\delta_{\xi_n}$. Then for every $A \in \mathcal{F}$,

$$\mathbb{P}(A) - Q_n(A) = \varepsilon\big(\mathbb{P}(A) - \delta_{\xi_n}(A)\big),$$

so $\mathrm{TV}(\mathbb{P}, Q_n) = \varepsilon\,\mathrm{TV}(\mathbb{P}, \delta_{\xi_n}) \leq \varepsilon$ and thus $Q_n \in \mathcal{B}^{\varepsilon}_{\mathrm{TV}}(\mathbb{P})$. Moreover,

$$\mathbb{E}_{Q_n}[f] = (1-\varepsilon)\mathbb{E}_{\mathbb{P}}[f] + \varepsilon f(\xi_n) \xrightarrow[n\to\infty]{} +\infty.$$

Hence $\mathcal{R}_{\mathcal{B}^{\varepsilon}_{\mathrm{TV}}(\mathbb{P})}(f) = +\infty$. We assume $M < \infty$ for the rest of the proof.

Fix $Q$ with $\mathrm{TV}(\mathbb{P}, Q) \leq \varepsilon$. If $\mathrm{TV}(\mathbb{P}, Q) = 0$ then $Q = \mathbb{P}$ and

$$\mathbb{E}_Q[f] = \mathbb{E}_{\mathbb{P}}[f] \leq \mathrm{CVaR}_{1-\varepsilon}^{\mathbb{P}}(f) \leq (1-\varepsilon)\mathrm{CVaR}_{1-\varepsilon}^{\mathbb{P}}(f) + \varepsilon M,$$

so Proposition 2.4 holds. Thus we may assume $\mathrm{TV}(\mathbb{P}, Q) > 0$.

Let $\alpha := \mathrm{TV}(\mathbb{P}, Q) \in (0, \varepsilon]$. By Alvarez-Esteban et al. (2012, Proposition 2), there exist probability measures $C, R_1, R_2$ such that

$$\mathbb{P} = (1-\alpha)C + \alpha R_1, \qquad Q = (1-\alpha)C + \alpha R_2. \tag{8}$$

In particular, from the first identity in (8) we know that $C$ lies in the *reverse LV ball* of radius $\alpha$ around $\mathbb{P}$: $\mathcal{A}_\alpha^{\leftarrow}(\mathbb{P}) := \{Q \in \mathcal{P}(\Xi) : \exists R \in \mathcal{P}(\Xi) \text{ s.t. } \mathbb{P} = (1-\alpha)Q + \alpha R\}$. Taking expectations of $f$ under the second identity in (8) yields

$$\begin{aligned}
\mathbb{E}_Q[f] &= (1-\alpha)\mathbb{E}_C[f] + \alpha\mathbb{E}_{R_2}[f] \\
&\leq (1-\alpha)\mathbb{E}_C[f] + \alpha M \\
&\leq (1-\alpha)\sup_{C \in \mathcal{A}_\alpha^{\leftarrow}(\mathbb{P})} \mathbb{E}_C[f] + \alpha M \\
&= (1-\alpha)\mathrm{CVaR}_{1-\alpha}^{\mathbb{P}}(f) + \alpha M. \quad \text{by Föllmer \& Schied (2016, Theorem 4.52)}
\end{aligned} \tag{9}$$

Since $\alpha \leq \varepsilon$, we have the set inclusion $\mathcal{A}_\alpha^{\leftarrow}(\mathbb{P}) \subseteq \mathcal{A}_\varepsilon^{\leftarrow}(\mathbb{P})$. Indeed, if $\mathbb{P} = (1-\alpha)C + \alpha R$ for some probability measures $(C, R)$, then

$$\mathbb{P} = (1-\varepsilon)C + \varepsilon R', \qquad R' := \frac{\varepsilon - \alpha}{\varepsilon}C + \frac{\alpha}{\varepsilon}R \in \mathcal{P}(\Xi).$$

Therefore,

$$\sup_{C \in \mathcal{A}_\alpha^{\leftarrow}(\mathbb{P})} \mathbb{E}_C[f] \leq \sup_{C \in \mathcal{A}_\varepsilon^{\leftarrow}(\mathbb{P})} \mathbb{E}_C[f] = \mathrm{CVaR}_{1-\varepsilon}^{\mathbb{P}}(f).$$

Substituting this bound into (9) gives

$$\mathbb{E}_Q[f] \leq (1-\alpha)\mathrm{CVaR}_{1-\varepsilon}^{\mathbb{P}}(f) + \alpha M \leq (1-\varepsilon)\mathrm{CVaR}_{1-\varepsilon}^{\mathbb{P}}(f) + \varepsilon M,$$

where the last inequality uses $\alpha \leq \varepsilon$ and $M \geq \mathrm{CVaR}_{1-\varepsilon}^{\mathbb{P}}(f)$. We now show that the right-hand side is tight. By the representation

$$\mathrm{CVaR}_{1-\varepsilon}^{\mathbb{P}}(f) = \sup_{C \in \mathcal{A}_\varepsilon^{\leftarrow}(\mathbb{P})} \mathbb{E}_C[f],$$

there exists a sequence of probability measures $(C_n)_{n \geq 1} \subseteq \mathcal{A}_\varepsilon^{\leftarrow}(\mathbb{P})$ such that

$$\lim_{n \to \infty} \mathbb{E}_{C_n}[f] = \sup_{C \in \mathcal{A}_\varepsilon^{\leftarrow}(\mathbb{P})} \mathbb{E}_C[f] = \mathrm{CVaR}_{1-\varepsilon}^{\mathbb{P}}(f).$$

For each $n$, the definition of $\mathcal{A}_\varepsilon^{\leftarrow}(\mathbb{P})$ guarantees the existence of some $R_{1,n}$ with

$$\mathbb{P} = (1-\varepsilon)C_n + \varepsilon R_{1,n}.$$

Fix $n$, take $R_{2,n}$ supported on $\{\xi : f(\xi) \geq M - 1/n\}$, which is nonempty by the definition of $\sup$. Define

$$Q_n := (1-\varepsilon)C_n + \varepsilon R_{2,n}.$$

Then, by construction,

$$\mathbb{E}_{Q_n}[f] = (1-\varepsilon)\mathbb{E}_{C_n}[f] + \varepsilon\mathbb{E}_{R_{2,n}}[f] \geq (1-\varepsilon)\mathbb{E}_{C_n}[f] + \varepsilon(M - 1/n).$$

Moreover, using $\mathbb{P} = (1-\varepsilon)C_n + \varepsilon R_{1,n}$ and $Q_n = (1-\varepsilon)C_n + \varepsilon R_{2,n}$, we have for every $A \in \mathcal{F}$

$$\mathbb{P}(A) - Q_n(A) = \varepsilon\big(R_{1,n}(A) - R_{2,n}(A)\big).$$

Taking the supremum over $A$ yields

$$\mathrm{TV}(\mathbb{P}, Q_n) = \varepsilon\,\mathrm{TV}(R_{1,n}, R_{2,n}) \leq \varepsilon,$$

so each $Q_n$ lies in the TV ball $\mathcal{B}_{\mathrm{TV}}^{\varepsilon}(\mathbb{P})$.

Thus,

$$\sup_{Q:\mathrm{TV}(\mathbb{P},Q)\leq\varepsilon} \mathbb{E}_Q[f] \geq \mathbb{E}_{Q_n}[f] \geq (1-\varepsilon)\mathbb{E}_{C_n}[f] + \varepsilon(M - 1/n).$$

Letting $n \to \infty$ and using $\mathbb{E}_{C_n}[f] \to \mathrm{CVaR}_{1-\varepsilon}^{\mathbb{P}}(f)$ yields

$$\sup_{Q:\mathrm{TV}(\mathbb{P},Q)\leq\varepsilon} \mathbb{E}_Q[f] \geq (1-\varepsilon)\mathrm{CVaR}_{1-\varepsilon}^{\mathbb{P}}(f) + \varepsilon M.$$

Combined with the upper bound (9), this proves Proposition 2.4. □

### B.5. Proof of Lemma 3.2: High-probability Bulk-mass Certificate

*Proof.* Because $\mathcal{D}_{\mathrm{select}}$ is independent of $s(\cdot)$ and $\mathcal{D}_{\mathrm{fit}}$, the random variables $Z_j := s(\tilde{\xi}_j)$ are i.i.d. with common CDF $F(t)$ conditional on $s(\cdot)$. All statements proved below conditional on $s(\cdot)$ also hold unconditionally after averaging over $s(\cdot)$.

Existence of $\hat{t}_{\mathrm{DKW}}$: let $Z_{\max} := \max_{1\leq j\leq m} Z_j$. Since $F_m(Z_{\max}) = 1$, we have $L^{\mathrm{DKW}}(Z_{\max}) = 1 - r_{m,\delta}$, so the selection set $\{t : L^{\mathrm{DKW}}(t) \geq 1 - \gamma\}$ is nonempty whenever $\gamma \geq r_{m,\delta}$.

Define the DKW event

$$\mathcal{E}_{\mathrm{DKW}} := \left\{ \sup_{t\in\mathbb{R}} |F_m(t) - F(t)| \leq r_{m,\delta} \right\}.$$

By (4), $\Pr(\mathcal{E}_{\mathrm{DKW}}) \geq 1 - \delta$. On $\mathcal{E}_{\mathrm{DKW}}$ we have, for every $t \in \mathbb{R}$, $F(t) \geq F_m(t) - r_{m,\delta}$. Combining this with $F(t) \geq 0$ gives

$$F(t) \geq [F_m(t) - r_{m,\delta}]_+ = L^{\mathrm{DKW}}(t).$$

Since $F_m$ (and hence $L^{\mathrm{DKW}}$) is right-continuous, the infimum in the definition of $\hat{t}_{\mathrm{DKW}}$ is attained and $L^{\mathrm{DKW}}(\hat{t}_{\mathrm{DKW}}) \geq 1 - \gamma$. Combining this with $F(t) \geq L^{\mathrm{DKW}}(t)$ gives, on $\mathcal{E}_{\mathrm{DKW}}$,

$$F(\hat{t}_{\mathrm{DKW}}) \geq L^{\mathrm{DKW}}(\hat{t}_{\mathrm{DKW}}) \geq 1 - \gamma.$$

Since $F(t) = \mathbb{P}^\star\{\xi : s(\xi) \leq t\} = \mathbb{P}^\star(\Xi_0(t))$, this is equivalent to $\mathbb{P}^\star(\Xi_0(\hat{t}_{\mathrm{DKW}})) \geq 1 - \gamma$. Taking probabilities and using $\Pr(\mathcal{E}_{\mathrm{DKW}}) \geq 1 - \delta$ finishes the proof.

Moreover, with the generalised inverse $F_m^{-1}(u) := \inf\{t : F_m(t) \geq u\}$,

$$L^{\mathrm{DKW}}\big(F_m^{-1}(1 - \gamma + r_{m,\delta})\big) \geq (1 - \gamma + r_{m,\delta}) - r_{m,\delta} = 1 - \gamma,$$

so the empirical quantile $\hat{t}_{\mathrm{emp}} := F_m^{-1}(1 - \gamma + r_{m,\delta})$ is a valid choice; computing $\hat{t}_{\mathrm{emp}}$ reduces to sorting $\{Z_j\}_{j=1}^m$ and is $O(m\log m)$. □

### B.6. Proof of Theorem 3.4: Risk Certificate

Before proving Theorem 3.4, we first state and prove the two following lemmas.

**Lemma B.1** (Composition of in-bulk contaminations). *Let $\Xi_0 \in \mathcal{F}$. Let $C, P, Q \in \mathcal{P}(\Xi)$ and let $R_1, R_2 \in \mathcal{P}(\Xi_0)$. Throughout, we use the convention $\mathcal{P}(\Xi_0) := \{R \in \mathcal{P}(\Xi) : R(\Xi_0) = 1\}$, i.e. probability measures on $\Xi$ supported on $\Xi_0$. Assume that for some $\alpha, \beta \in [0, 1]$,*

$$Q = (1-\alpha)P + \alpha R_1, \qquad P = (1-\beta)C + \beta R_2.$$

*Define $\theta := \alpha + \beta - \alpha\beta = 1 - (1-\alpha)(1-\beta) \in [0, 1]$. If $\theta > 0$, define*

$$R := \frac{(1-\alpha)\beta}{\theta} R_2 + \frac{\alpha}{\theta} R_1 \in \mathcal{P}(\Xi_0).$$

*Then*

$$Q = (1-\theta)C + \theta R.$$

*If $\theta = 0$ (equivalently $\alpha = \beta = 0$), then $Q = C$.*

*Proof.* If $\theta = 0$, then $\alpha = \beta = 0$, hence $Q = P$ and $P = C$, so $Q = C$.

Assume now that $\theta > 0$. First note that $(1 - \alpha)\beta \geq 0$ and $\alpha \geq 0$. Moreover,

$$(1 - \alpha)\beta + \alpha = \beta - \alpha\beta + \alpha = \alpha + \beta - \alpha\beta = \theta.$$

Therefore the coefficients

$$\frac{(1 - \alpha)\beta}{\theta} \geq 0, \qquad \frac{\alpha}{\theta} \geq 0, \qquad \frac{(1 - \alpha)\beta}{\theta} + \frac{\alpha}{\theta} = 1,$$

form a convex combination, so $R$ is a probability measure. Since both $R_1$ and $R_2$ are supported on $\Xi_0$, their convex combination $R$ is also supported on $\Xi_0$, hence $R \in \mathcal{P}(\Xi_0)$.

Finally, using $1 - \theta = (1 - \alpha)(1 - \beta)$, we expand

$$\begin{aligned}
(1 - \theta)C + \theta R &= (1 - \theta)C + (1 - \alpha)\beta R_2 + \alpha R_1 \\
&= (1 - \alpha)(1 - \beta)C + (1 - \alpha)\beta R_2 + \alpha R_1 \\
&= (1 - \alpha)\big((1 - \beta)C + \beta R_2\big) + \alpha R_1 \\
&= (1 - \alpha)P + \alpha R_1 \\
&= Q,
\end{aligned}$$

which proves the claim. $\qquad\square$

**Lemma B.2** (Tail decomposition). *Fix a Borel set $\Xi_0 \subseteq \Xi$ and assume $\mathbb{P}_c(\Xi_0) > 0$. Let $f_x : \Xi \to \mathbb{R}$ be measurable in $\xi$. Given a probability distribution $Q$, let $p > 1$ and $q := p/(p - 1)$, and assume*

$$M_p(x) := \left(\mathbb{E}_{\xi \sim Q}\big[|f_x(\xi)|^p\big]\right)^{1/p} < \infty.$$

*Assume additionally that $f_x(\xi) \geq 0$ for all $\xi \in \Xi$. Suppose with probability at least $1 - \delta$, there exist $\varepsilon \in [0, 1]$ and $\gamma \in [0, 1)$ such that the following holds:*

1. *$Q(\Xi_0) \geq 1 - \gamma$;*

2. *$Q_{\Xi_0} \in \mathcal{A}^{\mathrm{LV}}_{\varepsilon, \Xi_0}(\mathbb{P}_{c, \Xi_0})$.*

*Then, with probability at least $1 - \delta$,*

$$\mathbb{E}_{\xi \sim Q}[f_x(\xi)] \leq (1 - \varepsilon)\mathbb{E}_{\xi \sim \mathbb{P}_{c, \Xi_0}}[f_x] + \varepsilon \sup_{\xi \in \Xi_0} f_x(\xi) + M_p(x)\gamma^{1/q}, \tag{10}$$

*Proof.* Let $\mathcal{E}$ denote the event on which the conditions hold; by assumption, $\Pr(\mathcal{E}) \geq 1 - \delta$. Throughout, we work on $\mathcal{E}$. Set $\gamma^\star := Q(\Xi_0^c) \leq \gamma$. Decompose the $Q$-expectation into the contribution from $\Xi_0$ and its complement:

$$\mathbb{E}_{\xi \sim Q}[f_x] = \int_{\Xi_0} f_x \mathrm{d}Q + \int_{\Xi_0^c} f_x \mathrm{d}Q = (1 - \gamma^\star)\mathbb{E}_{\xi \sim Q_{\Xi_0}}[f_x] + \mathbb{E}_{\xi \sim Q}\big[f_x \mathbf{1}_{\Xi_0^c}\big], \tag{11}$$

where the equality $(1 - \gamma^\star)\mathbb{E}_{\xi \sim Q_{\Xi_0}}[f_x] = \int_{\Xi_0} f_x \mathrm{d}Q$ follows from the definition of the normalised restriction $Q_{\Xi_0}$.

Since $Q_{\Xi_0} \in \mathcal{A}^{\mathrm{LV}}_{\varepsilon, \Xi_0}(\mathbb{P}_{c, \Xi_0})$, there exists some $R \in \mathcal{P}(\Xi_0)$ such that $Q_{\Xi_0} = (1 - \varepsilon)\mathbb{P}_{c, \Xi_0} + \varepsilon R$. Hence, using linearity and that $R$ is supported on $\Xi_0$,

$$\mathbb{E}_{\xi \sim Q_{\Xi_0}}[f_x] = (1 - \varepsilon)\mathbb{E}_{\xi \sim \mathbb{P}_{c, \Xi_0}}[f_x] + \varepsilon\mathbb{E}_{\xi \sim R}[f_x] \leq (1 - \varepsilon)\mathbb{E}_{\xi \sim \mathbb{P}_{c, \Xi_0}}[f_x] + \varepsilon \sup_{\xi \in \Xi_0} f_x(\xi).$$

Multiplying by $(1 - \gamma^\star)$ and using (11) gives

$$\int_{\Xi_0} f_x \mathrm{d}Q \leq (1 - \gamma^\star)\Big[(1 - \varepsilon)\mathbb{E}_{\xi \sim \mathbb{P}_{c, \Xi_0}}[f_x] + \varepsilon \sup_{\xi \in \Xi_0} f_x(\xi)\Big]. \tag{12}$$

By Hölder's inequality with conjugate exponents $(p, q)$,

$$\int_{\Xi_0^c} f_x \mathrm{d}Q = \mathbb{E}_{\xi \sim Q}\big[f_x \mathbf{1}_{\Xi_0^c}\big] \le \big(\mathbb{E}_{\xi \sim Q}[|f_x|^p]\big)^{1/p} \big(\mathbb{E}_{\xi \sim Q}[\mathbf{1}_{\Xi_0^c}^q]\big)^{1/q} = M_p(x)(\gamma^\star)^{1/q}.$$

Using $\gamma^\star \le \gamma$ yields

$$\int_{\Xi_0^c} f_x \mathrm{d}Q \ \le \ M_p(x)\gamma^{1/q}. \tag{13}$$

Adding (12) and (13) and dropping the factor $(1 - \gamma^\star) \le 1$ yields

$$\mathbb{E}_{\xi \sim Q}[f_x] \le (1 - \varepsilon)\mathbb{E}_{\xi \sim \mathbb{P}_{c,\Xi_0}}[f_x] + \varepsilon \sup_{\xi \in \Xi_0} f_x(\xi) + M_p(x)\gamma^{1/q},$$

which is exactly (10). Since all steps hold on $\mathcal{E}$ and $\Pr(\mathcal{E}) \ge 1 - \delta$, the bound holds with probability at least $1 - \delta$. □

We are now ready to prove the theorem.

*Proof of Theorem 3.4.* Let

$$\mathcal{E} := \{\mathbb{P}^\star(\Xi_0) \ge 1 - \gamma\}.$$

By assumption, $\Pr(\mathcal{E}) \ge 1 - \delta$. Throughout the proof we work on the event $\mathcal{E}$.

Using $\widetilde{\mathbb{P}} = (1 - \varepsilon^\star)\mathbb{P}^\star + \varepsilon^\star \widetilde{R}$, we have

$$\begin{aligned}
\widetilde{\mathbb{P}}(\Xi_0) &= (1 - \varepsilon^\star)\mathbb{P}^\star(\Xi_0) + \varepsilon^\star \widetilde{R}(\Xi_0) \\
&\ge (1 - \varepsilon^\star)\mathbb{P}^\star(\Xi_0) \ge (1 - \varepsilon^\star)(1 - \gamma),
\end{aligned} \tag{14}$$

so in particular $\widetilde{\mathbb{P}}(\Xi_0) > 0$. Moreover,

$$\begin{aligned}
\widetilde{\mathbb{P}}(\Xi_0^c) &= (1 - \varepsilon^\star)\mathbb{P}^\star(\Xi_0^c) + \varepsilon^\star \widetilde{R}(\Xi_0^c) \\
&= (1 - \varepsilon^\star)\mathbb{P}^\star(\Xi_0^c) + \varepsilon^\star\big(1 - \widetilde{R}(\Xi_0)\big) \\
&\le (1 - \varepsilon^\star)\gamma + \varepsilon^\star\big(1 - \widetilde{R}(\Xi_0)\big).
\end{aligned} \tag{15}$$

Let $r := \widetilde{R}(\Xi_0) \in [0, 1]$ and define

$$\eta := \frac{\varepsilon^\star r}{\widetilde{\mathbb{P}}(\Xi_0)}.$$

We check that $\eta \in [0, 1]$: clearly $\eta \ge 0$, and since $\widetilde{\mathbb{P}}(\Xi_0) = (1 - \varepsilon^\star)\mathbb{P}^\star(\Xi_0) + \varepsilon^\star r \ge \varepsilon^\star r$, we have $\eta \le 1$.

If $r = 0$, then $\widetilde{\mathbb{P}}(A \cap \Xi_0) = (1 - \varepsilon^\star)\mathbb{P}^\star(A \cap \Xi_0)$ for all $A \in \mathcal{F}$, hence $\widetilde{\mathbb{P}}_{\Xi_0}(A) = \mathbb{P}^\star_{\Xi_0}(A)$.

If $r > 0$, define $\widetilde{R}_{\Xi_0}(A) := \widetilde{R}(A \cap \Xi_0)/\widetilde{R}(\Xi_0)$. For all $A \in \mathcal{F}$,

$$\begin{aligned}
\widetilde{\mathbb{P}}_{\Xi_0}(A) &= \frac{\widetilde{\mathbb{P}}(A \cap \Xi_0)}{\widetilde{\mathbb{P}}(\Xi_0)} \\
&= \frac{(1 - \varepsilon^\star)\mathbb{P}^\star(A \cap \Xi_0) + \varepsilon^\star \widetilde{R}(A \cap \Xi_0)}{\widetilde{\mathbb{P}}(\Xi_0)} \\
&= \frac{(1 - \varepsilon^\star)\mathbb{P}^\star_{\Xi_0}(A)\mathbb{P}^\star(\Xi_0) + \varepsilon^\star \widetilde{R}_{\Xi_0}(A)\widetilde{R}(\Xi_0)}{\widetilde{\mathbb{P}}(\Xi_0)} \\
&= (1 - \varepsilon^\star)\frac{\mathbb{P}^\star(\Xi_0)}{\widetilde{\mathbb{P}}(\Xi_0)} \cdot \mathbb{P}^\star_{\Xi_0}(A) + \eta\widetilde{R}_{\Xi_0}(A).
\end{aligned}$$

Note that $\widetilde{\mathbb{P}}(\Xi_0) = (1 - \varepsilon^\star)\mathbb{P}^\star(\Xi_0) + \varepsilon^\star \widetilde{R}(\Xi_0)$, which gives

$$\mathbb{P}^\star(\Xi_0) = \frac{\widetilde{\mathbb{P}}(\Xi_0) - \varepsilon^\star \widetilde{R}(\Xi_0)}{(1 - \varepsilon^\star)}.$$

Therefore

$$(1 - \varepsilon^\star)\frac{\mathbb{P}^\star(\Xi_0)}{\widetilde{\mathbb{P}}(\Xi_0)} = (1 - \varepsilon^\star) \cdot \frac{\widetilde{\mathbb{P}}(\Xi_0) - \varepsilon^\star \widetilde{R}(\Xi_0)}{(1 - \varepsilon^\star)} \cdot \frac{1}{\widetilde{\mathbb{P}}(\Xi_0)} = 1 - \eta,$$

which gives

$$\widetilde{\mathbb{P}}_{\Xi_0}(A) = (1 - \eta)\mathbb{P}^\star_{\Xi_0}(A) + \eta\, \widetilde{R}_{\Xi_0}(A). \tag{16}$$

By assumption, $\mathbb{P}^\star_{\Xi_0} \in \mathcal{A}^{\mathrm{LV}}_{\varepsilon_c, \Xi_0}(\mathbb{P}_{c,\Xi_0})$, so there exists $R_c \in \mathcal{P}(\Xi_0)$ such that

$$\mathbb{P}^\star_{\Xi_0} = (1 - \varepsilon_c)\mathbb{P}_{c,\Xi_0} + \varepsilon_c R_c. \tag{17}$$

If $\widetilde{R}(\Xi_0) = 0$, then we have $\widetilde{\mathbb{P}}_{\Xi_0} = \mathbb{P}^\star_{\Xi_0}$ and we may take $\varepsilon_{\mathrm{eff}} := \varepsilon_c$ and $R_{\mathrm{eff}} := R_c$.

Assume now that $\widetilde{R}(\Xi_0) > 0$ so that (16) holds. Substituting (17) into (16) yields

$$\begin{aligned}
\widetilde{\mathbb{P}}_{\Xi_0} &= (1 - \eta)\big((1 - \varepsilon_c)\mathbb{P}_{c,\Xi_0} + \varepsilon_c R_c\big) + \eta\, \widetilde{R}_{\Xi_0} \\
&= (1 - \eta)(1 - \varepsilon_c)\mathbb{P}_{c,\Xi_0} + (1 - \eta)\varepsilon_c R_c + \eta\, \widetilde{R}_{\Xi_0}.
\end{aligned}$$

Define

$$\varepsilon_{\mathrm{eff}} := 1 - (1 - \eta)(1 - \varepsilon_c) = \eta + \varepsilon_c - \eta\varepsilon_c = 1 - (1 - \varepsilon_c)\left(1 - \frac{\varepsilon^\star \widetilde{R}(\Xi_0)}{\widetilde{\mathbb{P}}(\Xi_0)}\right).$$

If $\varepsilon_{\mathrm{eff}} = 0$, then $\eta = \varepsilon_c = 0$ and hence $\widetilde{\mathbb{P}}_{\Xi_0} = \mathbb{P}_{c,\Xi_0}$, so the inclusion $\widetilde{\mathbb{P}}_{\Xi_0} \in \mathcal{A}^{\mathrm{LV}}_{\varepsilon_{\mathrm{eff}}, \Xi_0}(\mathbb{P}_{c,\Xi_0})$ holds trivially.

Assume now that $\varepsilon_{\mathrm{eff}} > 0$. Define

$$R_{\mathrm{eff}} := \frac{(1 - \eta)\varepsilon_c}{\varepsilon_{\mathrm{eff}}}R_c + \frac{\eta}{\varepsilon_{\mathrm{eff}}}\widetilde{R}_{\Xi_0}.$$

The coefficients are nonnegative and sum to one because

$$\frac{(1 - \eta)\varepsilon_c}{\varepsilon_{\mathrm{eff}}} + \frac{\eta}{\varepsilon_{\mathrm{eff}}} = \frac{(1 - \eta)\varepsilon_c + \eta}{\varepsilon_{\mathrm{eff}}} = \frac{\eta + \varepsilon_c - \eta\varepsilon_c}{\varepsilon_{\mathrm{eff}}} = 1.$$

Since $R_c, \widetilde{R}_{\Xi_0} \in \mathcal{P}(\Xi_0)$, their convex combination $R_{\mathrm{eff}}$ is also in $\mathcal{P}(\Xi_0)$. Therefore,

$$\widetilde{\mathbb{P}}_{\Xi_0} = (1 - \varepsilon_{\mathrm{eff}})\mathbb{P}_{c,\Xi_0} + \varepsilon_{\mathrm{eff}}R_{\mathrm{eff}},$$

which is exactly the statement that

$$\widetilde{\mathbb{P}}_{\Xi_0} \in \mathcal{A}^{\mathrm{LV}}_{\varepsilon_{\mathrm{eff}}, \Xi_0}(\mathbb{P}_{c,\Xi_0}).$$

On $\mathcal{E}$, (15) shows that

$$\widetilde{\mathbb{P}}(\Xi_0^c) \leq (1 - \varepsilon^\star)\gamma + \varepsilon^\star\big(1 - \widetilde{R}(\Xi_0)\big).$$

By assumption, $M_p(x) = \big(\mathbb{E}_{\xi \sim \widetilde{\mathbb{P}}}[|f_x|^p]\big)^{1/p} < \infty$ for all $x$. Therefore we may apply Lemma B.2 with $Q = \widetilde{\mathbb{P}}$, tolerance $\varepsilon = \varepsilon_{\mathrm{eff}}$, and with tail budget $\gamma^\star := \widetilde{\mathbb{P}}(\Xi_0^c)$. This gives, for all $x \in \mathcal{X}$,

$$\mathbb{E}_{\xi \sim \widetilde{\mathbb{P}}}[f_x(\xi)] \leq (1 - \varepsilon_{\mathrm{eff}})\mathbb{E}_{\xi \sim \mathbb{P}_{c,\Xi_0}}[f_x(\xi)] + \varepsilon_{\mathrm{eff}} \sup_{\xi \in \Xi_0} f_x(\xi) + M_p(x)\big(\gamma^\star\big)^{1/q}.$$

Using $\gamma^\star = \widetilde{\mathbb{P}}(\Xi_0^c) \leq (1 - \varepsilon^\star)\gamma + \varepsilon^\star\big(1 - \widetilde{R}(\Xi_0)\big)$ from (15) yields

$$\mathbb{E}_{\xi \sim \widetilde{\mathbb{P}}}[f_x(\xi)] \leq (1 - \varepsilon_{\mathrm{eff}})\mathbb{E}_{\xi \sim \mathbb{P}_{c,\Xi_0}}[f_x(\xi)] + \varepsilon_{\mathrm{eff}} \sup_{\xi \in \Xi_0} f_x(\xi) + M_p(x)\big((1 - \varepsilon^\star)\gamma + \varepsilon^\star\big(1 - \widetilde{R}(\Xi_0)\big)\big)^{1/q}.$$

All steps above hold on $\mathcal{E}$, and $\Pr(\mathcal{E}) \geq 1 - \delta$. Therefore the bound holds with probability at least $1 - \delta$. $\qquad\square$

*Remark* B.3 (When the in-bulk centre mismatch $\varepsilon_c$ vanishes). Fix a bulk set $\Xi_0$ with $\mathbb{P}^\star(\Xi_0) > 0$ and $\mathbb{P}_c(\Xi_0) > 0$, and define

$$\varepsilon_c := \mathrm{LV}(\mathbb{P}^\star_{\Xi_0}, \mathbb{P}_{c,\Xi_0}) = \sup_{A : \mathbb{P}_{c,\Xi_0}(A) > 0} \frac{\mathbb{P}_{c,\Xi_0}(A) - \mathbb{P}^\star_{\Xi_0}(A)}{\mathbb{P}_{c,\Xi_0}(A)} \in [0, 1].$$

A convenient sufficient condition for $\varepsilon_c = o_p(1)$ is an *upper density-ratio* bound on $\Xi_0$: if $\mathbb{P}_{c,\Xi_0} \ll \mathbb{P}^\star_{\Xi_0}$ and there exists $\Delta_n = o_p(1)$ with $\Delta_n \geq 0$ such that

$$\underset{\xi \sim \mathbb{P}^\star_{\Xi_0}}{\mathrm{ess\,sup}} \frac{\mathrm{d}\mathbb{P}_{c,\Xi_0}}{\mathrm{d}\mathbb{P}^\star_{\Xi_0}}(\xi) \leq 1 + \Delta_n, \tag{18}$$

then for every measurable $A \subseteq \Xi_0$, $\mathbb{P}_{c,\Xi_0}(A) \leq (1 + \Delta_n)\mathbb{P}^\star_{\Xi_0}(A)$ and hence $\mathbb{P}^\star_{\Xi_0}(A) \geq \frac{1}{1+\Delta_n}\mathbb{P}_{c,\Xi_0}(A)$. Therefore $\mathbb{P}^\star_{\Xi_0} \in \mathcal{A}^{\mathrm{LV}}_{\Delta_n/(1+\Delta_n),\Xi_0}(\mathbb{P}_{c,\Xi_0})$ and

$$\varepsilon_c \leq \frac{\Delta_n}{1 + \Delta_n} = o_p(1).$$

1. **Bayesian centre.** Suppose $\mathbb{P}^\star = \mathbb{P}_{\theta^\star}$ for some $\theta^\star$ and, on $\Xi_0$, the model admits densities $p_\theta$ with respect to a common dominating measure $\mu$. Assume $\inf_{\xi \in \Xi_0} p_{\theta^\star}(\xi) \geq m_0 > 0$ and that the posterior predictive density $p_c(\xi) := \int p_\theta(\xi)\Pi(\mathrm{d}\theta \mid \mathcal{D}_n)$ satisfies

$$\sup_{\xi \in \Xi_0} |p_c(\xi) - p_{\theta^\star}(\xi)| = o_p(1).$$

   Then $\mathbb{P}_c(\Xi_0) \to \mathbb{P}^\star(\Xi_0)$ and $\sup_{\xi \in \Xi_0} p_c(\xi)/p_{\theta^\star}(\xi) = 1 + o_p(1)$, which implies (18) and hence $\varepsilon_c = o_p(1)$.

2. **Frequentist plug-in centre.** Let $\hat{\theta}_n \to \theta^\star$ in probability and set $\mathbb{P}_c := \mathbb{P}_{\hat{\theta}_n}$. Under the same density regularity on $\Xi_0$ (common dominating measure and $\inf_{\Xi_0} p_{\theta^\star} \geq m_0 > 0$), if $\sup_{\xi \in \Xi_0} |p_{\hat{\theta}_n}(\xi) - p_{\theta^\star}(\xi)| = o_p(1)$, then again (18) holds and $\varepsilon_c = o_p(1)$.

3. **Empirical centre (SAA interpretation).** If we take the empirical measure $\widehat{\mathbb{P}}_n$ as a literal centre in an atomless model (e.g. $\mathbb{P}^\star_{\Xi_0}$ admits a density), then we have $\mathrm{LV}(\mathbb{P}^\star_{\Xi_0}, \widehat{\mathbb{P}}_{n,\Xi_0}) = 1$ because $\widehat{\mathbb{P}}_{n,\Xi_0}$ assigns positive mass to singletons that have $\mathbb{P}^\star_{\Xi_0}$-mass 0. In our empirical option we instead treat $\mathbb{P}^\star$ as the underlying centre and use $\widehat{\mathbb{P}}_n$ only as an SAA for expectations (to approximate $\mathbb{E}_{\mathbb{P}^\star_{\Xi_0}}[f_x]$). In this regime it is not meaningful to define $\varepsilon_c$ as a predictive/estimation mismatch; the relevant error term is the usual SAA deviation $|\mathbb{E}_{\widehat{\mathbb{P}}_{n,\Xi_0}}[f_x] - \mathbb{E}_{\mathbb{P}^\star_{\Xi_0}}[f_x]|$, controlled by standard LLN and concentration assumptions. This interpretation is further demonstrated in Table 6.

*Table 6.* Theoretical centre and SAA target for each centre type. The theoretical target in each row is $(1 - \varepsilon)\mathbb{E}_{\xi \sim \mathbb{P}_{c,\Xi_0}}[f_x(\xi)] + \varepsilon \sup_{\xi \in \Xi_0} f_x(\xi)$, with $\mathbb{P}_c$ instantiated by the centre choice.

| Centre type | Theoretical centre | SAA target |
|---|---|---|
| Bayesian posterior predictive | $\mathbb{P}_c$ with density $p_c(\xi) = \int p_\theta(\xi)\Pi(\mathrm{d}\theta \mid \mathcal{D}_n)$ | $(1-\varepsilon)\frac{1}{M}\sum_{m=1}^{M} f_x(\xi^{\mathrm{B}}_m) + \varepsilon\sup_{\xi \in \Xi_0} f_x(\xi)$, $\xi^{\mathrm{B}}_m \overset{\text{i.i.d.}}{\sim} \mathbb{P}_{c,\Xi_0}$ via rejection sampling. |
| Frequentist plug-in | $\mathbb{P}_c = \mathbb{P}_{\hat{\theta}_n}$ | $(1-\varepsilon)\frac{1}{M}\sum_{m=1}^{M} f_x(\xi^{\mathrm{F}}_m) + \varepsilon\sup_{\xi \in \Xi_0} f_x(\xi)$, $\xi^{\mathrm{F}}_m \overset{\text{i.i.d.}}{\sim} \mathbb{P}_{\hat{\theta}_n,\Xi_0}$ via rejection sampling. |
| Empirical centre (SAA interpretation) | $\mathbb{P}_c = \mathbb{P}^\star$ | $(1-\varepsilon)\frac{\sum_{i=1}^{n} f_x(\xi_i)\mathbf{1}\{\xi_i \in \Xi_0\}}{\sum_{i=1}^{n} \mathbf{1}\{\xi_i \in \Xi_0\}} + \varepsilon\sup_{\xi \in \Xi_0} f_x(\xi)$ |

## B.7. Proof of Results in Table 1: Closed-form In-bulk Supremum

Since the first three rows (linear, ReLU, absolute value) are special cases of the piecewise-linear function, we formalise the following lemma for the piecewise-linear case:

**Lemma B.4** (Closed-form in-bulk supremum for piecewise-linear losses). *Assume $\Sigma_{\mathrm{fit}} \in \mathbb{R}^{d \times d}$ is symmetric positive definite and $w \in \mathbb{R}^d_{++}$. Fix $t \geq 0$ and define the bulk sets*

$$\Xi^{\mathrm{ellip}}_0(t) := \left\{\xi \in \mathbb{R}^d : \left\|\Sigma^{-1/2}_{\mathrm{fit}}(\xi - \mu_{\mathrm{fit}})\right\|_2 \leq t\right\}, \qquad \Xi^{\mathrm{box}}_0(t) := \left\{\xi \in \mathbb{R}^d : \max_i \left|\frac{\xi_i - \mu_{\mathrm{fit},i}}{w_i}\right| \leq t\right\}.$$

*Let $f_x : \mathbb{R}^d \to \mathbb{R}$ be piecewise-linear of the form*

$$f_x(\xi) = \max_{j \in \{1,\dots,J\}} \{a_{x,j}^\top \xi + b_{x,j}\}, \qquad J < \infty.$$

*Write $m_2(a) := \|\Sigma_{\mathrm{fit}}^{1/2} a\|_2$ and $m_1(a) := \sum_{i=1}^d w_i |a_i|$. Then*

$$\sup_{\xi \in \Xi_0^{\mathrm{ellip}}(t)} f_x(\xi) = \max_{j \leq J} \left\{ a_{x,j}^\top \mu_{\mathrm{fit}} + b_{x,j} + t m_2(a_{x,j}) \right\},$$

$$\sup_{\xi \in \Xi_0^{\mathrm{box}}(t)} f_x(\xi) = \max_{j \leq J} \left\{ a_{x,j}^\top \mu_{\mathrm{fit}} + b_{x,j} + t m_1(a_{x,j}) \right\}.$$

*Proof.* For any nonempty set $S \subseteq \mathbb{R}^d$ and any functions $\{g_j\}_{j=1}^J$, we have

$$\sup_{\xi \in S} \max_{j \leq J} g_j(\xi) = \max_{j \leq J} \sup_{\xi \in S} g_j(\xi). \tag{19}$$

Indeed, for every $\xi \in S$, $\max_j g_j(\xi) \leq \max_j \sup_{\zeta \in S} g_j(\zeta)$, hence $\sup_{\xi \in S} \max_j g_j(\xi) \leq \max_j \sup_{\xi \in S} g_j(\xi)$. Conversely, for each fixed $j$, $\sup_{\xi \in S} g_j(\xi) \leq \sup_{\xi \in S} \max_k g_k(\xi)$; taking $\max_j$ gives the reverse inequality.

Fix $a \in \mathbb{R}^d$, $b \in \mathbb{R}$, and let $S = \Xi_0^{\mathrm{ellip}}(t)$. We compute

$$\sup_{\xi \in S}(a^\top \xi + b) = a^\top \mu_{\mathrm{fit}} + b + \sup_{\|\Sigma_{\mathrm{fit}}^{-1/2}(\xi - \mu_{\mathrm{fit}})\|_2 \leq t} a^\top (\xi - \mu_{\mathrm{fit}}).$$

Make the change of variables $u := \Sigma_{\mathrm{fit}}^{-1/2}(\xi - \mu_{\mathrm{fit}})$, so $\xi = \mu_{\mathrm{fit}} + \Sigma_{\mathrm{fit}}^{1/2} u$ and $\|u\|_2 \leq t$. Then

$$a^\top (\xi - \mu_{\mathrm{fit}}) = a^\top \Sigma_{\mathrm{fit}}^{1/2} u = u^\top (\Sigma_{\mathrm{fit}}^{1/2} a).$$

By Cauchy–Schwarz,

$$u^\top (\Sigma_{\mathrm{fit}}^{1/2} a) \leq \|u\|_2 \, \|\Sigma_{\mathrm{fit}}^{1/2} a\|_2 \leq t \, \|\Sigma_{\mathrm{fit}}^{1/2} a\|_2.$$

If $\Sigma_{\mathrm{fit}}^{1/2} a \neq 0$, equality holds by choosing $u = t \Sigma_{\mathrm{fit}}^{1/2} a / \|\Sigma_{\mathrm{fit}}^{1/2} a\|_2$; if $\Sigma_{\mathrm{fit}}^{1/2} a = 0$, then $u^\top (\Sigma_{\mathrm{fit}}^{1/2} a) = 0$ for all $u$. Therefore,

$$\sup_{\xi \in \Xi_0^{\mathrm{ellip}}(t)} (a^\top \xi + b) = a^\top \mu_{\mathrm{fit}} + b + t \, \|\Sigma_{\mathrm{fit}}^{1/2} a\|_2. \tag{20}$$

Fix $a \in \mathbb{R}^d$, $b \in \mathbb{R}$, and let $S = \Xi_0^{\mathrm{box}}(t)$. Write $\xi_i = \mu_{\mathrm{fit},i} + w_i u_i$ with $|u_i| \leq t$, so

$$a^\top \xi + b = a^\top \mu_{\mathrm{fit}} + b + \sum_{i=1}^d (w_i a_i) u_i.$$

For any $(u_i)$ with $|u_i| \leq t$,

$$\sum_{i=1}^d (w_i a_i) u_i \leq \sum_{i=1}^d w_i |a_i| \, |u_i| \leq t \sum_{i=1}^d w_i |a_i| = t \, m_1(a).$$

Equality is achieved by taking $u_i = t \operatorname{sign}(a_i)$ (and any choice in $[-t, t]$ when $a_i = 0$). Thus,

$$\sup_{\xi \in \Xi_0^{\mathrm{box}}(t)} (a^\top \xi + b) = a^\top \mu_{\mathrm{fit}} + b + t \sum_{i=1}^d w_i |a_i|. \tag{21}$$

Apply (19) with $g_j(\xi) = a_{x,j}^\top \xi + b_{x,j}$, and then apply (20) (ellipsoid) or (21) (box) to each $g_j$ to complete the proof. □

*Remark* B.5 (Tractability and special cases.). The expressions above reduce $\sup_{\xi \in \Xi_0(t)} f_x(\xi)$ to a finite maximum of convex terms.

1. For ellipsoids, each piece contributes $a_{x,j}^\top \mu_{\text{fit}} + b_{x,j} + t\|\Sigma_{\text{fit}}^{1/2} a_{x,j}\|_2$. If $a_{x,j}$ depends affinely on the decision variable, this term is SOCP-representable via an epigraph constraint $s_j \geq \|\Sigma_{\text{fit}}^{1/2} a_{x,j}\|_2$.

2. For boxes, each piece contributes $a_{x,j}^\top \mu_{\text{fit}} + b_{x,j} + t\sum_i w_i |a_{x,j,i}|$. If $a_{x,j}$ depends affinely on the decision variable, this term is LP-representable by linearising absolute values with auxiliary variables $u_{j,i} \geq a_{x,j,i}, u_{j,i} \geq -a_{x,j,i}$.

The first three rows of Table 1 are immediate special cases of the lemma:

1. linear loss uses $J = 1$.

2. ReLU uses $J = 2$ with $\max\{0, a_x^\top \xi + b_x\} = \max\{0^\top \xi + 0, a_x^\top \xi + b_x\}$.

3. Absolute value uses $J = 2$ with $|a_x^\top \xi + b_x| = \max\{a_x^\top \xi + b_x, -a_x^\top \xi - b_x\}$.

## C. Blockwise Bulk Construction

**Lemma C.1** (Blockwise bulk-mass certificate via DKW score selection). *Let $\{\tilde{\xi}_j\}_{j=1}^m$ be i.i.d. from $\mathbb{P}^\star$ and independent of $\mathcal{D}_{\text{fit}}$. Fix an integer $k \geq 1$ and a block decomposition $\xi = (\xi^{(1)}, \ldots, \xi^{(k)})$ with $\xi^{(i)} \in \mathbb{R}^{d_i}$ and $\sum_{i=1}^k d_i = d$. For each $i \in \{1, \ldots, k\}$, using $\mathcal{D}_{\text{fit}}$ only, fix a measurable score $s_i : \mathbb{R}^{d_i} \to \mathbb{R}$ and define the nested family*

$$\Xi_{0,i}(t) := \{\xi \in \Xi : s_i(\xi^{(i)}) \leq t\}, \qquad t \in \mathbb{R}.$$

*Let $(\gamma_i, \delta_i) \in (0,1)^2$ be target mass/confidence levels for each block, and define*

$$F_i(t) := \mathbb{P}^\star\{s_i(\tilde{\xi}^{(i)}) \leq t\}, \qquad F_{i,m}(t) := \frac{1}{m}\sum_{j=1}^m \mathbf{1}\{s_i(\tilde{\xi}_j^{(i)}) \leq t\}, \qquad r_{m,\delta_i} := \sqrt{\frac{1}{2m}\log\left(\frac{2}{\delta_i}\right)}.$$

*Assume $\gamma_i \geq r_{m,\delta_i}$ for all $i$. Define the one-sided DKW envelope for each block by*

$$L_i^{\text{DKW}}(t) := \left[F_{i,m}(t) - r_{m,\delta_i}\right]_+, \qquad \hat{t}_i := \inf\{t : L_i^{\text{DKW}}(t) \geq 1 - \gamma_i\}, \qquad \widehat{\Xi}_{0,i} := \Xi_{0,i}(\hat{t}_i),$$

*and form the blockwise intersection bulk set*

$$\widehat{\Xi}_0 := \bigcap_{i=1}^k \widehat{\Xi}_{0,i}.$$

*Then, with probability at least $1 - \sum_{i=1}^k \delta_i$,*

$$\mathbb{P}^\star(\widehat{\Xi}_0) \geq 1 - \sum_{i=1}^k \gamma_i.$$

*In particular, if $\sum_{i=1}^k \gamma_i \leq \gamma$ and $\sum_{i=1}^k \delta_i \leq \delta$, then*

$$\Pr\left\{\mathbb{P}^\star(\widehat{\Xi}_0) \geq 1 - \gamma\right\} \geq 1 - \delta.$$

*Proof.* Condition on $\mathcal{D}_{\text{fit}}$, so that each score $s_i$ (hence each family $\{\Xi_{0,i}(t)\}$) is fixed. For each $i$, apply the DKW inequality (4) to the one-dimensional random variable $Z_{i,j} := s_i(\tilde{\xi}_j^{(i)})$ with confidence level $\delta_i$:

$$\Pr\left\{\sup_{t\in\mathbb{R}}|F_{i,m}(t) - F_i(t)| \leq r_{m,\delta_i} \;\middle|\; \mathcal{D}_{\text{fit}}\right\} \geq 1 - \delta_i.$$

Let

$$A_i := \left\{\sup_{t\in\mathbb{R}}|F_{i,m}(t) - F_i(t)| \leq r_{m,\delta_i}\right\}.$$

By the union bound,

$$\Pr\left(\bigcap_{i=1}^k A_i \;\middle|\; \mathcal{D}_{\text{fit}}\right) \geq 1 - \sum_{i=1}^k \delta_i.$$

On the event $A_i$, for every $t \in \mathbb{R}$ we have

$$F_i(t) \geq F_{i,m}(t) - r_{m,\delta_i}.$$

Since $F_i(t) \geq 0$ for all $t$, it follows that

$$F_i(t) \geq \max\{F_{i,m}(t) - r_{m,\delta_i}, 0\} = \left[F_{i,m}(t) - r_{m,\delta_i}\right]_+ = L_i^{\mathrm{DKW}}(t).$$

Note that $F_{i,m}$ (hence $L_i^{\mathrm{DKW}}$) is nondecreasing and right-continuous in $t$, and since $\gamma_i \geq r_{m,\delta_i}$ the set $\{t : L_i^{\mathrm{DKW}}(t) \geq 1 - \gamma_i\}$ is nonempty (e.g. it contains $t = \max_j s_i(\tilde{\xi}_j^{(i)})$). Therefore the infimum $\hat{t}_i$ is well-defined and satisfies $L_i^{\mathrm{DKW}}(\hat{t}_i) \geq 1 - \gamma_i$.

$$\mathbb{P}^\star(\widehat{\Xi}_{0,i}) = \mathbb{P}^\star\big(\Xi_{0,i}(\hat{t}_i)\big) = F_i(\hat{t}_i) \geq L_i^{\mathrm{DKW}}(\hat{t}_i) \geq 1 - \gamma_i.$$

Therefore, on $\bigcap_{i=1}^k A_i$, we simultaneously have $\mathbb{P}^\star(\widehat{\Xi}_{0,i}^c) \leq \gamma_i$ for all $i$. Using $\widehat{\Xi}_0 = \bigcap_{i=1}^k \widehat{\Xi}_{0,i}$ and De Morgan's law,

$$\mathbb{P}^\star(\widehat{\Xi}_0^c) = \mathbb{P}^\star\Big(\bigcup_{i=1}^k \widehat{\Xi}_{0,i}^c\Big) \leq \sum_{i=1}^k \mathbb{P}^\star(\widehat{\Xi}_{0,i}^c) \leq \sum_{i=1}^k \gamma_i,$$

where we used the union bound for probabilities in the first inequality. Equivalently, on $\bigcap_{i=1}^k A_i$,

$$\mathbb{P}^\star(\widehat{\Xi}_0) \geq 1 - \sum_{i=1}^k \gamma_i.$$

Combining with $\Pr(\cap_i A_i \mid \mathcal{D}_{\mathrm{fit}}) \geq 1 - \sum_i \delta_i$ and then removing the conditioning yields

$$\Pr\Big\{\mathbb{P}^\star(\widehat{\Xi}_0) \geq 1 - \sum_{i=1}^k \gamma_i\Big\} \geq 1 - \sum_{i=1}^k \delta_i.$$

The final statement follows immediately when $\sum_i \gamma_i \leq \gamma$ and $\sum_i \delta_i \leq \delta$. $\qquad\square$

## D. Discussion on Tolerance Selection: Calibrating $\varepsilon$

Here we separate two scenarios:

1. At decision time, we only observe training data from $\mathbb{P}^\star$, so the deployment contamination level in $\tilde{\mathbb{P}} = (1-\varepsilon^\star)\mathbb{P}^\star + \varepsilon^\star \widetilde{R}$ is not identifiable from training data alone without strong additional assumptions. In this regime, $\varepsilon$ is treated as a robustness budget and tuned via validation. The score-based diagnostic in Lemma D.1 is only used to flag potential centre misalignment $\varepsilon_c$, and hence provides a lower bound on the effective distortion $\varepsilon_{\mathrm{eff}} \geq \varepsilon_c$.

2. If we have i.i.d. samples from $\tilde{\mathbb{P}}$, then we provide a lemma characterising $\varepsilon_{\Xi_0} = \mathrm{LV}(\tilde{\mathbb{P}}_{\Xi_0}, \mathbb{P}_{c,\Xi_0})$ as one minus the ess inf of the Radon–Nikodym derivative. This is *not* the setting in our paper: we only include it for completeness.

### D.1. A Score-based Lower-bound Diagnostic for $\varepsilon_c$

We show that nested score sets inside the calibrated bulk yield a simple lower bound on the in-bulk mismatch parameter $\varepsilon_c$. The diagnostic uses only the selection scores already computed for DKW calibration and the centre samples already generated for rejection sampling.

**Lemma D.1** (Score-based lower bound for $\varepsilon_c$). *Fix a measurable score $s : \Xi \to \mathbb{R}$ and thresholds $t_1 < \cdots < t_K$. Let $\Xi_0 := \{\xi : s(\xi) \leq t_K\}$ and $\Xi_0(t_k) := \{\xi : s(\xi) \leq t_k\} \subseteq \Xi_0$. Assume $\mathbb{P}^\star(\Xi_0) \geq 1 - \gamma$ for some $\gamma \in [0, 1)$ and $\mathbb{P}_c(\Xi_0) > 0$.*

*Assume further that the in-bulk training law is within a forward LV ball around the centre, i.e.*

$$\mathbb{P}_{\Xi_0}^\star \in \mathcal{A}_{\varepsilon_c, \Xi_0}^{\mathrm{LV}}(\mathbb{P}_{c,\Xi_0}) \quad \textit{for some } \varepsilon_c \in [0, 1). \tag{22}$$

Let $\mathcal{D}_{\text{select}} = \{\tilde{\xi}_j\}_{j=1}^m$ be i.i.d. from $\mathbb{P}^\star$ and let $\{\xi_c^{(i)}\}_{i=1}^{N_c}$ be i.i.d. from $\mathbb{P}_c$. The probability statement below is with respect to the randomness of $\mathcal{D}_{\text{select}}$ and $\{\xi_c^{(i)}\}_{i=1}^{N_c}$, and remains valid conditionally on $\mathbb{P}_c$ and on $s, t_1, \ldots, t_K$. Define, for $k = 1, \ldots, K$,

$$\widehat{p}_k^\star := \frac{1}{m} \sum_{j=1}^m \mathbf{1}\left\{s\left(\tilde{\xi}_j\right) \leq t_k\right\}, \qquad \widehat{p}_k^g := \frac{1}{N_c} \sum_{i=1}^{N_c} \mathbf{1}\left\{s\left(\xi_c^{(i)}\right) \leq t_k\right\}.$$

Fix $\delta \in (0, 1)$ and set the DKW radii

$$r_{m,\delta/2} := \sqrt{\frac{1}{2m} \log\left(\frac{4}{\delta}\right)}, \qquad r_{c,\delta/2} := \sqrt{\frac{1}{2N_c} \log\left(\frac{4}{\delta}\right)}.$$

Define the computable lower-bound diagnostic

$$\varepsilon_{c,\delta}^{\text{score}} := \begin{cases} \max\limits_{k:\widehat{p}_k^g > r_{c,\delta/2}} \left[1 - \frac{(\widehat{p}_k^\star + r_{m,\delta/2})(\widehat{p}_K^g + r_{c,\delta/2})}{(1-\gamma)(\widehat{p}_k^g - r_{c,\delta/2})}\right]_+, & \text{if } \exists k \quad s.t. \ \widehat{p}_k^g > r_{c,\delta/2}, \\ 0, & \text{otherwise.} \end{cases} \tag{23}$$

Then, with probability at least $1 - \delta$,

$$\varepsilon_c \geq \varepsilon_{c,\delta}^{\text{score}}.$$

*Proof.* Since (22) holds, there exists a probability measure $R \in \mathcal{P}(\Xi_0)$ such that, for all $A \in \mathcal{F}$,

$$\mathbb{P}_{\Xi_0}^\star(A) = (1 - \varepsilon_c)\mathbb{P}_{c,\Xi_0}(A) + \varepsilon_c R(A). \tag{24}$$

Fix $k \in \{1, \ldots, K\}$ and apply (24) with $A = \Xi_0(t_k)$. Because $R$ is a probability measure, $R(\Xi_0(t_k)) \geq 0$, hence

$$\mathbb{P}_{\Xi_0}^\star(\Xi_0(t_k)) = (1 - \varepsilon_c)\mathbb{P}_{c,\Xi_0}(\Xi_0(t_k)) + \varepsilon_c R(\Xi_0(t_k)) \geq (1 - \varepsilon_c)\mathbb{P}_{c,\Xi_0}(\Xi_0(t_k)).$$

Assume $\mathbb{P}_{c,\Xi_0}(\Xi_0(t_k)) > 0$ (equivalently $\mathbb{P}_c(\Xi_0(t_k)) > 0$). Dividing both sides by $\mathbb{P}_{c,\Xi_0}(\Xi_0(t_k))$ gives

$$\frac{\mathbb{P}_{\Xi_0}^\star(\Xi_0(t_k))}{\mathbb{P}_{c,\Xi_0}(\Xi_0(t_k))} \geq 1 - \varepsilon_c,$$

and thus

$$\varepsilon_c \geq 1 - \frac{\mathbb{P}_{\Xi_0}^\star(\Xi_0(t_k))}{\mathbb{P}_{c,\Xi_0}(\Xi_0(t_k))}. \tag{25}$$

We now rewrite the ratio in (25) using unnormalised masses. By definition of truncation,

$$\mathbb{P}_{\Xi_0}^\star(\Xi_0(t_k)) = \frac{\mathbb{P}^\star(\Xi_0(t_k) \cap \Xi_0)}{\mathbb{P}^\star(\Xi_0)} = \frac{\mathbb{P}^\star(\Xi_0(t_k))}{\mathbb{P}^\star(\Xi_0)},$$

since $\Xi_0(t_k) \subseteq \Xi_0$. Similarly,

$$\mathbb{P}_{c,\Xi_0}(\Xi_0(t_k)) = \frac{\mathbb{P}_c(\Xi_0(t_k))}{\mathbb{P}_c(\Xi_0)}.$$

Substituting these identities into (25) yields

$$\varepsilon_c \geq 1 - \frac{\mathbb{P}^\star(\Xi_0(t_k)) \, \mathbb{P}_c(\Xi_0)}{\mathbb{P}^\star(\Xi_0) \, \mathbb{P}_c(\Xi_0(t_k))}. \tag{26}$$

Finally, using $\mathbb{P}^\star(\Xi_0) \geq 1 - \gamma$ gives the weaker but simpler bound

$$\varepsilon_c \geq 1 - \frac{\mathbb{P}^\star(\Xi_0(t_k)) \, \mathbb{P}_c(\Xi_0)}{(1 - \gamma) \, \mathbb{P}_c(\Xi_0(t_k))}. \tag{27}$$

Define the score CDFs

$$F^\star(t) := \mathbb{P}^\star\{s(\xi) \leq t\}, \qquad F^c(t) := \mathbb{P}_c\{s(\xi) \leq t\}.$$

By construction, $F^\star(t_k) = \mathbb{P}^\star(\Xi_0(t_k))$ and $F^c(t_k) = \mathbb{P}_c(\Xi_0(t_k))$ for each $k$. Also, $\widehat{p}_k^\star$ and $\widehat{p}_k^g$ are the corresponding empirical CDF values at $t_k$.

Let $\mathcal{E}_\star$ be the DKW event for $\mathcal{D}_{\text{select}}$:

$$\mathcal{E}_\star := \left\{ \sup_{t \in \mathbb{R}} \left| \widehat{F}_m^\star(t) - F^\star(t) \right| \leq r_{m,\delta/2} \right\},$$

where $\widehat{F}_m^\star$ is the empirical CDF of $\{s(\tilde{\xi}_j)\}_{j=1}^m$. By the DKW inequality, $\Pr(\mathcal{E}_\star) \geq 1 - \delta/2$. On $\mathcal{E}_\star$, for every $k$,

$$\mathbb{P}^\star(\Xi_0(t_k)) = F^\star(t_k) \leq \widehat{F}_m^\star(t_k) + r_{m,\delta/2} = \widehat{p}_k^\star + r_{m,\delta/2}. \tag{28}$$

Similarly, let $\mathcal{E}_c$ be the DKW event for the centre samples:

$$\mathcal{E}_c := \left\{ \sup_{t \in \mathbb{R}} \left| \widehat{F}_{N_c}^c(t) - F^c(t) \right| \leq r_{c,\delta/2} \right\},$$

where $\widehat{F}_{N_c}^c$ is the empirical CDF of $\{s(\xi_c^{(i)})\}_{i=1}^{N_c}$. Again by DKW (conditionally on $\mathbb{P}_c$), $\Pr(\mathcal{E}_c) \geq 1 - \delta/2$. On $\mathcal{E}_c$, for every $k$,

$$\mathbb{P}_c(\Xi_0(t_k)) = F^c(t_k) \geq \widehat{F}_{N_c}^c(t_k) - r_{c,\delta/2} = \widehat{p}_k^g - r_{c,\delta/2}, \qquad \mathbb{P}_c(\Xi_0) = F^c(t_K) \leq \widehat{p}_K^g + r_{c,\delta/2}. \tag{29}$$

By a union bound, $\Pr(\mathcal{E}_\star \cap \mathcal{E}_c) \geq 1 - \delta$.

Fix any $k$ such that $\widehat{p}_k^g > r_{c,\delta/2}$, so that $\widehat{p}_k^g - r_{c,\delta/2} > 0$. On $\mathcal{E}_\star \cap \mathcal{E}_c$, substitute (28) and (29) into (27) to obtain

$$\varepsilon_c \geq 1 - \frac{(\widehat{p}_k^\star + r_{m,\delta/2})(\widehat{p}_K^g + r_{c,\delta/2})}{(1 - \gamma)(\widehat{p}_k^g - r_{c,\delta/2})}.$$

Taking $[\cdot]_+$ preserves the inequality, and taking the maximum over all $k$ with $\widehat{p}_k^g > r_{c,\delta/2}$ yields

$$\varepsilon_c \geq \underline{\varepsilon}_{c,\delta}^{\text{score}}$$

on $\mathcal{E}_\star \cap \mathcal{E}_c$. Since $\Pr(\mathcal{E}_\star \cap \mathcal{E}_c) \geq 1 - \delta$, the claim follows. $\square$

*Remark D.2.* The bound (23) is conservative by construction. Dropping the radii $r_{m,\delta/2}, r_{c,\delta/2}$ yields a simpler point diagnostic, but without a high-probability guarantee.

### D.2. Characterising $\varepsilon_{\Xi_0}$ when Both Measures are Observed

Let $\mathbb{P}$ denote the chosen centre on $\Xi_0$ (typically $\mathbb{P}_{c,\Xi_0}$) and let $Q$ denote the target law on $\Xi_0$ (e.g. $\mathbb{P}_{\Xi_0}^\star$ when train and test match, or $\tilde{\mathbb{P}}_{\Xi_0}$ when a post-deployment batch is available). We focus on

$$\varepsilon_{\Xi_0} := \text{LV}(Q, \mathbb{P}).$$

Lemma D.3 shows that $\varepsilon_{\Xi_0}$ reduces to an essential infimum of a Radon–Nikodym derivative.

**Lemma D.3** (Radon–Nikodym characterisation of LV). *Let $(\Xi, \mathcal{F})$ be a measurable space, and let $\mathbb{P}$ and $Q$ be probability measures on $(\Xi, \mathcal{F})$. Let $Q = Q^a + Q^s$ be the Lebesgue decomposition of $Q$ with respect to $\mathbb{P}$, i.e. $Q^a \ll \mathbb{P}$ and $Q^s \perp \mathbb{P}$. Let $f := \frac{dQ^a}{d\mathbb{P}}$ be the Radon–Nikodym derivative. Then*

$$\text{LV}(Q, \mathbb{P}) = 1 - \text{ess inf}_{\mathbb{P}} f. \tag{30}$$

*In particular, if $\mathbb{P}$ and $Q$ admit densities $p$ and $q$ with respect to Lebesgue measure $\lambda$ and $Q \ll \mathbb{P}$, then*

$$\text{LV}(Q, \mathbb{P}) = 1 - \text{ess inf}_{\mathbb{P}} \left( \frac{q}{p} \right) = 1 - \text{ess inf}_{\{\xi: p(\xi) > 0\}} \frac{q(\xi)}{p(\xi)}. \tag{31}$$

*Proof.* Recall the definition:

$$\mathrm{LV}(Q, \mathbb{P}) = \sup_{A \in \mathcal{F}: \mathbb{P}(A) > 0} \frac{\mathbb{P}(A) - Q(A)}{\mathbb{P}(A)} = 1 - \inf_{A \in \mathcal{F}: \mathbb{P}(A) > 0} \frac{Q(A)}{\mathbb{P}(A)}. \tag{32}$$

Let $Q = Q^a + Q^s$ be the Lebesgue decomposition of $Q$ with respect to $\mathbb{P}$. Then we have

$$\frac{Q(A)}{\mathbb{P}(A)} = \frac{Q^a(A) + Q^s(A)}{\mathbb{P}(A)} \geq \frac{Q^a(A)}{\mathbb{P}(A)}, \quad \forall A \in \mathcal{F}: \mathbb{P}(A) > 0 \implies \inf_{A \in \mathcal{F}: \mathbb{P}(A) > 0} \frac{Q(A)}{\mathbb{P}(A)} \geq \inf_{A \in \mathcal{F}: \mathbb{P}(A) > 0} \frac{Q^a(A)}{\mathbb{P}(A)} \tag{33}$$

Let $S \in \mathcal{F}$ be a $\mathbb{P}$-null set supporting $Q^s$, i.e., $Q^s(S^c) = 0$ and $\mathbb{P}(S) = 0$, which exists by definition of singularity. Then for any $A \in \mathcal{F}$, we have $\mathbb{P}(A) = \mathbb{P}(A \cap S^c)$. Since $Q^a \ll \mathbb{P}$, we have $Q^a(S) = 0$ and $Q^a(A) = Q^a(A \cap S^c)$, and $Q^s(A \cap S^c) = 0$. This means

$$\frac{Q^a(A)}{\mathbb{P}(A)} = \frac{Q^a(A \cap S^c)}{\mathbb{P}(A \cap S^c)} = \frac{Q(A \cap S^c)}{\mathbb{P}(A \cap S^c)} \quad \forall A \in \mathcal{F}: \mathbb{P}(A) > 0,$$

so

$$\inf_{A \in \mathcal{F}: \mathbb{P}(A) > 0} \frac{Q^a(A)}{\mathbb{P}(A)} = \inf_{A \in \mathcal{F}: \mathbb{P}(A) > 0} \frac{Q(A \cap S^c)}{\mathbb{P}(A \cap S^c)} = \inf_{B \subseteq S^c, B \in \mathcal{F}: \mathbb{P}(B) > 0} \frac{Q(B)}{\mathbb{P}(B)} \geq \inf_{A \in \mathcal{F}: \mathbb{P}(A) > 0} \frac{Q(A)}{\mathbb{P}(A)} \tag{34}$$

Combining (33) and (34) shows

$$\inf_{A \in \mathcal{F}: \mathbb{P}(A) > 0} \frac{Q(A)}{\mathbb{P}(A)} = \inf_{A \in \mathcal{F}: \mathbb{P}(A) > 0} \frac{Q^a(A)}{\mathbb{P}(A)}. \tag{35}$$

Since $Q^a \ll \mathbb{P}$, let $f = \frac{\mathrm{d}Q^a}{\mathrm{d}\mathbb{P}}$ be the Radon–Nikodym derivative, and we show that

$$\inf_{A \in \mathcal{F}: \mathbb{P}(A) > 0} \frac{Q^a(A)}{\mathbb{P}(A)} = \operatorname{ess\,inf}_{\mathbb{P}} f. \tag{36}$$

Write $c := \operatorname{ess\,inf}_{\mathbb{P}} f \in [-\infty, \infty)$. By definition of essential infimum

$$\mathbb{P}\big(\{\xi \in \Xi : f(\xi) < c\}\big) = 0. \tag{37}$$

For any $A \in \mathcal{F}$ with $\mathbb{P}(A) > 0$, (37) gives $\frac{Q^a(A)}{\mathbb{P}(A)} = \frac{1}{\mathbb{P}(A)} \int_A f \mathrm{d}\mathbb{P} \geq \frac{1}{\mathbb{P}(A)} \int_A c \, \mathrm{d}\mathbb{P} = c$. Taking the infimum gives

$$\inf_{A \in \mathcal{F}: \mathbb{P}(A) > 0} \frac{Q^a(A)}{\mathbb{P}(A)} \geq c. \tag{38}$$

For every $\varepsilon > 0$ the set $A_\varepsilon := \{\xi \in \Xi : f(\xi) < c + \varepsilon\}$ is measurable and satisfies $\mathbb{P}(A_\varepsilon) > 0$. Since $f \leq c + \varepsilon$ $\mathbb{P}$-a.s. on $A_\varepsilon$, we have

$$\frac{Q^a(A_\varepsilon)}{\mathbb{P}(A_\varepsilon)} = \frac{1}{\mathbb{P}(A_\varepsilon)} \int_{A_\varepsilon} f \mathrm{d}\mathbb{P} \leq c + \varepsilon.$$

Therefore

$$\inf_{A: \mathbb{P}(A) > 0} \frac{Q^a(A)}{\mathbb{P}(A)} \leq c + \varepsilon \quad \text{for all } \varepsilon > 0, \tag{39}$$

and letting $\varepsilon \downarrow 0$ yields the reverse inequality to (38), which proves (36).

For the density specialisation, suppose $\mathbb{P}$ and $Q$ admit densities $p$ and $q$ with respect to Lebesgue measure $\lambda$, and $Q \ll \mathbb{P}$. Since $\mathbb{P}(\{p = 0\}) = \int_{\{p=0\}} p \mathrm{d}\lambda = 0$ and $Q \ll \mathbb{P}$, we have $Q(\{p = 0\}) = 0$, i.e. $\int_{\{p=0\}} q \mathrm{d}\lambda = 0$. Define

$$f(\xi) := \begin{cases} \dfrac{q(\xi)}{p(\xi)}, & \text{if } p(\xi) > 0, \\ 0, & \text{if } p(\xi) = 0, \end{cases}$$

which is $\mathbb{P}$-measurable. Then for any $A \in \mathcal{F}$,

$$\int_A f \mathrm{d}\mathbb{P} = \int_{A \cap \{p > 0\}} \frac{q}{p} \cdot p \, \mathrm{d}\lambda + \underbrace{\int_{A \cap \{p = 0\}} 0 \cdot p \, \mathrm{d}\lambda}_{=0} = \int_A q \mathrm{d}\lambda = Q(A),$$

so $f = \frac{\mathrm{d}Q}{\mathrm{d}\mathbb{P}}$ $\mathbb{P}$-a.s. Since $\mathbb{P}(\{p = 0\}) = 0$, the essential infimum with respect to $\mathbb{P}$ ignores the values of $f$ on $\{p = 0\}$, and (31) follows from (30). $\square$

## E. Discussion on other Contamination-related Ambiguity Sets

This appendix discusses two practical issues that arise when $\Xi$ is unbounded and the loss family $\{f_x\}_{x\in\mathcal{X}}$ is unbounded above. Throughout, $\mathbb{P}^\star$ denotes the data-generating law of $\xi$. First, discretising $\Xi$ can make the max-based objectives diverge as the number of scenarios grows, including more extreme values. Second, while bounding $\Xi$ via a confidence set $\Omega$ can stabilise the objective, without a certificate of the form $\mathbb{P}^\star(\Omega) \geq 1 - \gamma$, the truncation is uncontrolled, so the robustness target depends on an uncalibrated modelling choice without any OOS coverage guarantee.

### E.1. TV Ambiguity Sets

For a TV ball around a centre $\mathbb{P}$, the worst-case risk satisfies

$$\sup_{Q:\mathrm{TV}(Q,\mathbb{P})\leq\varepsilon} \mathbb{E}_Q[f_x] = (1-\varepsilon)\mathrm{CVaR}^{\mathbb{P}}_{1-\varepsilon}(f_x) + \varepsilon \sup_{\xi\in\Xi} f_x(\xi),$$

so if $\sup_\Xi f_x = \infty$, then the robust risk is infinite for every $\varepsilon > 0$. To ensure finiteness, Jiang & Guan (2018) work on a bounded sample space $\Omega \subset \Xi$ and assume $f_x$ is bounded on $\mathcal{X} \times \Omega$ (their Assumption (A1)). In their general-sample-space discussion (Jiang & Guan, 2018, Section 3.3), they approximate $\sup_{\xi\in\Omega} f_x(\xi)$ by a scenario max over sampled support points $\{\hat{\xi}^1, \ldots, \hat{\xi}^{S_\Omega}\} \subset \Omega$. Their resulting SAA objective (omitting deterministic first-stage terms) takes the form

$$\widehat{\mathcal{R}}_{\mathcal{B}^\varepsilon_{\mathrm{TV}}(\mathbb{P})}(f_x) := \min_{x\in\mathcal{X},\eta\in\mathbb{R}}(1-\varepsilon)\eta + \frac{1}{S}\sum_{s=1}^{S}\big(f_x(\xi^s)-\eta\big)^+ + \varepsilon \max_{1\leq s\leq S_\Omega} f_x(\hat{\xi}^s), \qquad \varepsilon \in (0,1], \tag{40}$$

where $\{\xi^s\}_{s=1}^S$ are scenarios used for the CVaR term and $\{\hat{\xi}^s\}_{s=1}^{S_\Omega}$ are support scenarios used for the max term.

If $f_x \geq 0$ for all $x \in \mathcal{X}$, then the first two terms in (40) are nonnegative, hence

$$\widehat{\mathcal{R}}_{\mathcal{B}^\varepsilon_{\mathrm{TV}}(\mathbb{P})}(f_x) \geq \varepsilon \inf_{x\in\mathcal{X}} \max_{1\leq s\leq S_\Omega} f_x(\hat{\xi}^s). \tag{41}$$

When the $\{\hat{\xi}^s\}_{s=1}^{S_\Omega}$'s are i.i.d. from $\mathbb{P}^\star$ with unbounded support, the lower bound in (41) can diverge as $S_\Omega \to \infty$ for losses that grow at least linearly in the tail, because $\max_{s\leq S_\Omega} f_x(\hat{\xi}^s)$ is driven by increasingly extreme samples while a single $x$ must hedge all scenarios simultaneously. Examples include:

1. (Newsvendor loss in Section 5.1.) With $f_x(\xi) = \sum_{j=1}^d (h[x_j - \xi_j]_+ + b[\xi_j - x_j]_+)$ and $h, b > 0$, fix any coordinate $i$ and set $\alpha_i := \min_{s\leq S_\Omega} \hat{\xi}_i^s$ and $\beta_i := \max_{s\leq S_\Omega} \hat{\xi}_i^s$. A one-dimensional reduction gives

$$\inf_{x\in\mathcal{X}} \max_{s\leq S_\Omega} f_x(\hat{\xi}^s) \geq \inf_{x_i\in\mathbb{R}} \max_{s\leq S_\Omega}\big(h[x_i - \hat{\xi}_i^s]_+ + b[\hat{\xi}_i^s - x_i]_+\big) = \frac{hb}{h+b}(\beta_i - \alpha_i).$$

   If $\mathbb{P}^\star(\xi_i > t) > 0$ for all $t$, then $\beta_i \to \infty$ almost surely as $S_\Omega \to \infty$, so the right-hand side diverges.

2. (Regression MAE in Section 5.2.) In the California housing regression we use $f_{w,b}(\xi) = |Y - w^\top X - b|$ with decision $x = (w, b)$ and outcome $\xi = (X, Y)$. Assume $\|w\|_2 \leq B_w$ and $|b| \leq B_b$ for all $(w, b) \in \mathcal{X}$. If there exists $R < \infty$ such that

$$\mathbb{P}^\star\big(\|X\|_2 \leq R, \, |Y| > t\big) > 0 \qquad \forall t > 0, \tag{42}$$

   then for any sample $\hat{\xi}^s = (\hat{X}^s, \hat{Y}^s)$ with $\|\hat{X}^s\|_2 \leq R$,

$$f_{w,b}(\hat{\xi}^s) = |\hat{Y}^s - w^\top \hat{X}^s - b| \geq |\hat{Y}^s| - \|w\|_2\|\hat{X}^s\|_2 - |b| \geq |\hat{Y}^s| - B_w R - B_b.$$

   Hence

$$\inf_{(w,b)\in\mathcal{X}} \max_{s\leq S_\Omega} f_{w,b}(\hat{\xi}^s) \geq \max_{s\leq S_\Omega: \|\hat{X}^s\|_2\leq R}\big(|\hat{Y}^s| - B_w R - B_b\big),$$

   which diverges almost surely as $S_\Omega \to \infty$ under (42).

To justify a bounded $\Omega$ in practice, Jiang & Guan (2018, Section 2.1) discuss estimating confidence sets (which they refer to as confidence regions) of the sample space. In particular, they propose using a "componentwise three-sigma" box or a Markov-type ellipsoid based on first and second moments. While the form of these confidence sets resembles our bulk-set geometries, without a high-probability certificate of the form $\mathbb{P}^\star(\Omega) \geq 1 - \gamma$, the tail mass outside $\Omega$ is uncontrolled, so a term like $\sup_{\xi \in \Omega} f_x(\xi)$ lacks a certified OOS interpretation.

For the three-sigma box, consider $d \geq 10$ and $\xi \in \mathbb{R}^d$ distributed as

$$\xi = \pm e_i \quad \text{with probability } 1/(2d) \text{ for each } i \in \{1, \ldots, d\}.$$

Then $\mathbb{E}[\xi] = 0$ and $\mathrm{Var}(\xi_k) = 1/d$. The three-sigma bound gives $|\xi_k| \leq 3/\sqrt{d} < 1$, hence $\Omega = [-3/\sqrt{d}, 3/\sqrt{d}]^d$, but $\|\xi\|_\infty = 1$ almost surely and thus $\mathbb{P}^\star(\Omega) = 0$.

In contrast, our DKW bulk calibration can recover a meaningful truncation. For the box geometry, we can take $s(\xi) = \|\xi\|_\infty$. Since $s(\xi) = 1$ almost surely under $\mathbb{P}^\star$, the calibrated threshold is $\hat{t}_{\mathrm{DKW}} = 1$ and the resulting bulk set $\Xi_0 = \{\xi : \|\xi\|_\infty \leq 1\}$ satisfies $\mathbb{P}^\star(\Xi_0) = 1$ and is certified with probability at least $1 - \delta$ by Lemma 3.2.

For the Markov-type ellipsoid, Jiang & Guan (2018, Section 2.1) define $\Omega_\varphi := \{\xi : (\xi - \bar{\mu})^\top \bar{\Sigma}^{-1}(\xi - \bar{\mu}) \leq \varphi\}$ using sample moments $(\bar{\mu}, \bar{\Sigma})$ and then invoke a second-moment bound to provide a lower bound on $\mathbb{P}^\star(\Omega_\varphi)$ by Markov's inequality. However, if $\mathbb{P}^\star$ is heavy-tailed with infinite second moment, the Markov bound is vacuous, and the coverage guarantee fails.

Our DKW procedure avoids such moment assumptions: we can still use a Mahalanobis-type score $s(\xi) := \|\hat{\Sigma}^{-1/2}(\xi - \hat{\mu})\|_2$ (with $(\hat{\mu}, \hat{\Sigma})$ fitted on a score-fitting set), and select $\hat{t}_{\mathrm{DKW}}$ from an independent selection set so that, with probability at least $1 - \delta$,

$$\mathbb{P}^\star\{s(\xi) \leq \hat{t}_{\mathrm{DKW}}\} \geq 1 - \gamma,$$

yielding an ellipsoidal bulk set $\Xi_0 = \{\xi : (\xi - \hat{\mu})^\top \hat{\Sigma}^{-1}(\xi - \hat{\mu}) \leq \hat{t}_{\mathrm{DKW}}^2\}$ without requiring finite second moments.

Additionally, the polyhedral-support assumption used for tractability (Jiang & Guan, 2018, Assumption (A2)) can substantially inflate the sup term by ignoring dependence. For instance, if $\xi = (Z, -Z)$ with $|Z| \leq 1$ almost surely, then the true support is the line segment $\{(z, -z) : |z| \leq 1\}$, but the box relaxation $[-1, 1]^2$ allows both coordinates to move adversely and enlarges the sup term, for example under the linear loss function:

$$\sup_{\xi \in [-1,1]^2} u^\top \xi = |u_1| + |u_2|, \quad \text{whereas} \quad \sup_{|z| \leq 1} u^\top(z, -z) = |u_1 - u_2|.$$

Whenever a TV-type objective includes $\sup_\Omega f_x$, such relaxations can lead to overly conservative decisions.

This is why we choose an ellipsoidal bulk set by default when dimensions may be dependent (e.g., per-item demands in the newsvendor experiment, covariates in the California housing regression experiment, hashed features in the CivilComments experiment), and construct separate-dimension bulk sets when we intend to encode a targeted conditional shift (e.g., treating the outcome label separately in California housing regression). This keeps the ambiguity set aligned with the intended shift type and avoids over-conservatism arising from allowing adverse coordinate-wise movements to occur simultaneously.

## E.2. Kernel-based Ambiguity Sets

Let $k$ be a reproducing kernel on $\Xi$ with a reproducing kernel Hilbert space (RKHS) $\mathcal{H}$, and assume $\kappa^2 := \sup_{\xi \in \Xi} k(\xi, \xi) < \infty$. Define the maximum mean discrepancy (MMD) as

$$\mathrm{MMD}_k(\mathbb{P}, Q) = \|\mu_\mathbb{P} - \mu_Q\|_\mathcal{H}, \qquad \mu_\mathbb{P} := \int k(\cdot, \xi) \, \mathrm{d}\mathbb{P}(\xi),$$

following Gretton et al. (2012). For any $\mathbb{P}, R \in \mathcal{P}(\Xi)$ and $\varepsilon \in [0, 1]$, linearity of kernel mean embeddings gives

$$\mathrm{MMD}_k(\mathbb{P}, (1 - \varepsilon)\mathbb{P} + \varepsilon R) = \varepsilon\|\mu_R - \mu_\mathbb{P}\|_\mathcal{H} \leq \varepsilon(\|\mu_R\|_\mathcal{H} + \|\mu_\mathbb{P}\|_\mathcal{H}) \leq 2\kappa\varepsilon.$$

Thus every Huber contamination $(1 - \varepsilon)\mathbb{P} + \varepsilon R$ lies in an MMD ball of radius $2\kappa\varepsilon$, and in particular

$$\mathcal{B}_{\mathrm{LV}}^\varepsilon(\mathbb{P}) \subseteq \mathcal{B}_{\mathrm{MMD}}^{2\kappa\varepsilon}(\mathbb{P}).$$

When forward Huber contamination yields an infinite worst-case risk, the corresponding MMD worst-case risk is also infinite for any radius that contains such contaminations.

In MMD-DRO, we usually analyse an upper bound on the worst-case risk, which involves an RKHS-norm penalty. See, for example, Staib & Jegelka (2019, Corollary 3.2):

$$\sup_{Q:\text{MMD}_k(Q,\mathbb{P})\leq\rho} \mathbb{E}_Q[f_x] \leq \mathbb{E}_{\mathbb{P}}[f_x] + \rho\|f_x\|_{\mathcal{H}}.$$

When $\kappa^2 = \sup_\xi k(\xi,\xi) < \infty$, every $f \in \mathcal{H}$ is bounded:

$$|f(\xi)| = |\langle f, k(\xi,\cdot)\rangle_{\mathcal{H}}| \leq \|f\|_{\mathcal{H}}\|k(\xi,\cdot)\|_{\mathcal{H}} \leq \kappa\|f\|_{\mathcal{H}}.$$

Hence, if $f_x$ is unbounded above on $\Xi$, then $f_x \notin \mathcal{H}$ and the RKHS-penalised bound is not applicable (formally $\|f_x\|_{\mathcal{H}} = +\infty$).

Finally, Kernel DRO implementations may approximate worst-case distributions by restricting to discrete distributions supported on sampled candidates $\{\hat{\xi}^1, \ldots, \hat{\xi}^{S_\Omega}\}$; see, e.g., the support-sampling procedure in Zhu et al. (2021, Appendix C.4–C.5). Outside the compact-$\Xi$ and bounded-loss setting typically assumed for kernel duality arguments, such finite-support approximations can be driven by increasingly extreme samples as $S_\Omega$ grows, similar to the scenario-max instability in (41).

## F. Synthetic Heavy-tailed Newsvendor (Section 5.1): Additional Details and Results

This appendix provides additional details (Sections F.1–F.5) and results (Sections F.6–F.8) for the synthetic heavy-tailed newsvendor experiment in Section 5.1.

Section F.1 describes the demand distribution, bulk-set calibration split, and test distribution; Section F.2 describes the Bayesian Student-$t$ model and posterior predictive sampling procedure; in Section F.3, we specify the tolerance grids; Section F.4 lists the optimisation objectives used by each method; Section F.5 lists the software, solver, hardware, and resource settings.

Section F.6 gives the OR-WDRO parameterisation and the additional Wasserstein-radius sweep; Section F.7 gives the SAA-budget sensitivity study for the Bayesian methods; finally, Section F.8 gives the sensitivity study for the bulk-set tail-budget parameter $\gamma$.

### F.1. Data-generating Process and Spike Contamination

We consider $d = 5$ products. Clean demand vectors are sampled i.i.d. as

$$\xi \sim t_\nu(\mu, \Sigma), \qquad \nu = 3, \qquad \mu = 30\,\mathbf{1}_d,$$

with correlated scale matrix $\Sigma$ constructed as $\Sigma = DRD$, where $D = \text{diag}(s_1, \ldots, s_d)$ and $s_j = 10\,(1 + 0.1(j-1))$. We use a Toeplitz correlation matrix $R$ with $(R)_{ij} = \rho^{|i-j|}$ and $\rho = 0.6$. We generate $n_{\text{train}} = 2000$ training samples and $n_{\text{test}} = 500$ test samples, and repeat the experiment over 100 independent replications. The bulk-set calibration process randomly splits the training data into $|\mathcal{D}_{\text{fit}}| = 1000$ samples for score fitting and $|\mathcal{D}_{\text{select}}| = 1000$ samples for score selection.

To model promotion-like shocks, we apply spike contamination to the test data:

$$\tilde{\xi} = \begin{cases} \xi^{\text{spike}}, & \text{with probability } \varepsilon_{\text{cont}}, \\ \xi, & \text{with probability } 1 - \varepsilon_{\text{cont}}, \end{cases} \qquad \varepsilon_{\text{cont}} \in \{0, 0.1, 0.2\},$$

where $\xi \sim t_\nu(\mu, \Sigma)$ is a clean draw and $\xi^{\text{spike}}$ is drawn from a high-demand Gaussian component

$$\xi^{\text{spike}} \sim \mathcal{N}(\mu_{\text{spike}}, \Sigma_{\text{spike}}).$$

We set $\mu_{\text{spike}} = \mu + 6\sigma$ and $\sigma_j = \sqrt{\Sigma_{jj}}$, and $\Sigma_{\text{spike}} = 0.05\Sigma$. This produces a small proportion of test points exhibiting an across-the-board surge in demand, mimicking a short-lived promotion regime.

### F.2. Bayesian Model and SAA

We place a normal-inverse-Wishart (NIW) prior on the unknown mean and scale matrix in the multivariate Student-$t$ likelihood (with fixed $\nu = 3$). Since this Student-$t$ likelihood is not conjugate to the NIW prior, we approximate the

posterior and posterior predictive by Gibbs sampling. We use the standard scale-mixture representation with latent Gamma weights. After burn-in, each retained Gibbs iterate $(\mu, \Sigma)$ is followed by a draw $\xi \sim t_\nu(\mu, \Sigma)$, which provides posterior predictives for SAA.[4] All Bayesian methods (LV, KL-BDRO, and KL-BAS$_{\mathrm{PP}}$) share this posterior predictive; KL-Empirical and OR-WDRO use the empirical distribution. We use the following sampling budgets:

- KL-BAS$_{\mathrm{PP}}$: $M_{\mathrm{pred}} = 2500$ predictive samples.

- LV: $M_{\mathrm{pred}} = 2500$ predictive samples.

- KL-BDRO: $M_{\mathrm{post}} = 50$ posterior draws and $M_{\mathrm{pred}} = 50$ predictive samples per draw.

### F.3. Tolerance Sweep

We sweep tolerances on two independent grids. For LV we use 24 $\varepsilon_{\mathrm{LV}}$ values in $(0, 1]$, and OR-WDRO uses the same set but truncated at 0.5, which gives 21 values. For KL-based methods we sweep KL radii $\varepsilon_{\mathrm{KL}}$ separately in $(0, 25]$, with 24 values in total. In the MSD plots, we display the inverse mapping on a secondary axis to share a single x-axis scale:

$$\varepsilon_{\mathrm{KL}} = \tfrac{1}{2}\left(t\varepsilon_{\mathrm{LV}}\right)^2, \qquad t := \sqrt{50}.$$

The quadratic scaling matches the Gaussian penalty-matching heuristic, but here $t$ is chosen purely for visual alignment (so that $\varepsilon_{\mathrm{LV}} = 1$ corresponds to $\varepsilon_{\mathrm{KL}} = 25$).

### F.4. Optimisation Objectives

**KL-Empirical.** For a KL radius $\varepsilon_{\mathrm{KL}} \geq 0$, KL-Empirical solves the forward-KL divergence-ball DRO problem centred at the empirical distribution (Hu & Hong, 2013):

$$\min_{x \in \mathbb{R}^d_+} \ \inf_{\lambda > 0} \left\{ \lambda \varepsilon_{\mathrm{KL}} + \lambda \log \left( \frac{1}{n} \sum_{i=1}^n \exp\left( \frac{f_x(\xi_i)}{\lambda} \right) \right) \right\}. \tag{43}$$

**KL-BAS$_{\mathrm{PP}}$.** KL-BAS$_{\mathrm{PP}}$ uses the same KL objective as (43), but centres the KL ball at the Bayesian posterior predictive $\mathbb{P}_c$ and approximates the log-MGF using posterior predictive scenarios $\tilde{\xi}_1, \ldots, \tilde{\xi}_{M_{\mathrm{pred}}} \sim \mathbb{P}_c$. So its objective is (43) with $n$ replaced by $M_{\mathrm{pred}}$ and $\{\xi_i\}_{i=1}^n$ replaced by $\{\tilde{\xi}_j\}_{j=1}^{M_{\mathrm{pred}}}$.

**KL-BDRO.** KL-BDRO averages the KL-robust objective across posterior draws. Concretely, for posterior samples $\theta^{(1)}, \ldots, \theta^{(M_{\mathrm{post}})}$ (Student-$t$ parameters) and predictive scenarios $\tilde{\xi}_{i,1}, \ldots, \tilde{\xi}_{i,M_{\mathrm{pred}}} \sim \mathbb{P}(\cdot \mid \theta^{(i)})$, KL-BDRO solves

$$\min_{x \in \mathbb{R}^d_+} \ \frac{1}{M_{\mathrm{post}}} \sum_{i=1}^{M_{\mathrm{post}}} \inf_{\lambda_i > 0} \left\{ \lambda_i \varepsilon_{\mathrm{KL}} + \lambda_i \log \left( \frac{1}{M_{\mathrm{pred}}} \sum_{m=1}^{M_{\mathrm{pred}}} \exp\left( \frac{f_x(\tilde{\xi}_{i,m})}{\lambda_i} \right) \right) \right\}. \tag{44}$$

*Remark* F.1 (Saturation of KL-BDRO at large KL radii). In (44), each inner KL ball is centred at the uniform empirical measure on the $M_{\mathrm{pred}}$ predictive scenarios, $\hat{\mathbb{P}}_i := \frac{1}{M_{\mathrm{pred}}} \sum_{m=1}^{M_{\mathrm{pred}}} \delta_{\tilde{\xi}_{i,m}}$. Since $\mathrm{KL}(Q\|\hat{\mathbb{P}}_i) < \infty$ forces $Q$ to be supported on $\{\tilde{\xi}_{i,m}\}_{m=1}^{M_{\mathrm{pred}}}$, once $\varepsilon_{\mathrm{KL}} \geq \log M_{\mathrm{pred}}$ the KL ball contains a Dirac mass on any scenario (because $\mathrm{KL}(\delta_{\tilde{\xi}_{i,m}} \| \hat{\mathbb{P}}_i) = \log M_{\mathrm{pred}}$). This means that for all such radii, the worst-case expectation inside (44) reduces to $\max_{m \leq M_{\mathrm{pred}}} f_x(\tilde{\xi}_{i,m})$, and the KL-BDRO objective (and its minimiser) no longer changes with $\varepsilon_{\mathrm{KL}}$ beyond this threshold (in Figure 3 $\log M_{\mathrm{pred}} = \log 50 \approx 3.91$, after which the OOS performance stays constant for KL-BDRO). This relationship also follows straightforwardly from the Shannon entropy–KL relationship: since $\hat{\mathbb{P}}_i$ is uniform, $\mathrm{KL}(Q\|\hat{\mathbb{P}}_i) = \log M_{\mathrm{pred}} - H(Q) \leq \log M_{\mathrm{pred}}$, where $H(Q)$ is the Shannon entropy of $Q$ (Cover & Thomas, 2006, Theorem 2.7.3). The same effect exists for KL-BAS$_{\mathrm{PP}}$ and KL-Empirical, but they have larger saturation limits ($\log 2500 \approx 7.82$ and $\log 2000 \approx 7.60$, respectively) and demonstrate a wider range of OOS mean–variance trade-offs.

---

[4]We use 200 burn-in and no thinning. For numerical stability, we add a ridge term $10^{-6} I$ to the scale matrix.

**LV.** The ellipsoidal bulk set is

$$\Xi_0 := \left\{ \xi \in \mathbb{R}^d : \|L^{-1}(\xi - \hat{\mu})\|_2 \leq \hat{t}_{\mathrm{DKW}} \right\}, \qquad \hat{\Sigma} := LL^{\top}, \tag{45}$$

where $(\hat{\mu}, \hat{\Sigma})$ are fitted on $\mathcal{D}_{\mathrm{fit}}$ and $\hat{t}_{\mathrm{DKW}}$ chosen with $\mathcal{D}_{\mathrm{select}}$ for $\gamma = \delta = 0.05$. For a tolerance $\varepsilon_{\mathrm{LV}} \in [0, 1]$, LV solves the objective

$$\min_{x \in \mathbb{R}^d_+} (1 - \varepsilon_{\mathrm{LV}}) \mathbb{E}_{\mathbb{P}_{c,\Xi_0}} [f_x(\xi)] + \varepsilon_{\mathrm{LV}} \sup_{\xi \in \Xi_0} f_x(\xi), \tag{46}$$

where the truncated expectation $\mathbb{E}_{\mathbb{P}_{c,\Xi_0}} [f_x(\xi)] = \mathbb{E}_{\mathbb{P}_c} [f_x(\xi) | \xi \in \Xi_0]$ is approximated by an SAA over samples from $\mathbb{P}_c(\cdot \mid \Xi_0)$ (via rejection sampling). For fixed $x$, the newsvendor loss is piecewise-linear in $\xi$:

$$f_x(\xi) = \sum_{j=1}^d \max\{-h(\xi_j - x_j), b(\xi_j - x_j)\} = \max_{a \in \{-h,b\}^d} \{a^{\top}\xi - a^{\top}x\}.$$

Therefore, the ellipsoid entry for piecewise-linear losses in Table 1 applies directly with $(\mu_{\mathrm{fit}}, \Sigma_{\mathrm{fit}}, t_k) = (\hat{\mu}, \hat{\Sigma}, \hat{t}_{\mathrm{DKW}})$ and $(a_{x,j}, b_{x,j}) = (a, -a^{\top}x)$, yielding

$$\sup_{\xi \in \Xi_0} f_x(\xi) = \max_{a \in \{-h,b\}^d} \left\{ a^{\top}(\hat{\mu} - x) + \hat{t}_{\mathrm{DKW}} \|L^{\top}a\|_2 \right\}. \tag{47}$$

**OR-WDRO.** We implement the outlier-robust Wasserstein DRO baseline of Nietert et al. (2023). The optimisation target is solved via the convex conic reformulation in their Theorem 5.

### F.5. Computing and Implementation Details

We implement all optimisation problems in Python 3.11 using CVXPY version 1.7.3 (Diamond & Boyd, 2016; Agrawal et al., 2018) and solve the resulting conic programs with the MOSEK solver version 11.0.29 (MOSEK ApS, 2025). Experiments are scheduled with SLURM and run on CPU nodes equipped with AMD EPYC processors (64 cores / 128 threads) and 384 GB RAM. Unless stated otherwise, each run used a single CPU core (one thread) and 10 GB of memory.

### F.6. OR-WDRO Parameterisation

OR-WDRO has two hyperparameters: a Wasserstein radius $\rho$ and an outlier fraction $\varepsilon_{\mathrm{OR}}$. In the main experiments we fix $\rho = 5$ and sweep $\varepsilon_{\mathrm{OR}} \in [0, 0.5]$. As a sensitivity check, for the 20% OOS contamination experiment, we fix $\varepsilon_{\mathrm{OR}} = 0.2$ as the nominal contamination level and sweep $\rho \in [0, 50]$; the resulting OOS performance, reported in Figure 7, varies only modestly over the sweep.

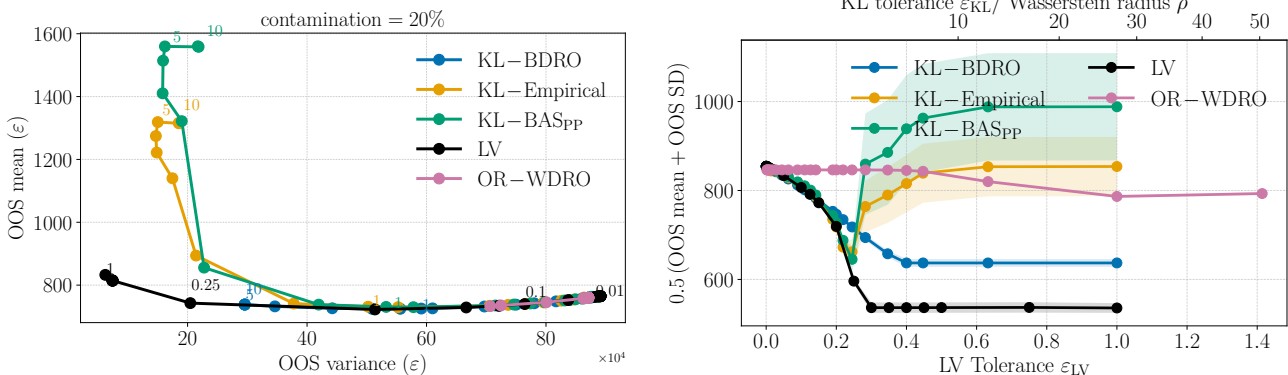

*Figure 7.* Additional sensitivity plots for OR-WDRO in the synthetic newsvendor experiment with 20% contamination. Left: OOS mean–variance frontier at 20% contamination. Right: MSD $= \frac{1}{2}$(OOS mean + OOS SD) versus the tolerance level (top axis shows the corresponding Wasserstein radius $\rho$ as plotted).

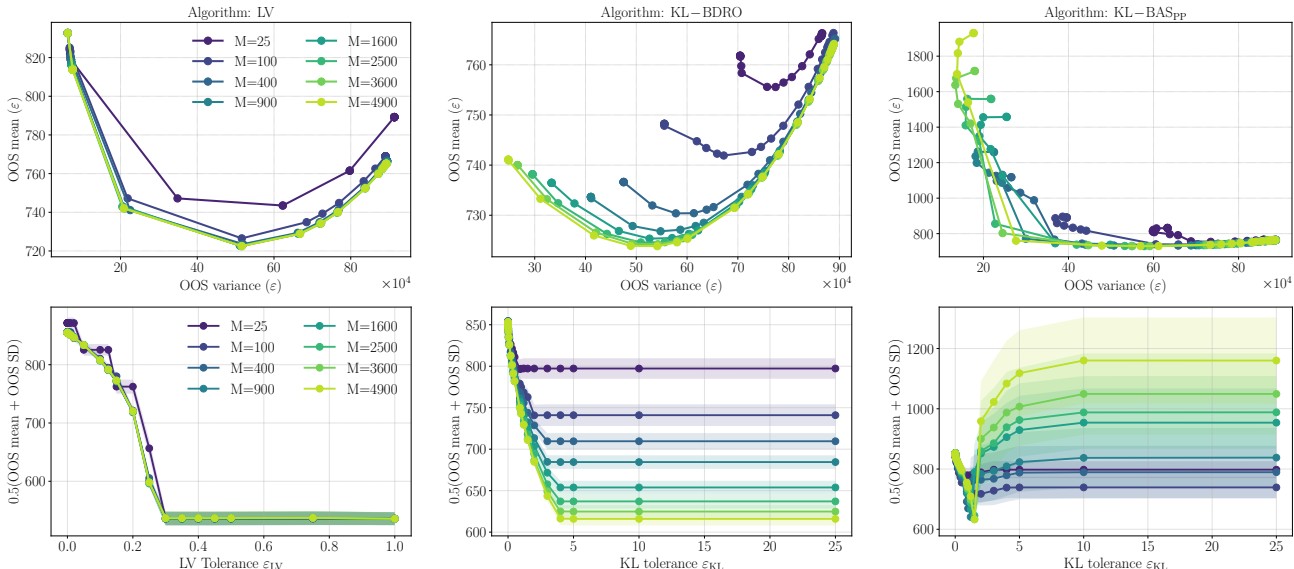

*Figure 8.* Sample-efficiency diagnostics for Student-$t$ newsvendor under $20\%$ spike contamination. Left column: LV; middle column: KL-BDRO; right column: KL-BAS$_{\text{PP}}$. Top row: OOS mean–variance frontiers for different total model-sample budgets $M = M_{\text{post}} M_{\text{pred}}$. Bottom row: MSD $= \frac{1}{2}(\mu + \sigma)$ versus tolerance for the same budgets.

### F.7. Sensitivity to SAA Budgets: Sample Efficiency of Bayesian Methods

In the Student-$t$ newsvendor experiment at contamination level $\varepsilon_{\text{cont}} = 0.2$, we study how sensitive the SAA in each Bayesian method (LV, KL-BDRO, and KL-BAS$_{\text{PP}}$) is to the Monte Carlo budget.

We vary the total model-sample budget

$$M := M_{\text{post}} \times M_{\text{pred}} \in \{25, 100, 400, 900, 1600, 2500, 3600, 4900\},$$

holding fixed all other experimental settings. For LV and KL-BAS$_{\text{PP}}$ we only need posterior predictive samples: $M_{\text{pred}} = M$. For KL-BDRO we use a balanced nested allocation $M_{\text{post}} = M_{\text{pred}} = \sqrt{M}$, so that the total number of model samples is comparable across methods.

In LV we estimate a truncated posterior predictive expectation by drawing $M_{\text{pred}}$ predictive samples and keeping only those that fall inside the calibrated bulk set. The number of accepted samples is therefore random and is not known before sampling. Since `CVXPY` requires a fixed problem size, we build the optimisation model with a fixed scenario count and set it to $0.5 M_{\text{pred}}$, which is a conservative lower bound on the expected acceptance rate in our settings. We take the first $0.5 M_{\text{pred}}$ accepted samples to approximate the optimisation target. The sample-efficiency results below show that LV already reaches its performance plateau at small $M_{\text{post}} M_{\text{pred}}$, so using this conservative fixed size does not change the qualitative conclusions.

Figure 8 shows, for each method, (top row) the OOS mean–variance frontiers traced out as the tolerance varies, and (bottom row) the corresponding MSD $= \frac{1}{2}(\mu + \sigma)$ curves versus tolerance, with one curve per budget $M$. To quantify stability of the entire MSD curve, Figure 9 reports the mean absolute deviation of the MSD curve relative to the highest-budget run ($M_{\text{max}} = 4900$), averaged over the tolerance grid:

$$\Delta_{\text{MSD}}(M) := \frac{1}{|\mathcal{E}|} \sum_{\varepsilon \in \mathcal{E}} \left| \widehat{\text{MSD}}_M(\varepsilon) - \widehat{\text{MSD}}_{M_{\text{max}}}(\varepsilon) \right|,$$

where $\widehat{\text{MSD}}_M(\varepsilon)$ denotes the replication-averaged MSD for tolerance $\varepsilon$ under budget $M$.

We observe that LV is highly sample efficient: across all budgets shown, the mean–variance frontiers and MSD curves in Figure 8 (left column) are visually indistinguishable. This is reflected quantitatively in Figure 9: even at $M = 100$, the

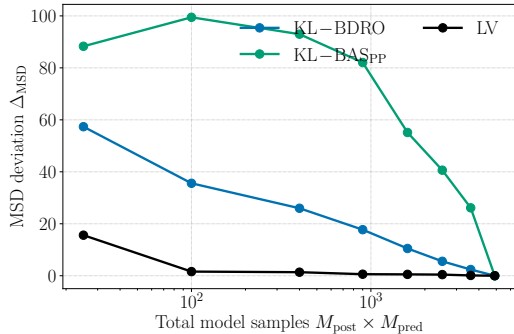

*Figure 9.* Curve-level stability versus total samples. Mean absolute deviation $\Delta_{\text{MSD}}(M)$ of the MSD curve from the highest-budget run ($M_{\max} = 4900$), averaged over the tolerance grid $\mathcal{E}$, for $\varepsilon_{\text{cont}} = 0.2$. Lower values indicate higher sampling efficiency (greater stability under smaller Monte Carlo budgets).

average deviation from the $M = 4900$ curve is negligible. This is because LV uses SAA only to approximate a truncated predictive expectation, which is easy to estimate and inherently stable. This is confirmed by the SAA convergence diagnostic for the truncated expectation reported in Figure 10.

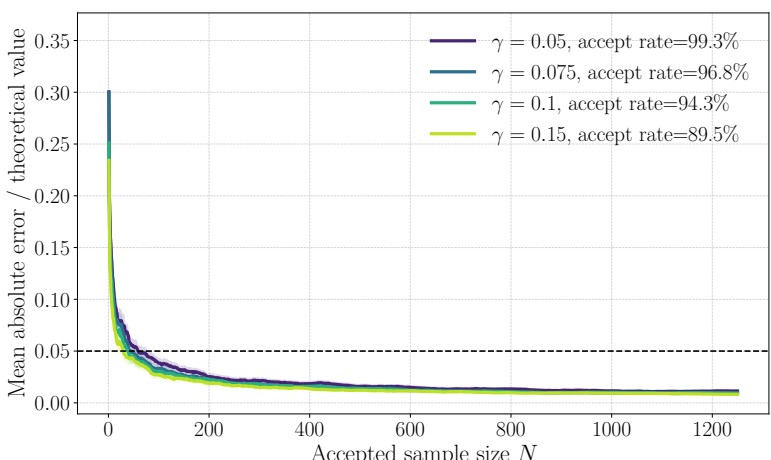

*Figure 10.* Convergence of the rejection sampling SAA to the truncated expectation. For each $\gamma$, the curve plots the mean absolute error of the running SAA mean, normalised by the theoretical truncated expectation, as a function of the accepted sample size $N$. Shaded bands show pointwise 95% confidence intervals over 100 independent replications. The legend reports the corresponding rejection-sampling accept rate for each $\gamma$. The horizontal dashed line marks the 5% relative-error level.

On the contrary, KL-BDRO needs substantially larger budgets to stabilise. Figure 8 (middle column) shows a clear separation between small budgets and larger ones. This is confirmed by Figure 9. KL-BDRO uses a nested Monte Carlo approximation in which posterior uncertainty and predictive uncertainty both enter the approximation of the robust objective. This double approximation is inherently more sample-hungry than the single-loop estimation required by LV. The SAA budget also affects the saturation limit as discussed in Remark F.1. With a small $M$, KL-BDRO can only explore a small trade-off region.

Unlike LV and KL-BDRO, increasing $M$ does not simply reduce Monte Carlo noise for KL-BAS$_{\text{PP}}$. In Figure 8 (right column), the large-tolerance regime becomes *worse* at larger budgets: the frontier develops very low-variance but high-mean (high-cost) points, and the MSD curve increases sharply for larger tolerances at large $M$. KL-BAS$_{\text{PP}}$ admits the classical KL dual form

$$\sup_{Q:\text{KL}(Q\|\mathbb{P})\leq\rho} \mathbb{E}_{\xi\sim Q}[f_x(\xi)] = \inf_{\eta>0} \left\{\eta\rho + \eta\log\mathbb{E}_{\xi\sim\mathbb{P}}\big[\exp\big(f_x(\xi)/\eta\big)\big]\right\},$$

so the Monte Carlo approximation must estimate the log-MGF of the loss under $\mathbb{P}$ via a log-sum-exp. This is well known to be dominated by rare extreme samples because of the exponential weighting. In our Student-$t$ setting, the predictive distribution is heavy-tailed while the newsvendor loss is unbounded and grows (at least) linearly in demand; this combination makes exponential-moment estimation particularly brittle. As $M$ increases, the probability of drawing extreme demands increases, and these rare draws can inflate the estimated log-MGF, pushing the optimiser towards overly conservative decisions at large tolerances.

### F.8. $\gamma$-Sensitivity Analysis

The DKW selection step requires hyperparameters $\delta$ and $\gamma$, and $\delta$ is fixed at a required significance level (e.g., 0.01 or 0.05). In this section, we perform a sensitivity analysis on the tail-budget parameter $\gamma$, which is fixed at 0.05 in the main newsvendor experiment. The minimum certifiable $\gamma$ for this experiment, given our selection-set size, is 0.043, and we studied $\gamma$ between 0.043 and 0.15. The results are displayed in Figure 11.

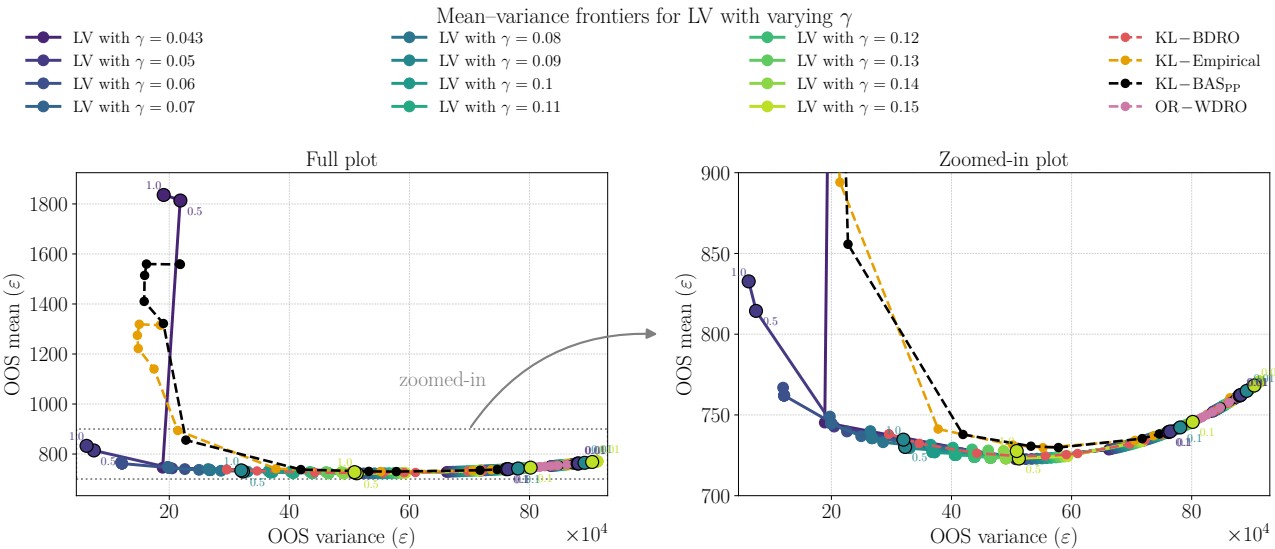

*Figure 11.* Synthetic newsvendor (cost: lower-left is better) $\gamma$-sensitivity analysis for LV under 20% contamination. Each LV curve sweeps $\varepsilon_{\mathrm{LV}}$ for a fixed $\gamma$; selected $\varepsilon_{\mathrm{LV}}$ values are marked for $\gamma \in \{0.043, 0.05, 0.09, 0.15\}$. Dashed curves are the KL and Wasserstein baselines. Left: full OOS mean–variance frontiers. Right: zoomed-in view of the same frontiers.

For $\gamma \in [0.05, 0.10)$, the frontiers are similar and stay in the same favourable trade-off region relative to the baselines. Some unstable behaviour appears at the minimum certifiable value 0.043, which means that the DKW threshold is the single largest selection score. At large $\varepsilon$, the optimisation can be overly driven by that point. Moving from 0.043 to 0.05 is sufficient to remove this effect. For larger $\gamma \geq 0.10$, the bulk becomes too restrictive, and the supremum term becomes less informative, so LV can plateau in a low-mean, mid-variance region, similar to that of KL-BDRO.

Therefore, when the selection sample size is approximately of order $10^3$, we recommend choosing $\gamma$ just above the minimum certifiable value to keep the supremum term effective while keeping a tail of around 10 selection scores to ensure stability. When the selection sample size is large ($> 10^5$) and the minimum certifiable $\gamma < 10^{-2}$, we recommend choosing $\gamma \in [0.01, 0.05]$ to avoid emphasising overly rare scenarios.

## G. California Housing Regression (Section 5.2): Additional Details and Results

We use the publicly available California housing dataset (Pace & Barry, 1997). We use the same optimisation solver and computing setup as in the synthetic newsvendor experiment (Appendix F.5). Throughout Appendix G, a California housing example is denoted by $\xi = (X, Y)$, where $X \in \mathbb{R}^d$ with $d = 8$ is the covariate vector and $Y \in \mathbb{R}$ is the housing-price outcome; the decision variables are the linear predictor slope and intercept $(w, b)$, and the per-example absolute-error loss is $f_{w,b}(\xi) := |Y - w^\top X - b|$. In addition to the baselines reported in Section 5.2, this appendix includes two $f$-divergence

baselines, KL-DRO and $\chi^2$-DRO, which are implemented using the Python DRO package (Liu et al., 2025).

Subsection G.1 describes the East–West deployment split, gap ratios, and evaluation metrics; Section G.2 describes the frequentist Gaussian copula centre $\mathbb{P}_c$; Section G.3 describes geo-block cross-validation; Section G.4 lists the optimisation objectives for LV and each baseline.

Section G.5 gives the CV-selected results and runtime comparison with additional $f$-DRO baselines; Section G.6 provides hyperparameter sweep trajectories for different gap ratios and oracle selection results; finally, Section G.7 discusses the choice of the bulk set $\Xi_{0,X} \times \Xi_{0,Y}$ in regression.

### G.1. Geographic Split, Gap Construction, and Evaluation

We use longitude to define an East–West deployment shift. We sort the data by longitude and take the West as the deployment (test) region. For a gap ratio $g \in \{0\%, 10\%, 20\%, 30\%\}$, we assign the Eastern 50% to the training region, exclude a contiguous band of width $g$ on the train–test boundary, and assign the remaining Western $(50\% - g)$ to the test region. For the tables in Section 5.2, we use $g = 30\%$.

We standardise features using the training split only, and apply the same transform to all other splits. The outcome (housing price) is not standardised. We report all metrics in the original target scale (US dollars). We evaluate MAE, RMSE, $p_{98}(|y - \hat{y}|)$, and $\mathrm{CVaR}_{2\%}(|y - \hat{y}|)$ on the West test region.

### G.2. Frequentist Gaussian Copula Centre

In the California housing experiment we instantiate the centre distribution $\mathbb{P}_c$ as a simple semiparametric plug-in law for $\xi = (X, Y)$ fitted on the East training split. Specifically, we model the covariates $X \in \mathbb{R}^d$ using a Gaussian copula with empirical marginals, and we model $Y \mid X$ using a homoscedastic Gaussian linear regression.

For each coordinate $j \in \{1, \ldots, d\}$, let $\widehat{F}_j$ be the empirical CDF of $X_j$ on $\mathcal{D}_{\text{train}}$, and define pseudo-observations $U_{ij} := \widehat{F}_j(X_{ij}) \in (0, 1)$. Let $Z_{ij} := \Phi^{-1}(U_{ij})$, where $\Phi$ is the standard normal CDF, and estimate the latent Gaussian covariance by

$$\widehat{\Sigma} := \mathrm{Cov}(Z) + \eta I_d$$

for a small jitter $\eta > 0$. To sample $X \sim \mathbb{P}_c$, draw $Z \sim \mathcal{N}(0, \widehat{\Sigma})$, set $U = \Phi(Z)$, and apply the coordinatewise inverse CDF $X_j = \widehat{F}_j^{-1}(U_j)$.

We fit an OLS predictor $\hat{y}(X) = \widehat{w}_{\text{OLS}}^\top X + \widehat{b}_{\text{OLS}}$ on $\mathcal{D}_{\text{train}}$, and set $\widehat{\sigma}_Y^2$ to the training mean squared residual. Conditional on a sampled $X$, we then draw

$$Y = \widehat{w}_{\text{OLS}}^\top X + \widehat{b}_{\text{OLS}} + \widehat{\sigma}_Y \varepsilon, \qquad \varepsilon \sim \mathcal{N}(0, 1),$$

which defines the joint sampling distribution $\mathbb{P}_c$ on $\xi = (X, Y)$.

### G.3. Geo-block Cross-validation

We tune hyperparameters using geo-block CV within the East region only. Blocks are formed by binning longitude and latitude into a $6 \times 6$ grid and discarding empty bins, yielding 34 non-empty geographic blocks in this split. We use 3 folds over blocks, and select hyperparameters by minimising the mean validation MAE across folds. We use fixed grids for each method:

- LV and CVaR: $\varepsilon \in \{0.025, 0.05, 0.075, 0.10, 0.125, 0.15, 0.175, 0.20\}$.

- Wasserstein: $\rho \in \{0.05, 0.10, 0.50, 1.0, 1.5, 2.0, 2.5, 3.0\}$.

- KL-DRO: $\varepsilon_{\text{KL}} \in \{0.0, 0.01, 0.02, 0.05, 0.10, 0.50, 1.0, 5.0\}$.

- $\chi^2$-DRO: $\varepsilon_{\chi^2} \in \{0.0, 0.05, 0.10, 0.20, 0.50, 1.0, 5.0, 10.0\}$.

- Ridge: $\lambda \in \{10^{-2}, 10^{-1}, 0.5, 1, 5, 10, 50, 100\}$.

### G.4. Optimisation Objectives

Let $\{\xi_i = (X_i, Y_i)\}_{i=1}^n$ denote the (East) training set used to fit a linear predictor $\hat{y}(X) = w^\top X + b$ with $w \in \mathbb{R}^d$ and $b \in \mathbb{R}$. Write the absolute residual loss as $f_{w,b}(\xi) := |Y - w^\top X - b|$. Below we list the optimisation problems solved by each method.

**ERM.**

$$\min_{w,b} \frac{1}{n} \sum_{i=1}^n f_{w,b}(\xi_i). \tag{48}$$

**CVaR.** For a tail mass parameter $\varepsilon \in (0, 1)$, we minimise the empirical $\mathrm{CVaR}_{1-\varepsilon}$ of the absolute residual:

$$\min_{w,b,\tau \in \mathbb{R}} \tau + \frac{1}{\varepsilon n} \sum_{i=1}^n \left( f_{w,b}(\xi_i) - \tau \right)_+. \tag{49}$$

**Ridge.** For $\lambda \geq 0$, we solve $\ell_2$-regularised least squares (intercept unpenalised):

$$\min_{w,b} \frac{1}{n} \sum_{i=1}^n \left( Y_i - w^\top X_i - b \right)^2 + \lambda \|w\|_2^2. \tag{50}$$

**Wasserstein.** For $\rho \geq 0$, we use a joint $(X, Y)$ 1-Wasserstein DRO baseline that allows transport in both covariates and labels. We equip the data space with the weighted Euclidean ground cost

$$c\big((X,Y),(X',Y')\big) := \sqrt{\|X - X'\|_2^2 + \kappa^2(Y - Y')^2},$$

where $\kappa > 0$ controls the relative cost of transporting the label. Since we do not standardise $Y$ in this experiment, we set $\kappa := 1/\sigma_Y$ with $\sigma_Y$ the empirical SD of the training responses, so that label transport is measured in units of one training SD (Chen & Paschalidis, 2018). For the LAD loss $f_{w,b}(\xi) = |Y - w^\top X - b|$, the corresponding dual form yields the convex objective

$$\min_{w,b} \frac{1}{n} \sum_{i=1}^n f_{w,b}(\xi_i) + \rho \, \|(w, 1/\kappa)\|_2 = \min_{w,b} \frac{1}{n} \sum_{i=1}^n f_{w,b}(\xi_i) + \rho \, \|(w, \sigma_Y)\|_2, \tag{51}$$

where the intercept $b$ is not regularised.

**KL.** For a KL radius $\varepsilon_{\mathrm{KL}} \geq 0$, the KL-DRO objective is

$$\min_{w,b} \inf_{\lambda > 0} \left\{ \lambda \varepsilon_{\mathrm{KL}} + \lambda \log \left( \frac{1}{n} \sum_{i=1}^n \exp \left( \frac{f_{w,b}(\xi_i)}{\lambda} \right) \right) \right\}. \tag{52}$$

**$\chi^2$.** For a $\chi^2$ radius $\varepsilon_{\chi^2} \geq 0$, by Duchi & Namkoong (2021, Lemma 1) we solve

$$\min_{w,b} \inf_{\lambda \in \mathbb{R}} \left\{ \lambda + \sqrt{(1 + 2\varepsilon_{\chi^2}) \left( \frac{1}{n} \sum_{i=1}^n \left( f_{w,b}(\xi_i) - \lambda \right)_+^2 \right)} \right\}. \tag{53}$$

**LV.** We use the intersection bulk set $\Xi_0 = \Xi_{0,X} \times \Xi_{0,Y}$ with $\gamma = 0.1$ and $\delta = 0.05$, and with the $X$-score and $Y$-score thresholds at levels $(\gamma/2, \delta/2)$ each. The DKW selection process provides the following:

$$\Xi_{0,X} := \{X \in \mathbb{R}^d : \|L_X^{-1}(X - \mu_X)\|_2 \leq t_X\}, \qquad \Xi_{0,Y} := \{Y \in \mathbb{R} : |Y - \mu_Y| \leq r_Y\}, \tag{54}$$

where $\mu_X \in \mathbb{R}^d$, $\mu_Y \in \mathbb{R}$, $t_X \geq 0$, $r_Y \geq 0$, and $L_X \in \mathbb{R}^{d \times d}$ is an invertible square-root factor (so $\Sigma_{XX} = L_X L_X^\top$). We use a random split of $80\%$ of the training data for score-fitting and $20\%$ for score selection. Although we standardise $X$ with the whole training dataset, the Mahalanobis score is invariant to such affine reparameterisations, so the DKW selection

procedure is unaffected (up to a $10^{-8}$ ridge for numerical stability in the Cholesky decomposition). Given a fitted centre distribution $\mathbb{P}_c$ on $(X, Y)$, LV solves the population objective

$$\min_{w,b}(1 - \varepsilon)\mathbb{E}_{\xi \sim \mathbb{P}_{c,\Xi_0}}\left[f_{w,b}(\xi)\right] + \varepsilon \sup_{\xi \in \Xi_0} f_{w,b}(\xi), \tag{55}$$

with the truncated expectation $\mathbb{E}_{\xi \sim \mathbb{P}_{c,\Xi_0}}\left[f_{w,b}(\xi)\right] = \mathbb{E}_{\mathbb{P}_c}\left[f_{w,b}(\xi) \mid \xi \in \Xi_0\right]$, approximated by SAA via rejection sampling from $\mathbb{P}_c(\cdot \mid \Xi_0)$. The next lemma gives the closed form for the worst-case term in (55) under the $\Xi_{0,X} \times \Xi_{0,Y}$ geometry.

**Lemma G.1** (Worst-case absolute loss on $\Xi_{0,X} \times \Xi_{0,Y}$). *Let $\Xi_0 = \Xi_{0,X} \times \Xi_{0,Y}$ be defined as in (54). Then for any $w \in \mathbb{R}^d$ and $b \in \mathbb{R}$,*

$$\sup_{\xi \in \Xi_0} f_{w,b}(\xi) = \left|\mu_Y - \mu_X^\top w - b\right| + r_Y + t_X\|L_X^\top w\|_2. \tag{56}$$

*Proof.* For $\xi = (X, Y)$, write

$$f_{w,b}(\xi) = |Y - w^\top X - b| = \max\{Y - w^\top X - b, -Y + w^\top X + b\}.$$

Applying (19) with $J = 2$ and $S = \Xi_0 = \Xi_{0,X} \times \Xi_{0,Y}$ gives

$$\sup_{\xi \in \Xi_0} f_{w,b}(\xi) = \max\left\{\sup_{(X,Y) \in \Xi_0}(Y - w^\top X - b),\ \sup_{(X,Y) \in \Xi_0}(-Y + w^\top X + b)\right\}.$$

For the first term, since $\Xi_0$ is a Cartesian product and the objective is affine and separable in $(X, Y)$,

$$\sup_{(X,Y) \in \Xi_0}(Y - w^\top X - b) = -b + \sup_{Y \in \Xi_{0,Y}} Y + \sup_{X \in \Xi_{0,X}}(-w)^\top X.$$

Because $\Xi_{0,Y} = \{\mu_Y + s : |s| \le r_Y\}$, we have $\sup_{Y \in \Xi_{0,Y}} Y = \mu_Y + r_Y$. Moreover, by the ellipsoid calculation in Lemma B.4 (the linear case $J = 1$),

$$\sup_{X \in \Xi_{0,X}}(-w)^\top X = (-w)^\top \mu_X + t_X\|L_X^\top(-w)\|_2 = -\mu_X^\top w + t_X\|L_X^\top w\|_2.$$

Thus

$$\sup_{(X,Y) \in \Xi_0}(Y - w^\top X - b) = (\mu_Y - \mu_X^\top w - b) + r_Y + t_X\|L_X^\top w\|_2.$$

Similarly,

$$\sup_{(X,Y) \in \Xi_0}(-Y + w^\top X + b) = b + \sup_{X \in \Xi_{0,X}} w^\top X + \sup_{Y \in \Xi_{0,Y}}(-Y) = -(\mu_Y - \mu_X^\top w - b) + r_Y + t_X\|L_X^\top w\|_2,$$

since $\sup_{Y \in \Xi_{0,Y}}(-Y) = -\mu_Y + r_Y$ and $\sup_{X \in \Xi_{0,X}} w^\top X = \mu_X^\top w + t_X\|L_X^\top w\|_2$.

Taking the maximum of these two values yields

$$\sup_{\xi \in \Xi_0} f_{w,b}(\xi) = \left|\mu_Y - \mu_X^\top w - b\right| + r_Y + t_X\|L_X^\top w\|_2,$$

which is (56). □

### G.5. CV-selected Results with Additional $f$-DRO Baselines

Tables 7–8 report the East $\to$ West shift with 30% gap performance under geo-block CV selection, now including the additional KL-DRO and $\chi^2$-DRO baselines. LV remains the method with the smallest mean and tail errors under CV, and the fastest DRO method.

*Table 7.* California housing under geographic East $\rightarrow$ West deployment shift with a 30% gap, including additional KL-DRO and $\chi^2$-DRO baselines. Entries are mean (SD) over 100 replications under geo-block CV selection. All error metrics are reported in units of $10^4$.

| Method | MAE | RMSE | $p_{98}(|y - \hat{y}|)$ | $\text{CVaR}_{2\%}(|y - \hat{y}|)$ |
|---|---|---|---|---|
| LV | **10.69** (0.72) | **12.19** (0.77) | **21.86** (1.23) | **23.89** (1.10) |
| CVaR | 12.99 (0.00) | 14.40 (0.00) | 24.73 (0.00) | 27.25 (0.00) |
| Wass | 12.32 (0.02) | 13.75 (0.02) | 24.31 (0.04) | 26.50 (0.03) |
| KL-DRO | 12.66 (0.05) | 14.09 (0.05) | 24.70 (0.08) | 27.64 (0.08) |
| $\chi^2$-DRO | 12.68 (0.10) | 14.11 (0.10) | 24.74 (0.11) | 27.68 (0.10) |
| Ridge | 13.20 (0.01) | 14.60 (0.01) | 25.18 (0.01) | 27.66 (0.01) |
| ERM | 12.66 (0.00) | 14.08 (0.00) | 24.71 (0.00) | 27.66 (0.00) |

*Table 8.* Runtime comparison for DRO methods on the 30% gap California East $\rightarrow$ West deployment shift, including the additional KL-DRO and $\chi^2$-DRO baselines. Values are mean (SD) seconds over 100 replications. Validation is geo-block CV; likelihood time is nonzero only for LV in our implementation.

| Method | Validation | Likelihood | Solve | Total |
|---|---|---|---|---|
| LV | **1.30** (0.02) | **0.01** (0.00) | **0.13** (0.00) | **1.44** (0.02) |
| CVaR | 5.14 (0.09) | – | 0.34 (0.01) | 5.47 (0.09) |
| Wass | 5.80 (0.27) | – | 0.36 (0.02) | 6.16 (0.29) |
| KL-DRO | 39.10 (3.77) | – | 2.80 (0.40) | 41.90 (4.05) |
| $\chi^2$-DRO | 12.36 (0.83) | – | 2.00 (0.96) | 14.36 (1.38) |

## G.6. Alternative Gaps and Parameter Sweeps

Figure 12 visualises the trade-off between average and tail performance under geographic shift for gap ratios in $\{0, 0.1, 0.2, 0.3\}$. For each panel (gap ratio indicated in the title), we sweep the method-specific robustness parameter and plot the resulting West-test performance in the $(\text{MAE}, \text{CVaR}_{2\%}(|y - \hat{y}|))$ plane; ERM and the trivial train-median predictor appear as fixed reference points.

We also report an oracle-tuned comparison in Table 9. For each method, we pick the swept hyperparameter that minimises test $\text{CVaR}_{2\%}(|y - \hat{y}|)$, breaking ties by test MAE. This uses the test set for selection and is therefore only for diagnostic comparisons.

*Remark* G.2 (Note on the clipped-prediction ablation). The California housing target is top-coded at housing price $5 \times 10^5$. As an ablation, we recompute all metrics after clipping predictions at the price cap ($5 \times 10^5$) for 0% and 30% gap ratios. Across the full hyperparameter sweep and all methods, clipping changes MAE, RMSE, $p_{98}(|y - \hat{y}|)$, and $\text{CVaR}_{2\%}(|y - \hat{y}|)$ by at most 3.6%, 1.9%, 2.4%, and 3.8%. The qualitative trade-offs and relative comparisons are unchanged.

## G.7. Discussion on the Bulk Set $\Xi_{0,X} \times \Xi_{0,Y}$ in Regression

This subsection explains why we model the bulk set as an intersection $\Xi_0 = \Xi_{0,X} \times \Xi_{0,Y}$ in the California housing experiment, as opposed to a simple joint ellipsoid or box in $(X, Y)$. Under deployment shift, $Y \mid X$ often changes across regions. A joint ellipsoid can inadvertently encode the *training* relationship between $X$ and $Y$ as part of the bulk geometry. Lemma G.3 below shows the extreme version of this effect under an elliptical centre: $\varepsilon$ stops selecting different trade-offs and the method collapses to a single predictor.

We consider the ellipsoid bulk set. For $X \in \mathbb{R}^d$, $Y \in \mathbb{R}$, let $\xi := (X, Y) \in \mathbb{R}^{d+1}$. Write $\mu := (\mu_X, \mu_Y) \in \mathbb{R}^{d+1}$ and let $\Sigma \in \mathbb{S}_{++}^{d+1}$. Let $L \in \mathbb{R}^{(d+1) \times (d+1)}$ satisfy $\Sigma = LL^\top$ (e.g. a Cholesky factor). For $t > 0$, define the *joint* ellipsoidal bulk set

$$\Xi_0^{\text{joint}}(t) := \left\{ \xi \in \mathbb{R}^{d+1} : \left\| L^{-1}(\xi - \mu) \right\|_2 \leq t \right\}. \tag{57}$$

For a linear predictor $X \mapsto w^\top X + b$ with $(w, b) \in \mathbb{R}^d \times \mathbb{R}$, we have the absolute residual loss

$$f_{w,b}(\xi) := |Y - w^\top X - b|.$$

Consider the theoretical LV objective

$$J_\varepsilon(w, b) := (1 - \varepsilon)\mathbb{E}\left[ f_{w,b}(\xi) \mid \xi \in \Xi_0^{\text{joint}}(t) \right] + \varepsilon \sup_{\xi \in \Xi_0^{\text{joint}}(t)} f_{w,b}(\xi), \qquad \varepsilon \in [0, 1]. \tag{58}$$

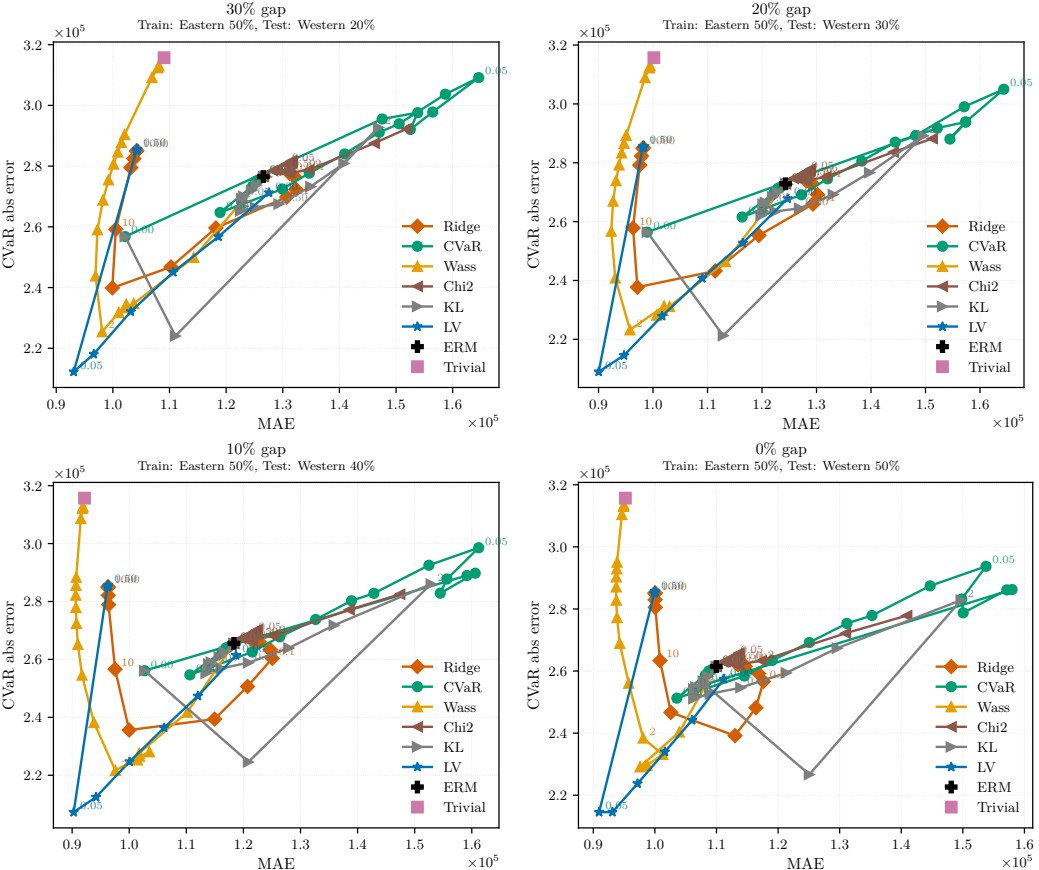

*Figure 12.* Parameter sweep trajectories on California housing regression under geographic shift. Each curve sweeps the method-specific robustness parameter. Some hyperparameter values are marked. The x-axis is MAE and the y-axis is $\mathrm{CVaR}_{2\%}(|y - \hat{y}|)$ on the West test set. Markers for ERM and the trivial train-median predictor are included as diagnostics when shown in the plots. Increasing the gap reduces the size of the West test region, as indicated by the titles.

We now show that, under an elliptical-centre model aligned with the joint ellipsoid, the minimiser of $J_\varepsilon$ does *not* depend on $\varepsilon$. This makes $\varepsilon$ obsolete for decision-making: it only rescales the same objective.

**Lemma G.3** (Joint-ellipsoid LV is $\varepsilon$-invariant)**.** *Assume the centre distribution is elliptically symmetric in the following explicit form:*

$$\xi = \mu + LU,$$

*where $U \in \mathbb{R}^{d+1}$ is spherically symmetric (i.e. for every orthogonal matrix $Q$, $QU \overset{d}{=} U$). Let $U_t$ denote $U$ conditioned on the event $\{\|U\|_2 \le t\}$ (assume $\Pr(\|U\|_2 \le t) > 0$ and $\Pr(U \ne 0 \mid \|U\|_2 \le t) > 0$), which is equivalent to conditioning $\xi$ on $\Xi_0^{\mathrm{joint}}(t)$ in (57). Define $J_\varepsilon$ as in (58).*

*Then, for every $\varepsilon \in [0,1]$, every minimiser $(w_\varepsilon, b_\varepsilon)$ of $J_\varepsilon$ satisfies*

$$b_\varepsilon = \mu_Y - \mu_X^\top w_\varepsilon, \qquad w_\varepsilon \in \arg\min_{w \in \mathbb{R}^d} \left\| L^\top [-w; 1] \right\|_2,$$

*and the argmin set is the same for all $\varepsilon \in [0,1]$. If the $d \times d$ block $\Sigma_{XX}$ is positive definite, the minimiser is unique and equals*

$$w_\varepsilon = \Sigma_{XX}^{-1} \Sigma_{XY}, \qquad b_\varepsilon = \mu_Y - \mu_X^\top \Sigma_{XX}^{-1} \Sigma_{XY}, \qquad \forall \varepsilon \in [0,1],$$

*where $\Sigma_{XY} \in \mathbb{R}^d$ is the X-Y cross-covariance block.*

*Proof.* For fixed $(w, b)$, define

$$a := \begin{bmatrix} -w \\ 1 \end{bmatrix} \in \mathbb{R}^{d+1}.$$

*Table 9.* Oracle selection on the 30% gap split using test data. For each method, the oracle chooses the swept parameter over a wider range that minimises test $\text{CVaR}_{2\%}(|y - \hat{y}|)$, breaking ties by MAE. Entries are mean (SD) over 100 replications. All error metrics are reported in units of $10^4$.

| Method | Oracle parameter | MAE | RMSE | $p_{98}(|y - \hat{y}|)$ | $\text{CVaR}_{2\%}(|y - \hat{y}|)$ |
|---|---|---|---|---|---|
| LV | $\varepsilon=0.05$ | **9.31** (0.31) | **10.79** (0.32) | **19.80** (0.43) | **21.23** (0.38) |
| CVaR | $\varepsilon=0.5$ | 11.89 (0.00) | 13.29 (0.00) | 23.46 (0.00) | 26.47 (0.00) |
| Wass | $\rho=2$ | 9.81 (0.03) | 11.55 (0.03) | 21.52 (0.02) | 22.55 (0.03) |
| KL-DRO | $\varepsilon_{\text{KL}}=5$ | 11.11 (0.00) | 12.74 (0.00) | 21.42 (0.00) | 22.41 (0.00) |
| $\chi^2$-DRO | $\varepsilon_{\chi^2}=1$ | 13.13 (0.00) | 14.53 (0.00) | 24.98 (0.00) | 27.77 (0.00) |
| Ridge | $\lambda=5$ | 9.99 (0.01) | 11.93 (0.01) | 23.31 (0.05) | 23.99 (0.04) |
| ERM | – | 12.66 (0.00) | 14.08 (0.00) | 24.71 (0.00) | 27.66 (0.00) |

Then for $\xi = (X, Y)$,

$$Y - w^\top X - b = a^\top \xi - b, \qquad f_{w,b}(\xi) = |a^\top \xi - b|.$$

Also define the scalars and vectors

$$c := a^\top \mu - b \in \mathbb{R}, \qquad v := L^\top a \in \mathbb{R}^{d+1}.$$

Using $\xi = \mu + Lu$, we have $a^\top \xi - b = c + a^\top Lu = c + v^\top u$.

We apply Lemma B.4 to the bulk set $\Xi_0^{\text{joint}}(t) = \{\xi : \|\Sigma^{-1/2}(\xi - \mu)\|_2 \leq t\}$ with $\Sigma := LL^\top$. This gives

$$\sup_{\xi \in \Xi_0^{\text{joint}}(t)} |a^\top \xi - b| = |a^\top \mu - b| + t\|\Sigma^{1/2} a\|_2 = |a^\top \mu - b| + t\|L^\top a\|_2. \tag{59}$$

By assumption, $U$ is spherically symmetric, i.e. $QU \overset{d}{=} U$ for every orthogonal $Q$. Also, $\|QU\|_2 = \|U\|_2$, so the event $\{\|U\|_2 \leq t\}$ is invariant under orthogonal transforms. Hence, for any orthogonal $Q$, the conditional distributions satisfy

$$QU \mid \{\|U\|_2 \leq t\} \overset{d}{=} U \mid \{\|U\|_2 \leq t\}.$$

That is, $U_t$ is also spherically symmetric.

Conditioning on $\xi \in \Xi_0^{\text{joint}}(t)$ is equivalent to conditioning on $\|U\|_2 \leq t$, so

$$\mathbb{E}\left[|a^\top \xi - b| \mid \xi \in \Xi_0^{\text{joint}}(t)\right] = \mathbb{E}\left[|c + v^\top U_t|\right].$$

Because $U_t \overset{d}{=} -U_t$ (take $Q = -I$), we have $v^\top U_t \overset{d}{=} -(v^\top U_t)$, so the random variable

$$Z := v^\top U_t$$

has a distribution symmetric about 0. Define $\phi(c) := \mathbb{E}[|c + Z|]$. The function $\phi$ is convex in $c$. Since $Z$ is symmetric about 0, 0 is a median. Therefore, $c = 0$ minimises $\mathbb{E}|c + Z|$, i.e. $b = a^\top \mu$ minimises the conditional expectation term for fixed $w$.

Independently, from (59), the supremum term is $|a^\top \mu - b| + t\|L^\top a\|_2$, and for fixed $w$ the only dependence on $b$ is through $|a^\top \mu - b|$, which is also minimised at $b = a^\top \mu$.

Hence, for every fixed $w$ and every $\varepsilon \in [0, 1]$, the value of $J_\varepsilon(w, b)$ is minimised over $b$ by

$$b^\star(w) = a^\top \mu = \mu_Y - \mu_X^\top w. \tag{60}$$

Substitute $b = b^\star(w)$, so $c = a^\top \mu - b = 0$. Then the supremum term becomes, by (59),

$$\sup_{\xi \in \Xi_0^{\text{joint}}(t)} |a^\top \xi - b^\star(w)| = t\|L^\top a\|_2.$$

The conditional expectation term becomes

$$\mathbb{E}\left[|v^\top U_t|\right] = \mathbb{E}\left[|(L^\top a)^\top U_t|\right].$$

We now show this expectation is exactly proportional to $\|L^\top a\|_2$. If $v = L^\top a = 0$, the expectation is $0$ and the proportionality holds trivially. Otherwise, set $u := v/\|v\|_2$ so $\|u\|_2 = 1$. Choose an orthogonal matrix $Q$ such that $Qu = e_1$ (the first canonical basis vector). Then

$$v^\top U_t = \|v\|_2 u^\top U_t = \|v\|_2 (Qu)^\top (QU_t) = \|v\|_2 e_1^\top (QU_t).$$

Since $U_t$ is spherically symmetric, $QU_t \stackrel{d}{=} U_t$, hence

$$|v^\top U_t| \stackrel{d}{=} \|v\|_2 |e_1^\top U_t|.$$

Taking expectations gives

$$\mathbb{E}[|v^\top U_t|] = \kappa_t \|v\|_2, \qquad \text{where } \kappa_t := \mathbb{E}\big[|e_1^\top U_t|\big] \in (0, t]. \tag{61}$$

Note that $|e_1^\top U_t| \leq \|U_t\|_2 \leq t$ holds a.s., hence $\kappa_t \leq t < \infty$; under the non-degeneracy assumption in the lemma statement, spherical symmetry implies $\Pr(|e_1^\top U_t| > 0) > 0$ and thus $\kappa_t > 0$. Combining the above, the objective at the optimal intercept (60) is

$$J_\varepsilon\big(w, b^\star(w)\big) = (1 - \varepsilon)\kappa_t \|L^\top a\|_2 + \varepsilon t \|L^\top a\|_2 = \big((1 - \varepsilon)\kappa_t + \varepsilon t\big)\|L^\top [-w; 1]\|_2.$$

The prefactor $(1 - \varepsilon)\kappa_t + \varepsilon t$ is a positive scalar depending on $\varepsilon$ but not on $w$. Therefore, for any $\varepsilon \in [0, 1]$, the set of minimisers over $w$ is exactly

$$\arg \min_{w \in \mathbb{R}^d} \|L^\top [-w; 1]\|_2,$$

which does not depend on $\varepsilon$. This proves the $\varepsilon$-invariance statement.

Since $\Sigma = LL^\top$ and $L$ is invertible, minimising $\|L^\top a\|_2$ is equivalent (by monotonicity of the square root) to minimising

$$\|L^\top a\|_2^2 = a^\top LL^\top a = a^\top \Sigma a.$$

Partition $\Sigma$ as

$$\Sigma = \begin{bmatrix} \Sigma_{XX} & \Sigma_{XY} \\ \Sigma_{YX} & \Sigma_{YY} \end{bmatrix}, \qquad \Sigma_{YX} = \Sigma_{XY}^\top.$$

Then, for $a = [-w; 1]$,

$$a^\top \Sigma a = \begin{bmatrix} -w^\top & 1 \end{bmatrix} \begin{bmatrix} \Sigma_{XX} & \Sigma_{XY} \\ \Sigma_{YX} & \Sigma_{YY} \end{bmatrix} \begin{bmatrix} -w \\ 1 \end{bmatrix} = w^\top \Sigma_{XX} w - 2w^\top \Sigma_{XY} + \Sigma_{YY}.$$

If $\Sigma_{XX} \succ 0$, this is a strictly convex quadratic in $w$ with unique minimiser given by the first-order condition

$$\nabla_w \big(w^\top \Sigma_{XX} w - 2w^\top \Sigma_{XY}\big) = 2\Sigma_{XX} w - 2\Sigma_{XY} = 0,$$

hence $w = \Sigma_{XX}^{-1} \Sigma_{XY}$. The corresponding intercept is $b = \mu_Y - \mu_X^\top w$ by (60). $\qquad \square$

A joint ellipsoid in $(X, Y)$ is a geometry where both "typical" behaviour (the conditional expectation term) and "worst-case" behaviour (the supremum term) are controlled by the *same* quadratic form $a^\top \Sigma a$, with $a = [-w; 1]$. Under an aligned elliptical centre, the expected absolute residual and the worst-case residual inside the ellipsoid are both exactly proportional to $\|L^\top [-w; 1]\|_2$. So $\varepsilon$ only changes a scalar prefactor and cannot change which $(w, b)$ minimises the objective. This matches our observations in practice: in Figure 13, we perform the same parameter sweep for LV with an ellipsoid bulk set. For different $\varepsilon$ values, the MAE and CVaR of LV barely move, and are clustered around ridge regression where the ridge parameter is small.

By contrast, the intersection construction

$$\Xi_0 = \Xi_{0,X} \times \Xi_{0,Y}, \qquad \Xi_{0,X} \subset \mathbb{R}^d, \ \Xi_{0,Y} \subset \mathbb{R},$$

is explicitly asymmetric in $(X, Y)$, which matches the asymmetry of regression itself. We use $\Xi_{0,X}$ to describe where covariates are "in distribution" (an ellipsoid over features), and $\Xi_{0,Y}$ to describe a plausible range of response values (an interval over prices).

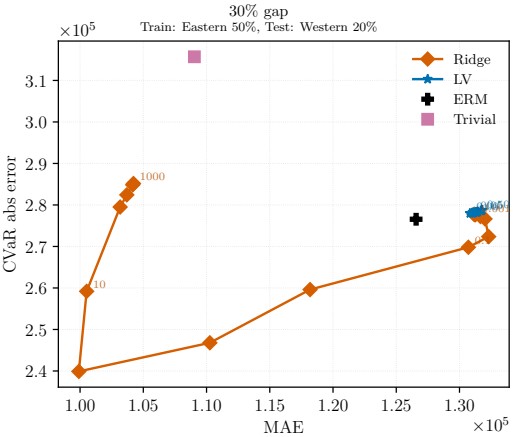

*Figure 13.* Parameter sweep trajectories of LV (ellipsoid bulk set) and ridge regression on California housing under East → West geographic shift with 30% gap.

Meanwhile, a joint box bulk set can be overly conservative because it permits independent worst-case movement along every dimension, as mentioned in Appendix E.1. This is supported by an additional ablation where we rerun California housing with a box bulk set on $(X, Y)$, defined by the score function $s_{\text{box}}(\xi) := \max_i |\xi_i - \mu_{\text{fit},i}|/w_i$, where $w_i := \sqrt{(\Sigma_{\text{fit}})_{ii}}$ is the fitted marginal scale of coordinate $i$, computed from $\mathcal{D}_{\text{fit}}$. Here we take $\gamma = 0.05$ because there are no longer two separate dimension blocks. The resulting parameter sweeps are reported in Figure 14. The MAE–CVaR trajectory for LV with the box bulk set is less stable than that for our $\Xi_{0,X} \times \Xi_{0,Y}$ construction, especially at small $\varepsilon$. This is consistent with the piecewise-linear optimisation target that a box bulk set implies, where small changes in $\varepsilon$ can essentially become tie-breakers between different extreme-point solutions.

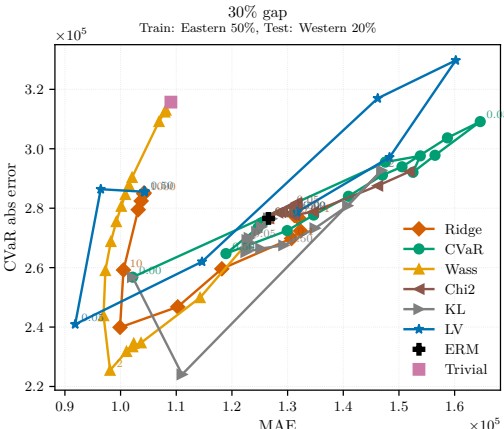

*Figure 14.* Parameter sweep trajectories of LV (box bulk set) and other baselines on California housing under East → West geographic shift with 30% gap.

## H. CivilComments Classification (Section 5.3): Additional Details and Results

This appendix provides additional details (Sections H.1–H.5) and results (Sections H.6–H.7) for the CivilComments classification experiment.

Section H.1 introduces the WILDS dataset splits and the worst-group metric; Section H.2 describes the classifier head, software, hardware, and runtime settings; Section H.3 lists the ERM, GroupDRO, CVaR, $\chi^2$-DRO, and LV-Group objectives; Section H.4 gives details for the empirical centre and the bulk set; Section H.5 describes minimax validation and checkpoint selection.

Section H.6 gives the $\varepsilon$-sweep and LV-Empirical comparison; finally, Section H.7 reports the bulk-filtering, smoothmax,

classifier-head, and feature-dimension ablations.

### H.1. Dataset and Task

We use the CivilComments dataset from the `WILDS` Python package (Koh et al., 2021). We report results on the official WILDS test split, and use the official WILDS validation split for selection. To facilitate DKW selection, we further split the official WILDS training split into an 80% subset for score fitting and a 20% subset for score selection.

Throughout Appendix H, a CivilComments example is denoted by $\xi = (x, y, g)$, where $x$ is the fixed hashed $n$-gram feature vector, $y \in \{0, 1\}$ is the binary label, and $g$ is the group indicator. We write $\theta$ for the classifier parameters and $f_\theta(\xi)$ for the per-example cross-entropy loss.

Worst-group accuracy is the minimum accuracy over the 16 overlapping identity×label slices. We use 8 identity indicators and 2 labels. A slice is $\{\text{identity}_j > 0\} \cap \{y = k\}$ for $k \in \{0, 1\}$. The identities are:

$$\{\text{male}, \text{female}, \text{LGBTQ}, \text{Christian}, \text{Muslim}, \text{other religions}, \text{black}, \text{white}\}.$$

### H.2. Model, Features, and Implementation

We use hashed $n$-gram features with a lightweight classifier head. We use $D = 4096$ hashed features and $n$-gram range = $(1, 1)$, with a linear head (a single affine layer). This is implemented in Python 3.11 using `PyTorch` version 2.9.1 (Ansel et al., 2024). Experiments were scheduled with SLURM and run on GPU nodes equipped with AMD EPYC CPUs (32 cores / 64 threads), 576 GB RAM, and 4× NVIDIA GPUs (48 GB). Wall-clock time in this experiment is dominated by model training (around 90% of runtime). The LV-Group discounting step adds a small overhead ($5 - 7\%$) compared to GroupDRO, and all methods have a similar end-to-end runtime for this experiment.

### H.3. Training Objectives

Let $\{\xi_i = (x_i, y_i, g_i)\}_{i=1}^n$ denote a minibatch of size $n$, and write $f_\theta(\xi_i)$ for the per-example cross-entropy loss. ERM minimises the empirical mean loss

$$\frac{1}{n} \sum_{i=1}^n f_\theta(\xi_i).$$

GroupDRO (Sagawa et al., 2020) maintains exponentiated-gradient weights $q \in \Delta_G$ over *disjoint* groups and minimises the robust group-weighted mean loss

$$\hat{R}_{\text{GDRO}}(\theta) = \sum_{g=1}^G q_g \left( \frac{1}{n_g} \sum_{i: g_i = g} f_\theta(\xi_i) \right),$$

where $g_i$ is the group label of example $i$ and $n_g$ is the number of minibatch examples in group $g$.

CVaR and $\chi^2$-DRO are implemented as batch-level DRO objectives. CVaR minimises the empirical tail risk

$$\min_{\eta \in \mathbb{R}} \left( \eta + \frac{1}{\varepsilon n} \sum_{i=1}^n (f_\theta(\xi_i) - \eta)_+ \right), \qquad (u)_+ := \max\{u, 0\}.$$

$\chi^2$-DRO uses a $\chi^2$-divergence uncertainty set (Namkoong & Duchi, 2016),

$$\sup_{p \in \Delta_n} \sum_{i=1}^n p_i f_\theta(\xi_i) \quad \text{s.t.} \quad \frac{n}{2} \sum_{i=1}^n \left( p_i - \frac{1}{n} \right)^2 \leq \rho,$$

and in our implementation $\chi^2$-DRO uses the $\chi^2$ loss. We parametrise the radius as $\rho = \varepsilon(n-1)/2$ so that $\varepsilon \in [0, 1)$ sweeps the full feasible radius range (ERM corresponds to $\varepsilon = 0$).

LV-Group uses

$$(1 - \varepsilon) \hat{R}_{\text{GDRO}}(\theta) + \varepsilon \, \text{smoothmax}_T(\{f_\theta(\xi_i)\}),$$

where $\text{smoothmax}_T$ is the log-sum-exp approximation to $\max_i f_\theta(\xi_i)$:

$$\text{smoothmax}_T(\{f_\theta(\xi_i)\}) = m + T \log \left( \frac{1}{n} \sum_i \exp((f_\theta(\xi_i) - m)/T) + 10^{-12} \right), \quad m = \max_i f_\theta(\xi_i).$$

We fix $T = 0.1$.

### H.4. Empirical Centre and Bulk Set

LV objectives use an in-bulk subset $\mathcal{B}$. The bulk score depends only on the feature component $x$ of $\xi = (x, y, g)$. We fit a class-conditional diagonal Gaussian in hashed feature space on the training subset to obtain an empirical centre $\{(\mu_y, \sigma_y)\}_{y \in \{0,1\}}$. We score each example by the label-free diagonal Mahalanobis distance

$$s(x) = \min_{y \in \{0,1\}} \sqrt{\sum_{j=1}^{D} \frac{(x_j - \mu_{y,j})^2}{\sigma_{y,j}^2}}.$$

We then estimate a threshold $\tau_{\gamma,\delta}$ with $\gamma = \delta = 0.05$ using a DKW-certified quantile on the score-selection split. Let $m$ be the number of calibration samples and set

$$r_{m,\delta} = \sqrt{\frac{\log(2/\delta)}{2m}}.$$

We take $q = 1 - \gamma + r_{m,\delta}$ and set $\tau_{\gamma,\delta}$ to the $q$-empirical quantile of the selection scores. We define

$$\Xi_0 = \{\xi = (x, y, g) : s(x) \le \tau_{\gamma,\delta}\}.$$

We apply this threshold to the official WILDS training split and define the in-bulk training set $\mathcal{B} := \{\xi = (x, y, g) : \xi \in \Xi_0\}$. LV-based methods and GroupDRO-B train on $\mathcal{B}$, whereas non-LV baselines (ERM, GroupDRO, CVaR, $\chi^2$-DRO) train on the full training split.

### H.5. Selection and Evaluation

Table 5 reports the mean and worst group accuracies selected by minimax validation for each method, using 20 replications. Minimax validation selection is implemented as follows: for each method, we choose the $\varepsilon$ that maximises the worst-group accuracy on the validation set. The chosen hyperparameters are: $\varepsilon_{\text{LV}} = 0.45$ for LV-Group; $\varepsilon_{\text{CVaR}} = 0.05$ for CVaR; and $\varepsilon_{\chi^2} = 0.5$ for $\chi^2$. We use epoch-wise checkpoint selection on the WILDS validation split, selecting the epoch with the lowest mean loss.

### H.6. Full Parameter Sweeps

Figure 15 reports the trade-offs across a sweep of hyperparameters in the range $\varepsilon \in [0, 1]$ with 21 evenly spaced points. We additionally report LV-Empirical, which minimises

$$(1 - \varepsilon)\frac{1}{n}\sum_{i=1}^{n} f_\theta(\xi_i) + \varepsilon\,\text{smoothmax}_T(\{f_\theta(\xi_i)\}),$$

without a GroupDRO term. Under minimax validation selection, it achieves the highest test mean accuracy $0.899\,(0.000)$ but worst-group accuracy $0.185\,(0.010)$ at $\varepsilon_{\text{LV}-\text{Em}} = 0.6$. Compared to LV-Group, it attains high mean accuracy while performing poorly on rare slices.

### H.7. Additional Ablations

In this section, we report four additional CivilComments ablations. These control for the effect of bulk filtering, the smoothmax approximation, the classifier head, and the feature dimension.

Figure 16 includes the bulk-filtered version of all baselines, which remain close to their unfiltered counterparts.

Figure 17 shows that the qualitative Pareto trade-off is stable under a $10\times$ sharper smoothmax approximation.

Figure 18a shows that the LV-Group effect persists beyond the linear-head setting in the same fixed-feature pipeline. Under minimax selection, LV-Group attains mean accuracy $0.829$ and worst-group accuracy $0.485$, while GroupDRO attains mean accuracy $0.856$ and worst-group accuracy $0.425$; this corresponds to a $14.1\%$ relative improvement in worst-group accuracy with a $3.2\%$ relative drop in mean accuracy. Figure 18b shows that the qualitative Pareto trade-off is also stable when doubling the number of hashed features.

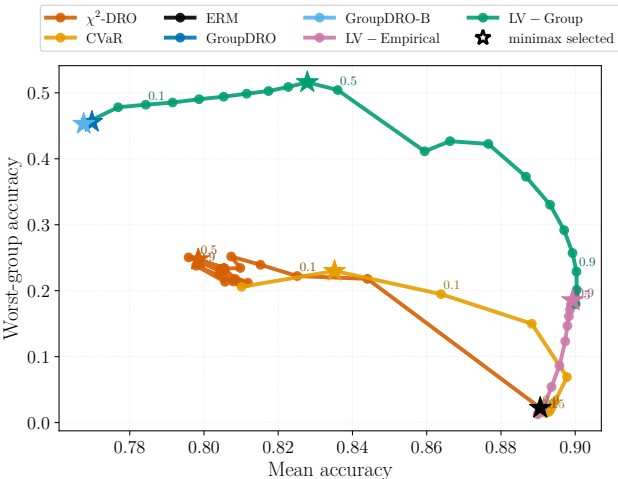

*Figure 15.* CivilComments test Pareto plot: worst-group accuracy vs mean accuracy. Each curve sweeps a hyperparameter grid. Some $\varepsilon$ values are marked. Stars mark the configuration selected by minimax validation, reported in Table 5.

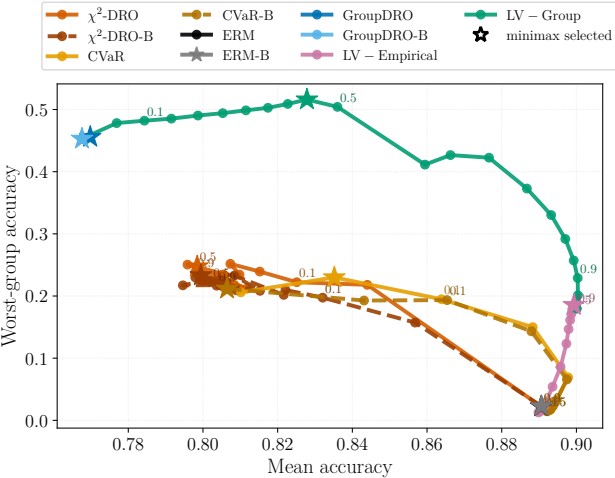

*Figure 16.* CivilComments test Pareto plot with bulk-filtered ablation in all baselines. $\chi^2$-DRO-B, CVaR-B, ERM-B, and GroupDRO-B are the $\chi^2$-DRO, CVaR, ERM, and GroupDRO baselines trained on the bulk-filtered training set used for LV-based methods. Each curve sweeps a hyperparameter grid. Some $\varepsilon$ values are marked. Stars mark the configuration selected by minimax validation. GroupDRO, GroupDRO-B, ERM, and ERM-B do not depend on $\varepsilon$ and are marked as fixed stars.

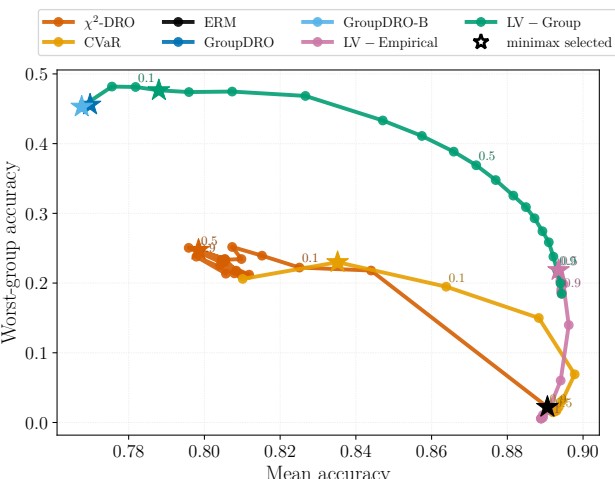

*Figure 17.* CivilComments test Pareto plot with a $10\times$ sharper smoothmax, using $T = 0.01$ instead of the $T = 0.1$ value used in the main experiment. Each curve sweeps a hyperparameter grid. Some $\varepsilon$ values are marked. Stars mark the configuration selected by minimax validation. GroupDRO and ERM do not depend on $\varepsilon$ and are marked as fixed stars.

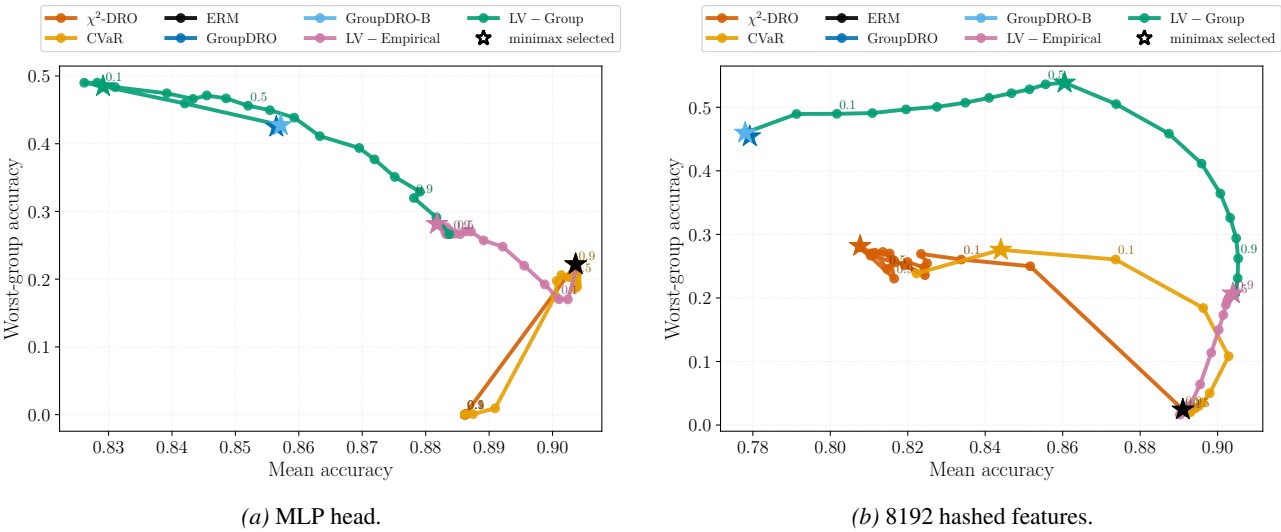

*(a)* MLP head.                                                   *(b)* 8192 hashed features.

*Figure 18.* CivilComments test Pareto plots for classifier-head and feature-dimension ablations: worst-group accuracy vs mean accuracy. Left: the linear classifier head is replaced by an MLP head of hidden width 256. Right: the same parameter sweep is run with double the number of hashed features, $2D = 8192$. Each curve sweeps a hyperparameter grid. Some $\varepsilon$ values are marked. Stars mark the configuration selected by minimax validation. GroupDRO and ERM do not depend on $\varepsilon$ and are marked as fixed stars.

