# OpenReview forum: "Bulk-Calibrated Credal Ambiguity Sets: Fast, Tractable Decision Making under Out-of-Sample Contamination"
_ICML.cc/2026/Conference — ICML 2026 spotlight_

### Official Review · Reviewer_7sXt · 2026-03-09

**Soundness:** 3
**Presentation:** 2
**Significance:** 2
**Originality:** 2
**Overall Recommendation:** 4
**Confidence:** 3

**Summary:**

The paper compares distributional robust optimization with minmax decision making under $\varepsilon$-contamination models.
For this, it first matches the most important concepts from both theories with one another, emphasizing that they are two sides of the same medal in many aspects. The paper then goes on addressing a fundamental issue of both theories: they cannot deal with situations where the domain of the loss function is unbounded. For this, the paper proposes to learn a suitable (i.e. respecting a mass certificate) bounded subset of the loss-domain from data (proposing a number of strategies) and to control the error outside this set via moment constraints. The framework is evaluated for both synthetic and real world data.

**Compliance With Llm Reviewing Policy:**

Affirmed.

**Final Justification:**

After reading the rebuttal from the authors, I raise my score from 3 to 4.

**Key Questions For Authors:**

1. I'm a bit confused by Equation (1): How is the operator Pr in this equation defined? Is it a probability measure? If yes, on which space? (my confusion stems from the fact that for $\Xi_0$ and $\mathbb{P}^*$ fixed, the inequality within Pr{} is either true or false.) Could you please clarify?

2. Please add a clear motivation of the relevance of your contributions to the introduction. My understanding (which might be limited) is that the main purpose of identifying a bulk set is to handle upper expectations of losses that are unbounded over their domain. Why is this an important problem to address?

3. The authors introduce $f_x$ as a loss function on page 1, line 014, second column. So my understanding of the expression $f_x(\xi)$ is "the loss that choosing $x$ yields if $\xi$ turns out to be the true state". However, Table 1 suggests that common choices for $f_x$ are hypotheses/models rather than loss functions. Could you please clarify what role $f_x$ is intended to play?

4. Lemma 3.2 demonstrates that a specific construction constructs bulk sets that satisfy the mass condition (1). But I think this set is not necessarily unique, right? Could there be a situation where you have two bulk sets $\Xi$ and $\Xi'$ and two acts $x$ and $y$ with $\mathbb{E}_{\Xi}(f_x -f_y) >0$ and vice versa for $\Xi'$? If yes, what does this tell about the usefulness for comparing expected losses?

**Limitations:**

yes

**Strengths And Weaknesses:**

While I believe that the paper might well have clear strengths, I find it extremely difficult to assess them adequately based on the presentation. The main contribution seems to me to be that upper expectations over unrestricted domains can be restricted in a data-driven manner, while at the same time certain mass restrictions for the obtained bulk set are guaranteed. In my opinion, the paper can only be recommended for publication if the presentation is significantly improved, both in terms of motivation and in terms of explanations and context. My specific comments are as follows:


Strengths:

- In principle, I think it is worth exploring how DRO and decision-making under IP (as formally very related issues) can benefit from each other. This avoids the same discoveries having to be made separately in both disciplines and thus helps to streamline the research process.

- I think the results given in Table 1 (i.e. the closed-form expressions for the for the supremal losses over the bulk set depending on the chosen threshold t and the score function s) are really useful. However, it is not entirely clear to me, whether these results are (all, in parts) due to the authors, or if this is rather a collection of already known results. Please clarify.

- The authors extensively and carefully evaluate and demonstrate their results both on synthetic and real world data.




Weaknesses:

- I find it somewhat difficult to discern the actual original contribution of this paper. A significant part of the results seems to consist of translating DRO and IP nomenclature into each other. While it is undoubtedly interesting to make these parallels explicit, I am unsure whether this constitutes a sufficient independent contribution.

- The paper clearly states its contributions at the end of Section 1. I think this is in general good. However, I am really missing a motivation or justification why these contributions are relevant and if, for what.

- Sections 2 and 4 are written extremely dense; the provided results are neither motivated nor interpreted enough. Especially in Section 2, the selection of presented makes it hard to see a clear red line. Also, I do not fully get the value of presenting an alternative proof for an already existing (quite recent) result in the main text. I think the space would be better devoted to stating the actual intention of the work.

- The papers states (lines 150 to 153) that the truncated expectations can be approximated using SAA with rejection sampling, without further explanation or elaboration of approximation quality. I think some more details would be helpful here.


Minor remarks:

- page 2/3, lines 109-110: Please write $\mathbb{P}_c=...$ also for the plug-in predictive and the empirical distribution to avoid confusion.

---

> ### Author Rebuttal · Authors · 2026-03-30
>
> Thank you for your time and feedback. We appreciate you recognising our results as *extensively and carefully evaluated*. We will revise the paper to improve clarity and address the concerns raised.
>
> **Original contribution and importance**
> While the IP–DRO duality is a valuable conceptual framework and motivation for this work, it is not the main contribution. We will revise the contributions list to highlight the following:
> - Bulk-calibrated ambiguity sets that make forward Huber contamination meaningful, instead of vacuous, in unbounded sample spaces with unbounded losses. This setting encompasses commonly used loss functions and data domains (e.g., something as simple as linear regression with Gaussian samples), and it is relevant for any ambiguity set that admits Huber contamination, including TV and MMD balls.
> - DKW-based bulk calibration. This yields finite-sample bulk-mass guarantees which the ad hoc truncations in existing work cannot provide in general, as detailed in Appendix E. It is computationally efficient ($O(m \log m)$) and supports flexible and block-wise specification of bulk sets, enabling applicability across heterogeneous problem classes, as illustrated in our experiments.
> - A finite mean+sup objective with tractable LP/SOCP formulations for common losses: this leads to shorter runtimes, as empirically validated in our experiments.
> - A resulting risk certificate that provides a structural decomposition of the risk bound and enables practitioners to isolate, interpret, and diagnose the distinct contributing components.
>
> **Presentation clarity**
> We will revise the paper in the following ways:
> - Add a roadmap at the start of Section 2: (i) defining the bulk-restricted credal ambiguity set and providing two crucial quantities associated with it: the worst-case risk and worst-case distribution. (ii) showing that the ambiguity set can be written as a forward-LV ball, which places it in a familiar DRO form. (iii) linking the LV ball with two other commonly used contamination neighbourhoods (reverse LV and TV balls), providing a unifying interpretation.
> - Divide Section 2 into three subsections according to (i)-(iii) above, with clear subtitles and introductions to each subsection.
> - Move the alternative proof to the appendix; use the space to reduce notation density and provide more intuition behind every definition and theorem.
> - In section 4, clarify that under the settings of this paper, we select $\varepsilon$ by validation and move the additional discussion on bounding and estimating $\varepsilon$ to the appendix to avoid confusion.
> - Clarify that the results in Table 1 are applications of known convex optimisation results. They are tailored to our new framework to make it convenient for practitioners' implementation. We will cite the textbook “Convex Optimization” by Boyd and Vandenberghe (2004).
>
> - Use $\mathbb{P}_c$ for all centre choices.
>
> **Clarifying technical points**
> We will add these clarifications to Equation (1) and Table 1 in the paper:
> - $\Pr$ is the probability measure on the sample space $\Xi^n$ induced by the random training sample $\mathcal D\sim (\mathbb P^\star)^n$ used to construct the data-dependent set $\Xi_0(\mathcal D)$, so the event is random through $\mathcal D$.
> - For each fixed decision parameter $x$, $f_x(\xi)=f(x,\xi)$ is the induced loss as a function of the uncertain outcome $\xi$, so Table 1 lists tractable functional forms of this loss-in-$\xi$ (with coefficients depending on $x$), not hypothesis classes.
>
> **Bulk-set non-uniqueness**
> We may not have explained this clearly enough, so we will add the following discussion after Lemma 3.2: Indeed, the lemma does not imply uniqueness. Many bulk sets can satisfy this condition, and different certified geometries can rank two acts differently. This means that our bulk set geometry is a modelling choice: it determines which contamination types are plausible, just as ambiguity-set geometry matters in DRO more broadly. The role of Lemma 3.2 is not to identify a unique “true” bulk set, but to certify that the chosen $\Xi_0$ satisfies condition (1). This makes the resulting risk certificate meaningful and avoids ad hoc truncations that lack coverage control. We also refer you to our response to reviewer X6QV on **score choice** for a detailed discussion of the implications of bulk set choice on optimisation and a rule of thumb for practitioners.
>
> **Rejection sampling SAA for truncated expectation**
> We have plotted the convergence diagnostic of the rejection sampling SAA to the theoretical truncated expectation in Figure 5 at link https://anonymous.4open.science/r/credal-dro. For all $\gamma$ values, the SAA average stabilises around the theoretical value with about $400$ accepted samples. This is consistent with our sample efficiency study in Appendix F.7, where LV stabilises quickly with small SAA budgets. The new plot and explanation will be added to the appendix with a pointer on lines 150-153.

---

> > ### Author Rebuttal · Reviewer_7sXt · 2026-04-02
> >
> > I thank the authors for their detailed response.
> >
> > Their proposed changes, especially in regards of presentation clarity and discussing the non-uniqueness and its consequences for decision making, do indeed address my concerns. However, since these are relatively significant changes to the manuscript that can only be verified in the final paper, some lingering concerns remain. Nevertheless, I am raising my score from 3 to 4.

---

> > > ### Author Response · Authors · 2026-04-02
> > >
> > > Thank you very much for the thoughtful reply and for raising your score. We appreciate your recognition that the proposed changes address your main concerns. We are committed to implementing the revisions fully and carefully, so that the final manuscript clearly reflects the improvements we outlined in our rebuttal, which will be made publicly available upon acceptance.

---

### Official Review · Reviewer_Vvtc · 2026-03-12

**Soundness:** 3
**Presentation:** 4
**Significance:** 4
**Originality:** 4
**Overall Recommendation:** 5
**Confidence:** 4

**Summary:**

This paper addresses a fundamental limitation in Distributionally Robust Optimization (DRO) when using Huber (linear-vacuous) contamination models. Traditionally, including arbitrary out-of-sample contamination in an ambiguity set can lead to vacuous (infinite) worst-case risk unless strict boundedness or support assumptions are made. The authors introduce BulkCalibrated Credal Ambiguity Sets, a framework that partitions the problem into a data-driven "bulk set" (capturing high-mass regions) and a separately bounded tail contribution.

The primary contributions are:
1. Theoretical Framework: The authors establish a formal connection between the imprecise probability (IP) concept of upper expectation and the DRO worst-case risk.
2. Tractable Objectives: They derive a closed-form "finite mean + sup" robust objective, which allows for fast, tractable optimization via linear or second-order cone programming for common loss functions.
3. Empirical Validation: The method is tested on diverse tasks including heavy-tailed inventory control, house-price regression under geographical shifts, and text classification under demographic shifts.

**Compliance With Llm Reviewing Policy:**

Affirmed.

**Key Questions For Authors:**

1. Bulk Set Definition: How sensitive is the final decision to the geometry of the bulk set (e.g., box vs. ellipsoid) in high-dimensional settings?
2. Comparison to SOTA: Can you provide a more detailed runtime comparison against standard Wasserstein DRO or $f$-divergence DRO solvers for the house-price regression task?
3. Adaptive Calibration: In non-stationary environments where the "bulk" might shift over time, does the framework support efficient online updates to the ambiguity set?

**Limitations:**

Yes. The authors discuss the limitations of standard contamination models and honestly address where their bounds are most effective.

**Strengths And Weaknesses:**

Strengths:
- Originality: By bridging the gap between Imprecise Probability (IP) and DRO, the work provides a novel and interpretative way to set tolerance levels for robust optimization.
- Soundness: The proposed "bulk-calibration" elegantly avoids the vacuity issues of standard Huber contamination models without resorting to overly restrictive global support assumptions.
- Significance: The derivation of closed-form objectives is a major practical advantage, as it enables robust decision-making at a computational speed comparable to standard nonrobust methods.
- Presentation: The experimental section is thorough, illustrating the model's performance on Pareto frontiers that visualize the trade-off between accuracy and robustness across different feature dimensions.
Weaknesses:
- Hyperparameter Sensitivity: The framework relies on the definition of the "bulk set" and the contamination fraction $\epsilon$. While the authors provide Bayesian and frequentist reference distributions, the automated selection of these parameters in purely non-parametric settings could be more deeply discussed.
- Baseline Comparisons: While the results are competitive, additional comparisons with newer Wasserstein-based DRO methods in terms of computational time (not just accuracy) would further highlight the "Fast, Tractable" claim.

---

> ### Author Rebuttal · Authors · 2026-03-30
>
> Thank you for your time and feedback. We appreciate you recognising our work as *novel, interpretative, and sound; with major practical advantage and thorough experiments*. We will revise the paper to improve clarity and address the concerns raised.
>
> **Nonparametric selection of the bulk set and contamination fraction**
> Conditional on a chosen score family and coverage target $(\gamma,\delta)$, the DKW step selects the bulk set threshold nonparametrically and distribution-freely. We will discuss the score choice (which determines the bulk set geometry) in the next point, and $\varepsilon$ is treated separately because the true test-time contamination level is not identifiable from the training data alone. We tune $\varepsilon$ via validation, similar to Wasserstein DRO. We will revise Sections 3-4 to more clearly separate these choices. The value of $\delta$ is often fixed at the required significance level (e.g., 0.05, 0.01), and we refer you to our response to reviewer X6QV on **$\gamma$-sensitivity analysis** (see also Figure 1 at link https://anonymous.4open.science/r/credal-dro) for empirical guidance on choosing the hyperparameter $\gamma$, which we will include in the paper.
>
> **Bulk set definition**
> An ellipsoid respects dependence and gives a smoother SOCP robustification, whereas a box lets each coordinate move independently and can therefore be more conservative and less stable. Meanwhile, a joint ellipsoid bulk set can encode training dependence between coordinates, which makes it unsuitable for problems with heterogeneous dimensions such as regression (see a detailed discussion in Appendix G.8). This is why we choose an ellipsoid by default and use ellipsoid-$X$-times-interval-$Y$ for the California housing regression. See also our response to reviewer X6QV on **score choice** for the fuller geometry discussion and the added box-bulk-set ablation experiment. Our practical rule of thumb is: default to ellipsoid, use blockwise/separate bulk sets for asymmetric or heterogeneous problems such as regression, and use box scores only when independent coordinatewise deviations are the intended adversary.
>
> **SOTA comparison for California housing**
> The standard Wasserstein baseline is in Table 4 of the paper; we now extend it with two standard $f$-divergence baselines, KL and $\chi^2$ (cited in the paper), under the same CVXPY+MOSEK pipeline. LV remains favourable both statistically and computationally.
>
> Table for runtime comparison (reported mean (std) of seconds, please note that the absolute times for LV and Wasserstein differ from those in Table 4 of the paper due to hardware updates; the comparative conclusions remain unchanged):
>
> | Method | Validation | Likelihood | Solve | Total |
> |:--|:--:|:--:|:--:|:--:|
> | LV | **1.30**&nbsp;(0.02) | 0.01&nbsp;(0.00) | **0.13**&nbsp;(0.00) | **1.44**&nbsp;(0.02) |
> | Wasserstein | 5.80&nbsp;(0.27) | — | 0.36&nbsp;(0.02) | 6.16&nbsp;(0.29) |
> | $\chi^2$ | 12.36&nbsp;(0.83) | — | 2.00&nbsp;(0.96) | 14.36&nbsp;(1.38) |
> | KL | 39.10&nbsp;(3.77) | — | 2.80&nbsp;(0.40) | 41.90&nbsp;(4.05) |
>
> So LV is about $4.3\times$ faster end-to-end than Wasserstein, $10.0\times$ faster than $\chi^2$, and $29.1\times$ faster than KL on this task; its solve time is also the smallest. This is expected computationally: $\chi^2$ solves a larger conic program, and KL requires exponential cones, while LV keeps the finite mean-plus-sup structure with an SOCP formulation. Additionally, in the hyperparameter sweep trajectory plot with the added baselines (Figure 4 at link https://anonymous.4open.science/r/credal-dro), LV still gives the most favourable MAE-CVaR trade-off. The new baselines will be added to Section 5.2 and Appendix G of the paper.
>
> **Online bulk set calibration**
> This is a very interesting question, and it would be a useful and natural extension to our framework, which has the necessary modularity to achieve this. With new deployment data, we can refit or update the score model and the centre distribution, recompute scores, and rerun the same DKW bulk calibration procedure, while keeping the downstream LP/SOCP objective unchanged. Updating the bulk calibration threshold is computationally efficient, with complexity $O(m \log m)$, so it would not add substantial overhead. Appendix D provides a cheap score-based alignment diagnostic, which would be useful for deciding when a centre refit is needed. Another follow-up would be to link our DKW procedure with existing online conformal prediction methods to establish formal long-run or anytime guarantees; e.g., see Gibbs and Candès (2021) and the recent work by Gauthier et al. (2025). We will include this as a promising future direction in the paper's concluding remarks.
>
> Additional citations:
>
> [1] Gibbs and Candès (2021). Adaptive conformal inference under distribution shift. *NeurIPS*.
>
> [2] Gauthier et al. (2025). E-values expand the scope of conformal prediction. *arXiv:2503.13050*.

---

### Official Review · Reviewer_wnAV · 2026-03-13

**Soundness:** 3
**Presentation:** 2
**Significance:** 2
**Originality:** 3
**Overall Recommendation:** 4
**Confidence:** 4

**Summary:**

This paper studies distributionally robust optimization (DRO) under forward Huber (linear-vacuous) contamination models in settings where the data space is continuous and unbounded. In such settings, the standard worst-case formulation can become vacuous because the adversary can place probability mass arbitrarily far in the tails, making the worst-case loss infinite.

To address this issue, the paper proposes restricting the adversarial contamination to a high-probability “bulk” region of the data distribution that is learned from the training data. The idea is to identify a bounded set that captures most of the probability mass of a reference distribution and to allow contamination only within this region, while treating the tail outside the region separately. This produces a tractable robust objective that combines an average loss over the bulk with a worst-case loss within that region.

**Compliance With Llm Reviewing Policy:**

Affirmed.

**Final Justification:**

The authors have addressed my main concerns.

**Key Questions For Authors:**

None

**Limitations:**

Partially. The paper does discuss some limitations, especially the sample-size burden of the DKW calibration and the risk of overly conservative robustness choices. However, it should also explicitly discuss three additional limitations: (i) the main certificate is not operational because quantities such as εc, the contaminant bulk mass, and $M_p(x)$ are unknown at deployment; (ii) the empirical-center / CivilComments implementation is only loosely connected to the exact theory and uses filtering plus smoothmax approximations; and (iii) performance appears sensitive to the chosen bulk geometry, which requires nontrivial domain knowledge and may itself be a major modeling decision.

**Strengths And Weaknesses:**

**strengths and weaknesses**
The support-restricted construction is simple, interpretable, and computationally friendly. The core formal results, Theorem 2.1, Proposition 2.3, the DKW calibration lemma, and the appendix proof for Proposition 2.4 all look mathematically sound. I also think Appendix G.8 / Lemma G.3 provides a useful explanation for why the California experiment uses an asymmetric bulk geometry.

My main concern is the gap between the theory and the way the method is actually used. Section 4 is framed as tolerance selection / calibrating $\epsilon$, but the actual conclusion is that the true contamination level  $\epsilon^\*$ is not identifiable from training data and $\epsilon$ is chosen by validation. That is a reasonable practical choice, but it is much weaker than the framing sometimes suggests. Similarly, the high-probability certificate in Theorem 3.4 depends on quantities that are not operational in practice, including the in-bulk mismatch $\epsilon_c$, the deployment contaminant’s bulk mass, and the deployment p-moment $M_p(x)$. As a result, the theorem is more of a qualitative justification than a deployable prescription. At minimum, the paper should be much more explicit about this.

A second, related issue is that the empirical-center story does not line up cleanly with the theory. The paper claims flexible center choices (Bayesian, frequentist, empirical), but Remark B.3 states that if the empirical measure is treated as a literal center in an atomless model, then $LV(\mathbb{P}_{\Xi_0}, \hat{\mathbb{P}}^n_{\Xi_0})=1$.

The paper then reinterprets the empirical option as SAA rather than as the same center notion used in the theory. That is especially important because the CivilComments experiment is exactly the empirical-center case.

The CivilComments implementation makes the theory/practice gap even larger. In that experiment, the bulk restriction is approximated by filtering the official training data to an in-bulk subset B, and the max term is approximated by a smoothmax over minibatch losses. This is a practical heuristic inspired by the theory, not the exact population objective introduced earlier. In addition, LV-based methods and GroupDRO-B train on the filtered subset B, whereas the non-LV baselines train on the full training split. GroupDRO-B is a useful ablation, but the paper should be more careful not to over-attribute the gains to the LV objective alone when data filtering is also changing the problem.

I also found some of the empirical comparisons less clean than they first appear. In the Bayesian newsvendor experiment, the runtime and sample-efficiency advantage partly comes from the fact that the LV implementation solves a fixed-size truncated SAA with size 0.5M, while the KL-based baselines use all M scenarios. The appendix argues that LV stabilizes quickly and that this does not change the qualitative conclusion, but the runtime comparison is still not fully apples-to-apples. In the California experiment, the strongest results rely on a fairly tailored product geometry $\Xi_{0,X}\times \Xi_{0,Y}$. Appendix G.8 gives a rationale for this choice and even shows that a joint ellipsoid can make ε ineffective. That is interesting, but it also means the method’s success depends quite heavily on choosing the right bulk geometry for the application. I do not think the paper fully grapples with how a practitioner should do that in general.

Presentation is a major weakness for me. The paper is mathematically dense, notation-heavy, and not very accessible to a broad ICML audience. A large fraction of the exposition is organized as remarks, caveats, and side questions rather than a single coherent narrative. For example, Remark 5.1 is literally framed as a question (“Why an ellipsoid-interval product bulk set instead of an ellipsoid?”), which is symptomatic of a broader writing pattern. I frequently felt that the paper was assembled from separate explanatory fragments rather than written as one continuous story. Proposition 2.4 is another example: the text says it gives an “alternative proof,” but the main exposition never clearly says alternative to what; presumably it means an alternative proof of the known result cited from Kuhn et al. (2025), but the wording is unnecessarily unclear. More broadly, if the paper wants to speak to both IP and ML audiences, it needs substantially more intuition, fewer abrupt notation jumps, and clearer separation between exact theory, heuristics, and implementation details.  This point really drives my score.

On significance, I am mixed. The underlying problem is important and the bulk-restricted forward-contamination idea is interesting. However, the practical prescription is less complete than the theoretical framing suggests, and the empirical evidence is concentrated in settings where the method is especially tractable. The classification experiment deliberately avoids modern end-to-end models and uses a fixed hashed n-gram representation with a linear head, which is reasonable for control but limits the practical impact for current ML.

---

> ### Author Rebuttal · Authors · 2026-03-30
>
> Thank you for your time and feedback. We appreciate you recognising our framework as *interpretable, computationally friendly, and mathematically sound*. We will revise the paper to improve clarity and address the concerns raised.
>
> **$\varepsilon$ not identifiable; Theorem 3.4 involves unknown quantities**
> We stated in Section 4 that the true contamination level is not identifiable from training data, so $\varepsilon$ is a robustness budget and selected by validation, similar to Wasserstein radius tuning in Wasserstein DRO. This is an inherent property of the distributional shift we are targeting. Theorem 3.4 is a structural certificate that decomposes the OOS risk bound and identifies the contribution of each component. We agree that it is not a plug-in operational bound in the train-only setting, and we will state that plainly in Section 3.
>
> **Empirical centre is interpreted as SAA**
> We will clarify the following in Sections 2-3:
> - In Theorem 2.1, setting $\mathbb P_c=\hat{\mathbb{P}}_n$ gives the standard SAA sample analogue of the population risk used in most empirical DRO work.
> - In Theorem 3.4, parametric centres have a centre-mismatch term $\varepsilon_c$. With an empirical centre, there is no parametric mismatch, so $\mathbb P_c$ is theoretically $\mathbb P^\star_{\Xi_0}$, and the empirical centre only contributes to the SAA error in approximating $\mathbb E_{\mathbb P^\star_{\Xi_0}}[f_x]$, analogous to how the truncated expectation is approximated via rejection sampling. Remark B.3 clarifies the necessity of this interpretation, which is not unique to LV: KL sets centred at an empirical measure have the same support mismatch.
>
> **CivilComments experiment**
> We added new ablation experiments to address concerns about bulk filtering, approximation, and applicability; new figures are at https://anonymous.4open.science/r/credal-dro and will be added to Appendix H.
> - Bulk-filtered ablation: we added this to all baselines, confirming that bulk-filtering does not materially change baseline accuracies; see Figure 6.
> - Smoothmax on minibatches: this is a standard stochastic approximation, and we reran the experiment with a $10\times$ sharper smoothmax ($T=0.01$ instead of $0.1$), which leaves the Pareto picture essentially unchanged; see Figure 7.
> - Practical impact: We reran the experiment with an MLP head. LV-Group improves the worst-group accuracy of GroupDRO by 14.1% under minimax selection with a 3.1% relative drop in mean accuracy; the parameter sweep includes a region where LV-Group dominates GroupDRO and GroupDRO-B; see Figure 8. This shows that the benefits of LV can be transferred to modern ML architectures.
>
> **Newsvendor runtime**
> This is stated explicitly as an end-to-end runtime comparison. The paper attributes the faster runtime of LV to its sample efficiency and discloses that LV uses SAA size $0.5M$. Appendix F.7 explains that LV requires far fewer samples than the KL baselines. We will improve the pointer in Section 5.1 to Appendix F.7 with more explanation.
>
> **Bulk geometry choice**
> We agree that the bulk set geometry is an important choice. For the California housing experiment, Appendix G.8 provides intuitive, theoretical, and empirical explanations for why the product bulk set is a good choice for problems with heterogeneous dimensions. We also refer you to our response to reviewer X6QV on **score choice** for the practitioner rule of thumb and the additional box-bulk ablation; the former will be added to the main text, and the latter will be added to the appendix.
>
> **Empirical evidence concentrated in tractable settings**
> Our goal is not to claim a black-box solution for any loss function, but to make contamination models usable in typical DRO regimes. Table 1 gives closed-form in-bulk supremum formulas for various loss functions. The newsvendor and California housing regression objectives are instantiations of these templates. The empirical evidence is also not limited to exact convex solvers: Section 5.3 studies a stochastic cross-entropy training.
>
> **Paper Presentation**
> We agree that the paper's presentation can be made clearer. We refer you to our response to reviewer 7sXt on **presentation clarity** for our revision plan. We will also use the additional page allowed for the camera-ready version to:
> - Write Remark 5.1 into a full subsection and move the explanations in Appendix G.8 to the main text, along with the score-choice rule of thumb for practitioners
> - Provide a table that separates theoretical targets from approximations for all centre choices
> - Add paragraphs that connect the sections to improve the flow and coherence of the paper
> - Provide more background and intuition for IP and DRO concepts and cite recent surveys in both fields
> - Explain that the alternative proof is to the cited result in Kuhn et al. (2025)
>
> Please let us know of any further presentation concerns not addressed by our current plans, and we will revise the paper accordingly.

---

> > ### Author Rebuttal · Reviewer_wnAV · 2026-04-05
> >
> > Thank you, though the questions regarding clarity can only be verified in the final version of the paper, I have updated my scores accordingly.

---

> > > ### Author Response · Authors · 2026-04-06
> > >
> > > Thank you very much for the thoughtful reply and for updating your score. We appreciate your acknowledgement that your concerns have been resolved. We are committed to implementing the proposed clarity revisions fully and carefully so that the final manuscript clearly reflects the improvements outlined in our rebuttal, which will be made publicly available upon acceptance.

---

### Official Review · Reviewer_X6QV · 2026-03-13

**Soundness:** 3
**Presentation:** 3
**Significance:** 2
**Originality:** 3
**Overall Recommendation:** 4
**Confidence:** 4

**Summary:**

In this paper, the authors propose a method for making robust decisions when the credal set is the Huber-contamination set. They demonstrate the equivalence between the Huber-contamination set and the ambiguity set with LV distortion, and derive the expression for the worst-case risk. When both the space $\Xi$ and the loss function $f_x$ are unbounded, the worst-case risk is typically infinite, making it intractable to obtain a robust decision. To address this problem, the authors propose learning a bounded bulk set $\Xi_0 \subset \Xi$ with an explicit mass certificate and making robust decisions by minimizing the worst-case risk based on this bounded set $\Xi_0$. Finally, the authors present relevant theoretical properties and experimental results. The experimental results demonstrate that the proposed method outperforms baseline methods in some scenarios.

**Compliance With Llm Reviewing Policy:**

Affirmed.

**Final Justification:**

The authors addressed several questions regarding $\gamma$-sensitivity analysis, score choice, and the Bayesian-centre experiment under other shifts through experimental validation. After reading the response, I have a deeper understanding of the paper. My concerns have been resolved, and my curiosity has been satisfied.

**Key Questions For Authors:**

1 If the coverage rate $1-\gamma$ is either too small or too large, will the performance of the proposed method still be better than that of other baseline methods? A sensitivity analysis of the parameter  $1-\gamma$ could further demonstrate the robustness of the proposed method.

2 Using different score functions can yield distinct bulk sets $\Xi_0$, thereby leading to different final decisions. Is the proposed method sensitive to the choice of score functions? Furthermore, how can we choose a good score function to establish an effective bulk set $\Xi_0$?

3 In the experiment titled "Bayesian Centre: Heavy-tailed Synthetic Newsvendor with Demand Spikes", the out-of-sample distribution is contaminated by adding a certain fraction of "spike" components. If the distribution shift occurs in other forms (e.g., the data distribution changes from a t-distribution to a normal distribution), does the proposed method still perform better than other methods?

**Limitations:**

yes

**Strengths And Weaknesses:**

Strengths: To avoid the worst-case risk based on the unbounded space $\Xi$ being infinite, the authors propose learning a bounded bulk set $\Xi_0\subset \Xi$ with a mass certificate and making robust decisions by minimizing the worst-case risk on this bounded set $\Xi_0$. According to the experimental results, this method achieves better performance compared to the baseline methods, which indicates its effectiveness.

Weaknesses: However, the method seems to directly overlook the information in the complement set $\Xi/\Xi_0$. Moreover, there is no discussion of the influence of the coverage rate $1-\gamma$ on the performance of the proposed method in applications.

---

> ### Author Rebuttal · Authors · 2026-03-30
>
> Thank you for your time and feedback. We appreciate you recognising the *effectiveness* of our method. We will revise the paper to improve clarity and address the concerns raised.
>
> **$\gamma$-sensitivity analysis**
> This is a very helpful question: running the analysis both confirmed that LV is stable over a reasonable range of $\gamma$, and provided practical insights on how to choose it. We did a $\gamma$-sensitivity analysis in the Bayesian newsvendor experiment, see Figure 1 at link https://anonymous.4open.science/r/credal-dro. The minimum certifiable $\gamma$ for this experiment, given our selection-set size, is $0.043$, and we studied $\gamma$ between $0.043$ and $0.15$.
> - For $\gamma \in [0.05,0.10)$, the frontiers are similar and stay in the same favourable trade-off region relative to the baselines.
> - Some unstable behaviour appears at the minimum certifiable value $0.043$, which means that the DKW threshold is the single largest selection score. At large $\varepsilon$, the optimisation can be overly driven by that point. Moving from $0.043$ to $0.05$ already removes this effect.
> - For larger $\gamma\ge0.10$, the bulk becomes too restrictive, and the supremum term becomes less informative, so LV can plateau in a low-mean / mid-variance region.
>
> A *practical rule of thumb* suggested by this analysis is therefore to choose $\gamma$ just above the minimum certifiable value and leave a small tail outside the bulk (about 10 scores, so $\gamma \approx r_{m,\delta}+10/m$). We tested this rule in the California housing and CivilComments experiments, where the corresponding choice gave strong and stable performances (see Figures 9, 10 at link https://anonymous.4open.science/r/credal-dro). We will add this as practitioner guidance in Section 3.
>
> **Score choice**
> We agree that the score choice is important. It should be treated as a key modelling decision rather than something to stay invariant to. Two important considerations are: (i) keeping $\sup_{\xi\in\Xi_0} f_x(\xi)$ tractable, and (ii) aligning the bulk set geometry with the type of contamination we want to protect against. Appendix G.8 explains that a joint ellipsoid bulk set in $(X,Y)$ is ineffective for regression because it can hard-code the training $Y-X$ dependence; Appendix E.1 mentions that the box-type support assumptions used for existing TV-DRO methods (e.g. Jiang and Guan, 2018) can be overly conservative because they can allow every dimension to move independently for the adversary. This is supported by an additional ablation where we reran California housing with a box bulk set on $(X,Y)$. See Figure 3 at link https://anonymous.4open.science/r/credal-dro, where the MAE–CVaR trajectory for LV is less stable than that for our ellipsoid-$X$-interval-$Y$ construction, especially at small $\varepsilon$. This is consistent with the piecewise-linear optimisation target that a box bulk set implies, where small changes in $\varepsilon$ can essentially become tie-breakers between different extreme-point solutions.
>
> In practice, we advocate for the following *rule of thumb*: use Mahalanobis scores by default; use blockwise bulk sets to separate asymmetric or heterogeneous dimensions in problems such as regression; and only use box scores when independent coordinate-wise deviations are the intended adversary. We will add this to the main text. A general, unified decision theory for the bulk set geometry and the loss function would yield valuable future work, which we will further highlight in our concluding remarks.
>
> **Information in the complement set $\Xi/\Xi_0$**
> We agree that this set is not used in the optimisation target. However, this information is not overlooked but deliberately handled separately so that the optimisation target can stay finite. The risk certificate in Theorem 3.4 includes a tail term, and in the subsequent discussions, we also discuss how effective the bound is based on how much mass the out-of-sample contaminant $\tilde R$ places on the bulk vs the tail.
>
> **Bayesian-centre experiment under other shifts**
> We chose a spike contamination because our credal ambiguity set is designed for protecting against adversarial contaminations. We do not claim our method works best universally against any type of shift, which is unrealistic for any DRO framework. Our paper also includes geographic deployment shift in California housing and demographically shifted text classification in CivilComments. That said, we have implemented the Student-$t$-to-normal distributional shift (See Figure 2 at link https://anonymous.4open.science/r/credal-dro). Even in this regime of very smooth and structural distributional shift, our method outperforms KL-Empirical and KL-BAS$_{\text{PP}}$ and matches the best baselines at small $\varepsilon$ - it only gives larger OOS variances when $\varepsilon$ is too large ($>0.25$), which is intuitive because targeting a 25% arbitrary contamination is too conservative for this type of shift.

---

> > ### Author Rebuttal · Reviewer_X6QV · 2026-04-02
> >
> > Thanks for the detailed response. The authors addressed several questions regarding $\gamma$-sensitivity analysis, score choice, and the Bayesian-centre experiment under other shifts through experimental validation. After reading the response, my concerns have been resolved, and my curiosity has been satisfied. Regarding the question on "Information in the complement set $\Xi \setminus \Xi_0$", I accept the authors' explanation and agree that this issue does not affect the core contribution of the paper.

---

> > > ### Author Response · Authors · 2026-04-02
> > >
> > > We appreciate and are very encouraged by your acknowledgement that your concerns have been fully resolved. In light of your updated assessment, and of the overall review context, we would be very grateful if you might consider raising the score and supporting the paper in your final assessment.

---

### Decision · Program_Chairs · 2026-04-30

**Decision:**

Accept (spotlight)

**Comment:**

This paper addresses a key limitation of DRO under Huber contamination, where worst-case risk becomes infinite in unbounded settings. The proposed bulk-calibrated credal ambiguity sets restrict adversarial mass to a data-driven high-probability region, yielding a finite and tractable “bulk + tail” decomposition with efficient LP/SOCP formulations.

Reviewers agree that the work is technically sound, novel, and practically relevant, and that it establishes a compelling connection between DRO and imprecise probability. Empirical results demonstrate competitive robustness–accuracy trade-offs and strong computational efficiency.

Main concerns included sensitivity to hyperparameters and bulk geometry, theory–practice gaps, and presentation clarity. The authors addressed these through additional experiments, clearer positioning, and practical guidance (e.g., sensitivity analysis and recommendations for score functions and bulk geometry). Reviewers found these responses satisfactory and updated their assessments positively.

Overall, the contribution is meaningful and likely to influence future work. Remaining issues are primarily related to clarity, and the authors are urged to improve presentation in the camera-ready version.

**Recommendation: Accept**